# Host cell and viral protease targets of human SERPINs identified by in silico docking

Joaquin Rodriguez Galvan[1], Maren de Vries [ID][1], Shiraz Belblidia[1], Ashley Fisher[1,2], Rachel A Prescott[1], Keaton M Crosse [ID][1], Patrick Hearing [ID][3], Walter F Mangel [ID][4], Ralf Duerr [ID][1,2,5 ✉] & Meike Dittmann [ID][1 ✉]

## Abstract

**Serine protease inhibitors (SERPINs) are involved in various physiological processes and diseases, such as inflammation, cancer metastasis, and neurodegeneration. Their role in viral infections is poorly understood, as their expression patterns during infection and the range of proteases they target have yet to be fully characterized. Here, we show widespread expression of human SERPINs in response to respiratory virus infections, both in bronchioalveolar lavages from COVID-19 patients and in polarized human airway epithelial cultures. Using in silico docking of 10 SERPINs to 48 host proteases, we confirm known targets and predict new interactions. Protease activity assays validated selected interactions, confirming the newly predicted host targets for PAI-1 (*SERPINE1*) and PAI-2 (*SERPINB2*). PAI-1 inhibits cathepsin L, essential for SARS-CoV-2 maturation, and suppresses multi-cycle replication of both ancestral SARS-CoV-2 WA-1 and its variant Omicron BA.1. In addition, we identify PAI-2 as an antiviral SERPIN that reduces infectivity of human adenovirus 5 by directly inhibiting the adenoviral protease. Our study leverages in silico docking using full-length 3D protein structures to uncover new SERPIN targets, offering a range of candidate targets for therapeutic interventions.**

**Keywords** TMPRSS; Cathepsin; Plasminogen Activator Inhibitor 1; Plasminogen Activator Inhibitor 2; Adenovirus Protease
**Subject Categories** Microbiology, Virology & Host Pathogen Interaction; Post-translational Modifications & Proteolysis

## Introduction

Respiratory viruses are a group of clinically significant, highly heterogeneous viruses. Despite their differences in particle structure and replication strategies, many rely on a critical group of enzymes: proteases (Kemp et al, 1992). The origin of these proteases may be host or viral, and the roles they play in viral replication are multifarious (Aguilar and Buchholz, 2020; Kemp et al, 1992). The successful application of small-molecule protease inhibitors in treating HIV, HCV, and SARS-CoV-2 infections showcases the power of antiviral regimens that target proteases of viral origin (Amani and Amani, 2023; Lv et al, 2015; Scheel and Rice, 2013). Instead of or in addition to viral-encoded proteases, a number of viruses depend on host proteases to complete their life cycles (Bottcher-Friebertshauser et al, 2010; Essalmani et al, 2022; Klenk et al, 1975; Lenz et al, 2001; Ortmann et al, 1994; Volchkov et al, 1998); prominently, for cleavage of fusogenic viral glycoproteins, a process known as viral maturation (Klenk et al, 1975; Stieneke-Grober et al, 1992; Walls et al, 2020). Influenza A viruses (IAV), parainfluenzaviruses, and SARS-CoV-2 all rely on host proteases for maturation of hemagglutinin, fusion, or Spike glycoproteins, respectively (Bottcher-Friebertshauser et al, 2010; Essalmani et al, 2022; Klenk et al, 1975; Lenz et al, 2001; Ortmann et al, 1994; Stieneke-Grober et al, 1992; Volchkov et al, 1998). Cleavage occurs at specific sites (cleavage sites) to expose the fusion peptide and provide a hinge-like function to enable membrane merging during viral entry. Immature (uncleaved) particles are unable to cause infection. The specificity of host proteases to execute maturation depends on the amino acid motif at the glycoprotein cleavage site (Bottcher et al, 2006) and the glycoprotein structure adjacent to the cleavage site (Johnson et al, 2021; Vu et al, 2022).

The activity of host proteases is controlled at the protein level by host-encoded inhibitors. With over 1500 members encoded in the genomes of animals, plants, viruses, bacteria, and archaea, SERPINs constitute the largest known family of protease inhibitors (Gettins and Olson, 2016). The mechanism by which SERPINs inhibit target proteases is understood in molecular detail; it results in the formation of irreversible, covalent SERPIN:protease complexes (aka the "mousetrap" mechanism, Fig. EV1). A SERPIN's protease specificity is driven mainly by the fit of the SERPIN reactive center loop into the protease catalytic site (Fig. EV1A). The discovery of canonical SERPIN target proteases has historically focused on proteases in the blood or the brain (Gettins and Olson, 2016; Sanchez-Navarro et al, 2021), and the list of SERPIN targets is likely incomplete. The addition of SERPIN targets has historically occurred incrementally, with discoveries arising one at a time during investigations into specific diseases, rather than through

[1]New York University Grossman School of Medicine, Microbiology Department, New York, NY, USA. [2]New York University Grossman School of Medicine, Department of Medicine, New York, NY, USA. [3]Stonybrook University, Renaissance School of Medicine, Stonybrook, NY, USA. [4]Brookhaven National Laboratory, Biology Department, Upton, NY, USA. [5]New York University Grossman School of Medicine, Vaccine Center, New York, NY, USA. ✉E-mail: Ralf.Duerr@nyulangone.org; Meike.Dittmann@nyulangone.org

systematic discovery approaches. Hence, there is an opportunity to map comprehensive SERPIN target protease repertoires, particularly within organ-specific proteolytic environments, which remain poorly explored and understood.

To date, three SERPINs have been studied in the context of innate antiviral defense: PAI-1 (encoded by *SERPINE1*) against influenza viruses encoding hemagglutinin H1, Sendai virus, and SARS-CoV-2, through the inhibition of trypsin-like proteases by impeding proteolytic maturation of H1, F, or Spike, respectively (Dittmann et al, 2015; Rosendal et al, 2022); alpha-1-antitrypsin (encoded by *SERPINA1*) and antithrombin (encoded by *SERPINC1*) against SARS-CoV-2, likely through the inhibition of TMPRSS2, by reducing maturation of Spike, although direct inhibition of TMPRSS2 protease activity or formation of "mouse-trap" complexes by either SERPIN was not shown (Rosendal et al, 2022). In addition to innate defense SERPINs, SERPINs such as C1-inhibitor (C1-INH, encoded by *SERPING1*) and one virus-derived SERPIN have been analyzed for efficacy in treating SARS-CoV-2 (Urwyler et al, 2023). Beyond these studies, the role of other SERPINs in inhibiting host proteases during viral infections is unclear, and no SERPIN to date has been shown to inhibit a protease of viral origin.

Here, we identify SERPINs that are upregulated in airway epithelial cells in response to respiratory viral infections. We devised in-silico docking to predict binding between ten SERPINs that are upregulated upon viral infection and 48 airway proteases based on both partners' 3D full-length protein structures. All tested SERPINs, despite differences in their target repertoires, were predicted to bind previously unknown protease targets. We validated 8 predicted binders and non-binders for their protease inhibition (or lack thereof) in vitro. Among them, the well-studied SERPIN PAI-1 serves as a direct inhibitor of both active TMPRSS2 and cathepsin L (CTSL), with antiviral relevance for SARS-CoV-2 multi-cycle growth. Finally, we discover and validate a direct-acting antiviral SERPIN, in PAI-2, inhibiting the adenovirus protease (AVP). Our findings underscore the importance of expanding the understanding of SERPIN-protease interactions, prove the feasibility of in-silico docking strategies for SERPIN target discovery, and highlight the potential of noncanonical interactions as targets for the development of effective antiviral interventions in the context of respiratory viruses.

## Results

### SERPINs are differentially expressed in individuals with COVID-19 and in response to respiratory virus infection in a model of the human airway epithelium

Certain SERPINs are upregulated in response to specific stimuli, including interferons, tumor necrosis factors, or tumor growth factors (Dong et al, 2002; He et al, 2010; Prada et al, 1998), all of which can be produced during viral infections (McNab et al, 2015; Mirzaei and Faghihloo, 2018; Seo and Webster, 2002). To investigate SERPIN expression in viral infections, we analyzed a published single-cell RNA sequencing (scRNA-seq) dataset of bronchoalveolar lavage fluids (BALF) from individuals with mild or severe COVID-19, as well as uninfected control individuals (Liao et al, 2020) (Fig. 1A). In brief, scRNA-seq determines the gene expression patterns of specific cell types within a mixed cell population, such as found in BALF, by analyzing the RNA from individual cells. Clustering algorithms group cells with similar expression profiles together. Each cluster of cells is analyzed for marker gene expression to identify specific cell types. Known cell-type-specific markers are used to annotate the clusters, revealing the identity of each cell type within the mixed population of the BALF. The technique thus allowed us to analyze cell-type-specific SERPIN transcript levels. We observed SERPIN upregulation in immune cells such as macrophages, T cells, B cells, and NK cells, consistent with the literature, and consistent with the well-established role of SERPINs as innate and adaptive immune response regulators (Bouton et al, 2023). We also observed a high number of upregulated SERPINs in epithelial cells, particularly in samples from individuals with severe COVID-19 (Fig. 1B). This suggests that airway epithelial cells, which are the major cell type supporting SARS-CoV-2 replication (Chu et al, 2020; Hui et al, 2020; Hui et al, 2022; Puelles et al, 2020; Zou et al, 2020), are a source of SERPIN production during SARS-CoV-2 infection. It also suggests that SERPINs could be playing a role in the cell-intrinsic antiviral response within epithelial cells, a function of SERPINs that has not been extensively studied.

In epithelial cells, a group of SERPINs (including *SERPINB1*, *SERPINB3*, *SERPINB6*, *SERPING1*) was present at baseline and further upregulated in individuals with COVID-19 (Fig. 1C). Others were present at low levels in control individuals and upregulated in COVID-19 individuals (*SERPINB2*, *SERPINB5*, *SERPINB11*, and *SERPINF1*). Yet others were present at low levels irrespective of infection status, and one was downregulated (*SERPINA1*). These observed disparities in expression patterns suggest that specific SERPINs are controlled differentially.

Correlation analyses with SERPINs and gene sets involved in inflammatory pathways revealed that mRNA levels for most SERPINs positively correlated with canonical TNF-alpha and interferon-stimulated genes (Fig. EV2), indicating that they are upregulated in concert with virus-induced inflammation. Analysis of protease expression in epithelial cells revealed that select proteases were present irrespective of the infection status, while others appeared upregulated upon infection (Fig. 1C). This highlights the complexity of the proteolytic landscape produced by airway epithelial cells during SARS-CoV-2 infection.

Given that the mRNA levels of multiple SERPINs are elevated during SARS-CoV-2 infection in BALF-derived epithelial cells (Fig. 1), we next used polarized human airway epithelial cultures (HAEC) to study the expression patterns of SERPINs in a controlled environment in vitro. Softened the language. The HAEC model closely mimics the human bronchial epithelium, encompassing various cell types (basal, ciliated, and secretory), architectural features, and a secreted extracellular environment that includes mucus and proteases (Fig. 2A,B) (Prescott et al, 2023). We infected HAEC with a panel of clinically relevant respiratory viruses (Fig. 2B), all of which feature proteolytic steps in their life cycles. IFN-beta served as a positive control for expression of interferon-stimulated genes, as evidenced by the IFN-inducible expression of *IFITM3* (Anafu et al, 2013) (Fig. 2C; Table EV1).

All SERPINs, with the exception of *SERPINA1*, exhibited an upregulation of at least twofold under at least one experimental condition (Fig. 2C,D), suggesting that more SERPINs than previously reported are relevant in the defense against viral infections. A subset of SERPINs was consistently upregulated in response to all six viruses,

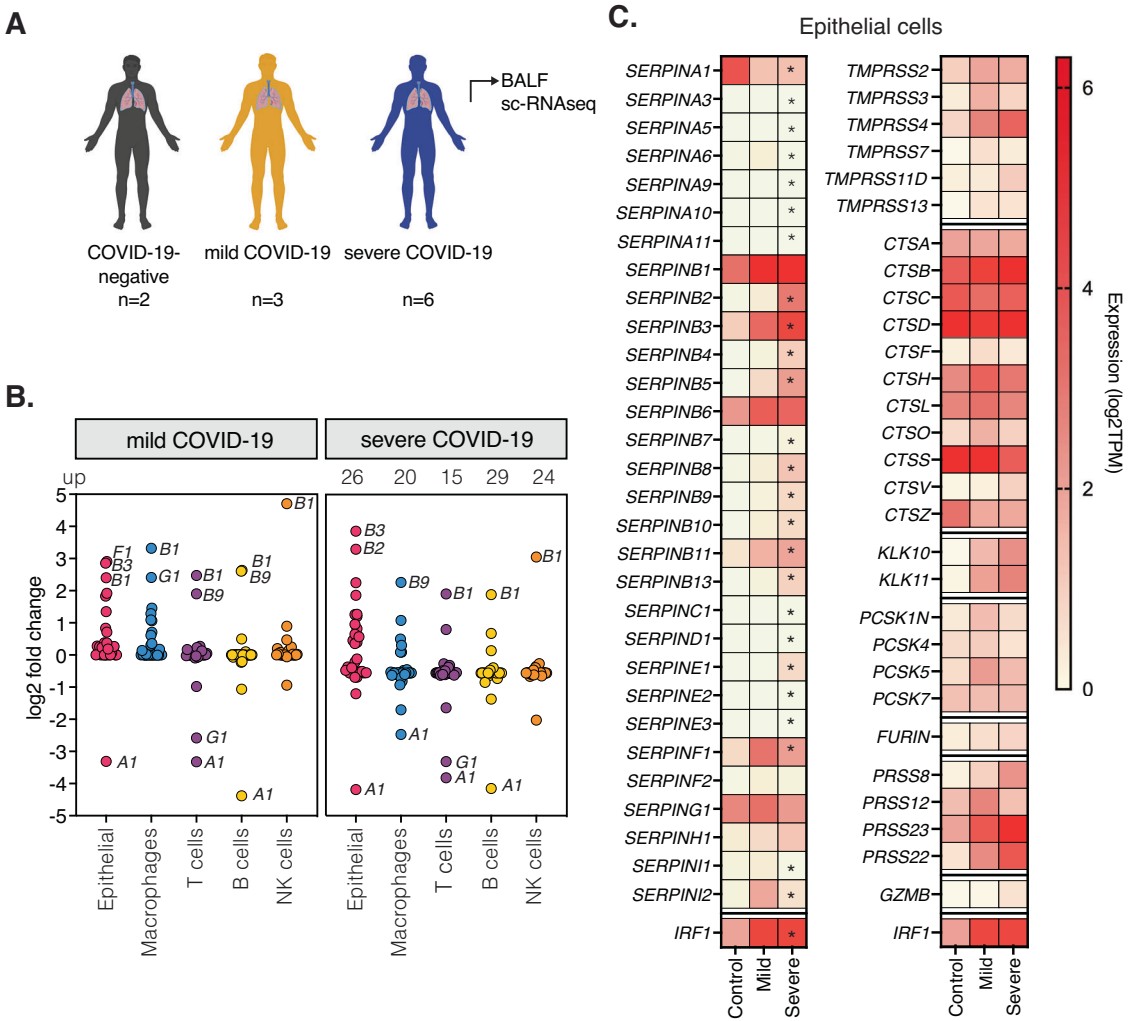

**Figure 1. Single-cell transcriptional analysis in bronchioalveolar lavage fluids (BALF) from COVID-19-negative and -positive individuals.**

(A) sc-RNAseq data (GSE145926(Liao et al, 2020)) from BALF of COVID-19-negative ($n = 2$), and COVID-19-positive individuals with mild ($n = 3$) or severe ($n = 6$) symptoms. (B) Fold change of SERPIN expression across cell types relative to control individuals. The number of SERPINs significantly upregulated or downregulated compared to the housekeeping gene B2M shown above. *P* values and labels are in Dataset EV1. (C) Expression of SERPINs, proteases, and prototype interferon-stimulated gene IRF1 in the epithelial cell cluster. TPM Transcripts per million. Only proteases over 0.5 log2TPM in any condition are shown. TMPRSS transmembrane protease, serine, CTS cathepsin, KLK kallikrein, PCSK pro-protein convertase subtilisin/kexin, PRSS serine protease, GZMB granzyme, IRF1 interferon regulatory factor 1. Statistical analysis by two-way ANOVA and Kruskal–Wallis test *$P < 0.05$ upregulation or downregulation as per analysis in Dataset EV1.

exemplified by *SERPING1*, suggesting their potential role in a broad antiviral response (Fig. 2C,D). Other SERPINs exhibited differential expression patterns in response to specific viruses, as seen most notably with *SERPINA7*, *SERPINA10,* and *SERPINE2*, indicating distinct regulatory mechanisms and potential selectivity against particular viruses. Different from all other SERPINs, *SERPINA1* was expressed at lower levels upon infection and IFN-treatment as compared to baseline. We had also found *SERPINA1* expressed at lower levels in BALF of COVID-19 individuals (Fig. 1). This suggests that *SERPINA1* mRNA expression is downregulated upon SARS-CoV-2 infection and in response to IFN, rather than lacking at baseline to allow for SARS-CoV-2 infection, as proposed by others (Rosendal et al, 2022). Finally, only approximately half of the SERPINs upregulated during viral infection were also upregulated upon IFN stimulation (Fig. 2D), indicating the involvement of other and/or additional gene regulatory mechanisms.

Together, our gene expression data demonstrate that most SERPINs are upregulated in response to respiratory virus infections in concert with antiviral effectors, particularly in epithelial cells.

## In-silico screen predicts noncanonical SERPIN-protease pairs relevant to viral life cycles

Mapping the full range of proteases targeted by SERPINs remains a challenge, limiting our understanding of SERPIN involvement in human disease. Experimental methods, like SERPIN tagging for pulldown assays and mass spectrometry, require genetic manipulation of model systems. However, many in vitro models fail to replicate the diverse proteolytic landscape observed in vivo, limiting the effectiveness of these techniques for target discovery. In-silico approaches using motif searches are restricted by known protease

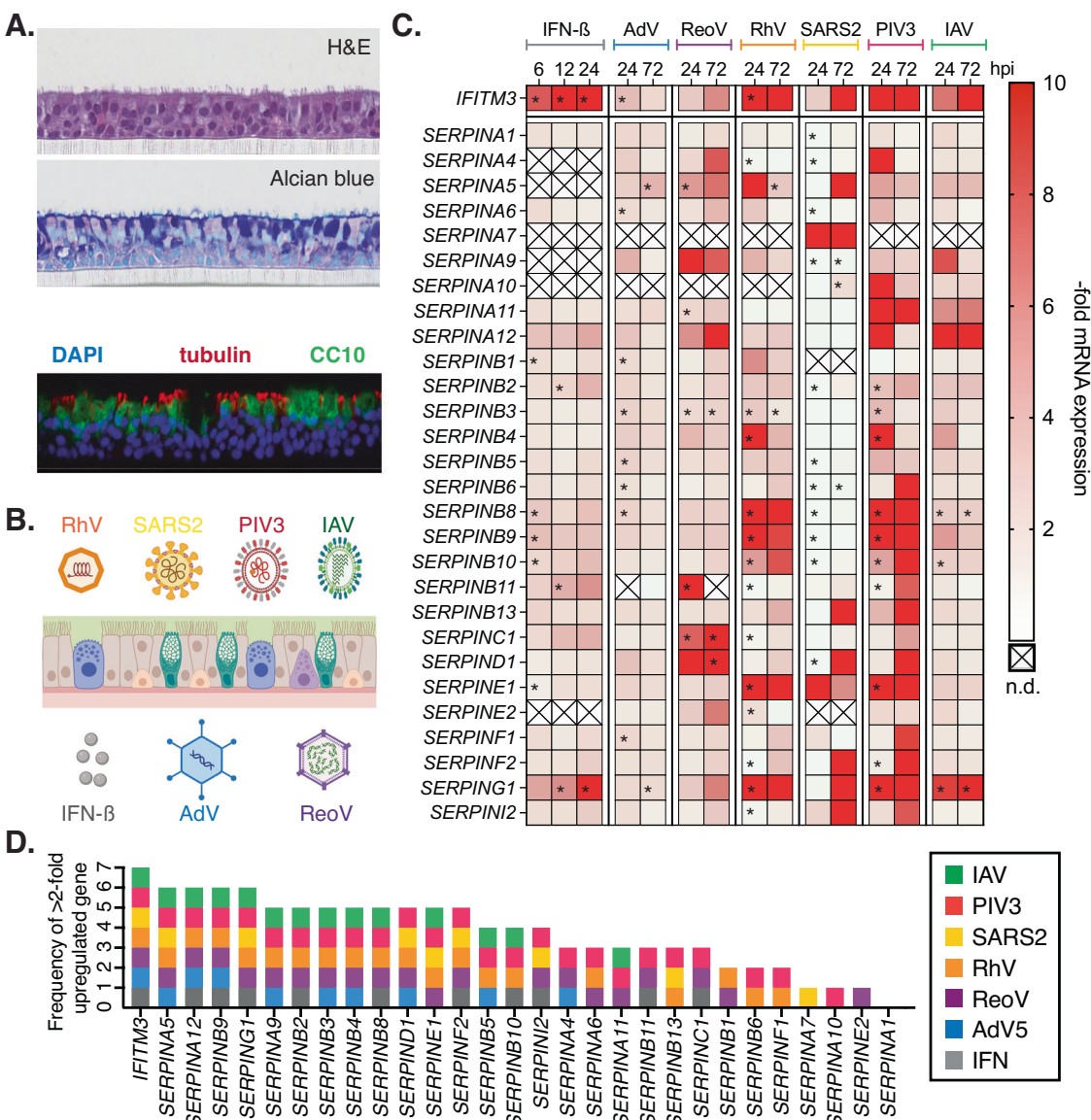

**Figure 2. SERPIN expression in virus-infected or interferon-treated human airway epithelial cultures (HAEC).**

(A) Analysis of HAEC morphology, with Hematoxylin & Eosin (H&E) staining, Periodic Acid–Schiff (PAS)-Alcian blue staining, and immunofluorescence staining for cell type markers (tubulin in red for ciliated cells, CC10 in green for secretory cells). (B) Schematic representation of respiratory viruses and a transwell with polarized HAEC. Apical infection with human rhinovirus A (RhV), influenza A/California/09 H1N1 virus (IAV), human parainfluenzavirus 3 (HPIV3), or SARS-CoV-2 WA-1 (SARS2); basolateral infection/treatment with human adenovirus 5 (AdV5), human reovirus (ReoV), or interferon-beta (IFN-ß). (C) The total RNA was isolated from lysed cultures at specific time points post-infection, and transcripts were quantified using RT-qPCR. SERPIN and prototype interferon-stimulated gene *IFITM3* mRNA levels shown as fold change over uninfected cultures over time. Data from $n = 3$ biologically independent experiments. *P* values in Table EV1. *Significant (*P* < 0.05) upregulation or downregulation. (D) Frequency of >twofold-upregulated genes from (C) per experimental condition. n.d. not detectable. Source data are available online for this figure.

motifs and do not integrate structural factors driving SERPIN-protease interactions.

To overcome this limitation, we used the software HADDOCK (High Ambiguity Driven protein-protein Docking (de Vries et al, 2010; van Zundert et al, 2016)) to predict interactions between SERPINs and proteases. HADDOCK predicts how two or more molecules interact to form a binding complex (in our case, the binding complex depicted in Fig. EV1A). The process begins with the full-length 3D structures of the protease and SERPIN, which are either experimentally solved or homology-modeled. Using experimental data or predictions,

HADDOCK fits the molecules together, refines the interaction sites, and evaluates multiple configurations to select the most likely models of the interaction. The final output is a set of possible models showing how the molecules might interact, in an isolated environment, with predicted binding energies for each model, where lower scores indicate stronger interactions. Since binding is the first step of the SERPIN "mousetrap" mechanism, we use these predictions as a proxy to assess SERPIN-protease inhibition.

We evaluated our approach by distinguishing known SERPIN binders from non-binders using Leukocyte elastase inhibitor (LEI,

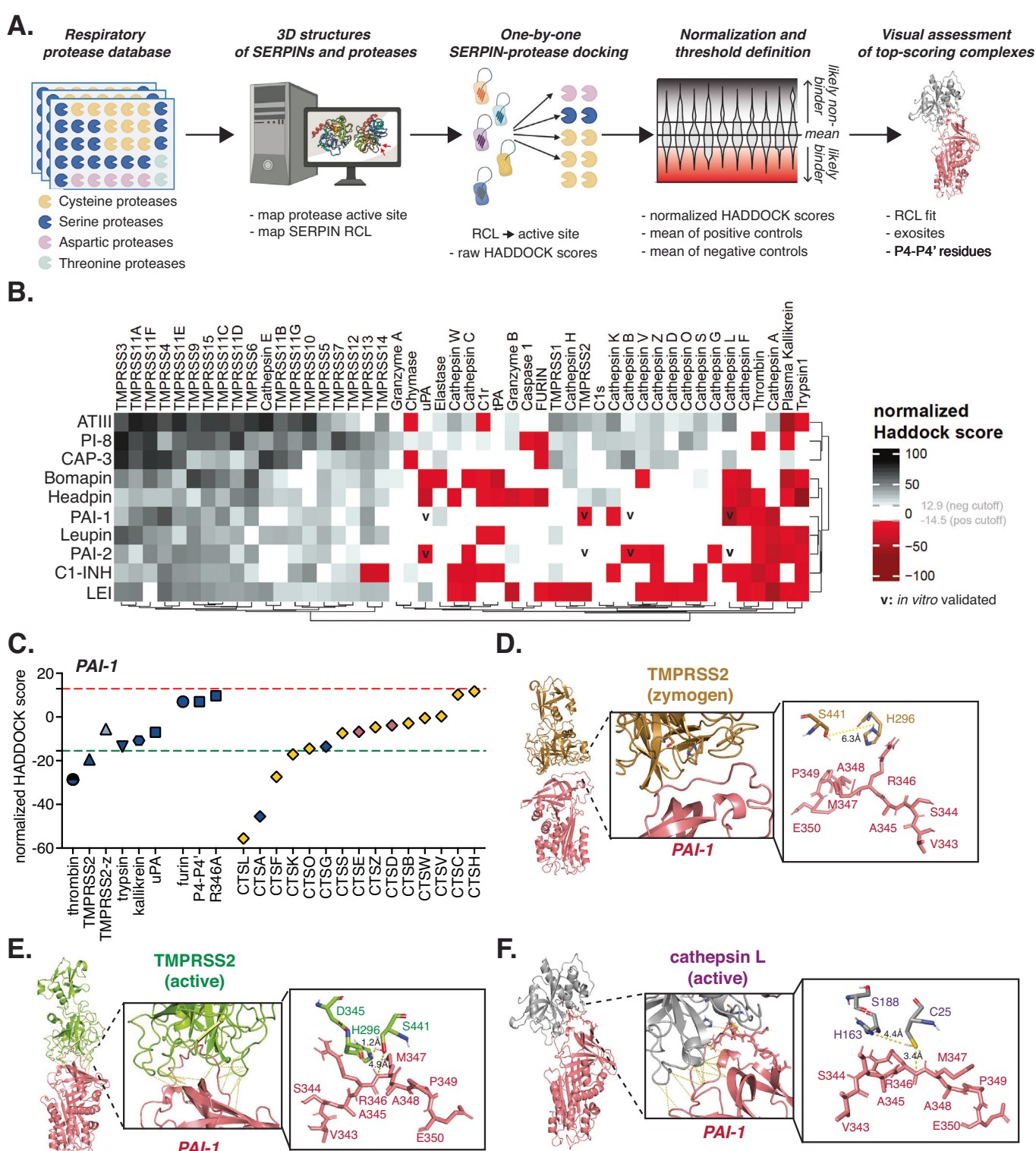

encoded by *SERPINB1*). LEI binds and inhibits Elastase and Cathepsin G (CTSG), but not Granzyme A (Torriglia et al, 2017). The top-scoring configurations show LEI's RCL inserting deeply into CTSG's active site but not Granzyme A's, with CTSG's nucleophile S181 and base H44 positioned close to LEI's M345↓L346, unlike Granzyme A's S184 and H41, which are further

apart from LEI F343↓C344 (Fig. EV3A,B). Predicted binding energies are low for LEI:CTSG but high for LEI:Granzyme A (Fig. EV3C), indicating favorable binding to CTSG and unfavorable binding to Granzyme A, consistent with the literature. High raw HADDOCK scores for LEI:Elastase and low scores for a mutated LEI RCL (see "Methods") with CTSG further validate HADDOCK's

◄ **Figure 3. In-silico binding screen of 10 SERPINs with 48 host respiratory proteases.**

(A) Schematic of in-silico docking screen workflow. A panel of 48 respiratory proteases and 10 SERPINs upregulated upon viral infection were assembled. 3D structures were generated, with annotated protease active sites and SERPIN reactive center loops (RCLs). SERPIN-protease docking was performed using HADDOCK, guiding the RCL into the protease active site, and obtaining raw HADDOCK scores to reflect predicted binding energies. The dataset was normalized to the mean, and thresholds set for high-confidence in-silico binders (mean of positive controls) and high-confidence in-silico non-binders (mean of negative controls). Top-scoring complexes of interest were visually assessed for 3D RCL fit, exosite formation, and the positioning of protease active residues relative to SERPIN P4-P4' residues. (B) Heatmap of docking results, with z-scores centered to the mean of control SERPIN-protease pairs and normalized for each SERPIN. The darker the red, the more favorable the binding energies. LEI leukocyte elastase inhibitor encoded by *SERPINB1*, PAI-2 plasminogen activator inhibitor 2 encoded by *SERPINB2*, Leupin encoded by *SERPINB4*, PI-8 protease inhibitor 8 encoded by *SERPINB8*, CAP-3 cytoplasmic anti-protease 3 encoded by *SERPINB9*, Bomapin encoded by *SERPINB10*, Headpin encoded by *SERPINB13*, ATIII antithrombin 3 encoded by *SERPINC1*, PAI-1 plasminogen activator inhibitor 1 encoded by *SERPINE1*, C1-INH C1 inhibitor encoded by *SERPING1*. V validated pairs in vitro as shown in Fig. 4. (C) Normalized HADDOCK scores of PAI-1 with its positive controls (thrombin, active TMPRSS2, trypsin, kallikrein, uPA), negative controls (furin, uPA to PAI-1 P4-P4' alanine substitutions, and uPA to PAI-1 P1 mutant R346A), the TMPRSS2 zymogen (TMPRSS2-z), and cathepsins. Green dotted line (mean of all positive controls in the in-silico screen) is the threshold for high-confidence binders; red dotted line (mean of all negative controls in the in-silico screen) is the threshold for high-confidence non-binders. Symbols: triangles, trypsin-like proteases; circles, subtilisin-like proteases; split circle, thrombin; hexagon, kallikreins; diamonds, cathepsins; light blue inactive serine protease, inactive; blue, serine proteases; yellow, cysteine proteases; salmon, aspartic proteases. Blue, serine proteases; yellow, cysteine proteases; red, aspartic proteases. (D-F) Docking structures of top-scoring complexes for PAI-1 with TMPRSS2 (zymogen), PAI-1 with TMPRSS2 (active) and CTSL, respectively. Source data are available online for this figure.

ability to identify SERPIN binders and non-binders and test in-silico-mutated interactions.

To explore the full target range of SERPINs in the respiratory epithelium, we selected 48 potential respiratory target proteases from the LungMAP human expression database (Ardini-Poleske et al, 2017b) (Fig. 3A). We created a panel of 10 SERPINs expressed upon viral infection (Figs. 1 and 2) and used HADDOCK to generate in-silico complexes with known target and non-target proteases for each. If no non-target proteases were known, we used an RCL mutant approach to ablate binding in-silico (see "Methods"). We docked each of the ten SERPINs with each of the 48 respiratory proteases, generating raw HADDOCK scores for each of the 480 combinations (Fig. 3A). We then normalized the raw HADDOCK scores to the means of each SERPIN's averaged positive and negative control HADDOCK scores (Figs. EV3D and 3A). This enabled us to set thresholds for high-confidence in-silico binders (the mean of positive controls) and for high-confidence in-silico non-binders (the mean of negative controls; Fig. EV3D).

The heatmap of normalized HADDOCK scores uses color-coding to indicate predicted binding energies: red for low scores (high-confidence binding) and gray for high scores (high-confidence non-binding, Fig. 3B). All SERPINs showed previously unknown predicted binders and non-binders. Unsupervised clustering of the data revealed unprecedented patterns in SERPIN-protease in-silico binding selectivity. ATIII (*SERPINC1*), PI-8 (*SERPINB8*), and CAP-3 (*SERPINB9*) shared similar protease binding patterns and formed a distinct cluster characterized by a predicted narrow protease target range. Bomapin (*SERPINB10*) and Headpin (*SERPINB13*) grouped together in another cluster, with a wider and largely overlapping protease target range. The final cluster, consisting of PAI-1 (*SERPINE1*), Leupin (*SERPINB4*), PAI-2 (*SERPINB2*), C1-INH (*SERPING1*), and LEI (*SERPINB1*), also displayed a broad range of predicted targets, with LEI being the most promiscuous.

Among the proteases, plasma kallikrein, thrombin, and trypsin emerged as the most promiscuous, predicted to bind to the majority of SERPINs. In contrast, many members of the TMPRSS family clustered separately from all other proteases, being predicted as high-confidence non-binders and thus suggesting that they are resistant to most SERPINs. Exceptions included TMPRSS13 and TMPRSS14, predicted to be bound by C1-INH; TMPRSS1, predicted to be bound by LEI; and TMPRSS2, predicted to be bound by both PAI-1 and LEI.

PAI-1 is arguably one of the most-studied SERPINs, with over 15,000 publications. It is also the first-described antiviral SERPIN (Dittmann et al, 2015). Our screen revealed favorable scores for PAI-1's canonical targets uPA/tPA (Thorsen et al, 1988), along with additional previously shown targets trypsin (Dittmann et al, 2015), kallikrein (Mikolajczyk et al, 1999), and thrombin (Urano et al, 2000), and, previously suggested but not directly shown, TMPRSS2 (Dittmann et al, 2015; Rosendal et al, 2022) (Fig. 3C). While the scores for PAI-1-complexes with some of these known targets did not fall below the high-confidence threshold, they were clearly distinct from the negative controls, wild-type PAI-1 with furin and both mutant PAI-1 (P4-P4' alanine substitutions and single P1 point mutation R346A) with uPA. Of note, the PAI-1:TMPRSS2 complex fell below the high-confidence threshold, whereas the PAI-1 complex with the inactive TMPRSS2 zymogen did not (Fig. 3C). This suggested that TMPRSS2 must be in its active form to bind PAI-1 effectively. HADDOCK allows for visual assessment of the modeled complexes, and we next analyzed the in-silico PAI-1:active TMPRSS2 complex and the PAI-1:TMPRSS2 zymogen complex (Fig. 3A,D,E). We found the PAI-1 RCL inserted into the TMPRSS2 active site for active TMPRSS2, but not for the TMPRSS2 zymogen (Fig. 3D,E). R346 is the canonical P1 residue for the PAI-1-uPA inhibitory mechanism. In the PAI-1:active TMPRSS2 complex, the nucleophilic S441 was positioned in close proximity for attack on PAI-1 R346 (Fig. 3E), thus setting the stage for the next step in the SERPIN "mousetrap" mechanism.

PAI-1 also docked favorably with a number of select CTS, some of which are cysteine proteases (Fig. 3C). While other SERPINs have been shown to inhibit cysteine proteases (Kantyka and Potempa, 2011; Masumoto et al, 2003; Torriglia et al, 2017; Viswanathan et al, 2012), the predicted PAI-1:cysteine protease family binding was unexpected, as PAI-1 was previously thought to have specificity for serine proteases exclusively (Sillen and Declerck, 2020). Among the predicted binding cathepsins was CTSL, which, along with TMPRSS2, is a key protease for SARS-CoV-2 Spike maturation. Visual assessment of the PAI-1:CTSL complex showed that the PAI-1 RCL inserted far into the CTSL active site (Fig. 3F). The nucleophilic C25 of CTSL was also positioned in close

proximity to canonical P1 PAI-1 R346 (Fig. 3F), suggesting that this may be the residue that is attacked should the "mousetrap" mechanism occur. Thus, while the modeling cannot test for the "mousetrap" mechanism, it showcases the usefulness of in-silico docking in evaluating the 3D positioning of critical residues.

Finally, we examined published protease target motifs to understand the specificity of proteases for particular substrate motifs, determined by the composition of their specificity pockets within the catalytic center. TMPRSS2 exhibited a preference for an R at P1 and a polar residue at P3, aligning with PAI-1's core RCL motif (Fig. EV4). Conversely, CTSL's motif flexibility was evident, with no clear consensus motif. However, the PAI-1 RCL core displayed similarities to both the Spike S2' motif and the minimal motif of CTSL, suggesting potential compatibility with CTSL specificity pockets.

In summary, the in-silico docking screen offers a comprehensive investigation into SERPIN targets within a specific proteolytic environment, overcoming obstacles for testing these interactions in the wet lab. We identify numerous potential candidates for antiviral SERPINs warranting further mechanistic investigation. Our data suggests that TMPRSS2 needs to be in its active form to be inhibited by PAI-1, which was previously unknown, and identifies CTSL as a potential target. We show that the 3D fit is crucial for SERPIN target protease recognition. Finally, we uncover previously unrecognized patterns in predicted SERPIN binding selectivity for airway host proteases hijacked by respiratory viruses.

## In vitro protease activity assays validate distinct protease targets for *SERPINB2* (encoding plasminogen activator inhibitor 2, PAI-2) and *SERPINE1* (encoding plasminogen activator inhibitor 1, PAI-1)

Our in-silico screen identified protease candidates as binders and non-binders for each SERPIN examined. To validate these predictions, we performed fluorometric protease activity assays for two SERPINs (PAI-1, PAI-2) and four proteases (uPA, TMPRSS2, CTSL, and CTSB). Fluorometric protease activity assays use synthetic reporter-labeled peptide substrates, which undergo changes in absorbance or fluorescence upon cleavage by recombinant proteases (Fig. 4A).

CTSL and CTSB represent the primary cathepsins involved in viral glycoprotein processing (Scarcella et al, 2022). Our docking analysis suggested CTSL as a PAI-1 high-confidence binder, while CTSB was suggested as likely a non-binder (Fig. 3C), despite both CTS sharing the same minimal consensus target motif, F-R (Fig. EV4C). Thus, we assessed CTSB protease activity with and without PAI-1, alongside uPA, TMPRSS2, and CTSL. While PAI-1 reduced the reaction rates of uPA, TMPRSS2, and CTSL, as expected, CTSB remained unaffected (Fig. 4A,E), consistent with our in-silico predictions. PAI-2 is a known inhibitor specific to uPA/tPA; our in-silico data confirmed PAI-2 as a uPA/tPA binder and predicts CTSB as an additional PAI-2 binder and, different from PAI-1, TMPRSS2, and CTSL as PAI-2 non-binders (Fig. 3B). Protease activity assays with recombinant PAI-2 confirmed these exact PAI-2 specificities (Fig. 4B).

We next tested whether PAI-1 physically inhibits both proteases critical for SARS-CoV-2 maturation, active TMPRSS2 and CTSL, through the canonical covalent "mousetrap" mechanism (as depicted in Fig. EV1B–D). We incubated recombinant active PAI-1 or heat-

inactivated PAI-1 with active uPA, TMPRSS2, or CTSL. After incubation, proteins were separated via SDS-PAGE, revealing high-molecular weight bands corresponding to predicted complexes (about 75 kDa for PAI-1:uPA, PAI-1:TMPRSS2, and PAI-1:CTSL), indicative of the SERPIN "mousetrap" mechanism (Fig. 4C–E). Complex formation was specific to conditions with both partners present, with PAI-1:CTSL complexes forming a pH of 6.5, resembling early endosomal conditions; however, the complex did not form at pH 5.5 resembling a later endosome, moreover PAI-1 decreased its molecular weight, suggesting its partial digestion by CTSL (Fig. 4F). PAI-1:CTSL high-molecular weight complexes did not form in the presence of heat-inactivated PAI-1.

In all, the in vitro validation results highlight the power of in-silico docking for proposing previously unknown SERPIN targets and non-targets. We confirm active TMPRSS2 and highlight CTSL as a previously unrecognized direct target of PAI-1 and show that CTSL inhibition occurs via the canonical "mousetrap" mechanism.

## PAI-1 inhibits multi-cycle SARS-CoV-2 replication and spike maturation

Next, we determined the antiviral potential of PAI-1 for SARS-CoV-2, given that PAI-1 inhibits both critical maturation proteases: TMPRSS2 plays a crucial role as the Spike S2'-cleaving protease for SARS-CoV-2 during viral release (in *cis*) or entry (in *trans*), and CTSL cleaves at SARS-CoV-2 Spike at S2' during endosomal entry, which is the alternative route for SARS-CoV-2 entry. To interrogate both of those proteolytic pathways experimentally, we tested both ancestral SARS-CoV-2 WA-1, which relies predominantly on TMPRSS2, and the variant Omicron BA.1, which relies predominantly on CTSL for Spike maturation.

Addition of active PAI-1 to the culture medium of Calu-3 cells significantly reduced the multi-cycle growth of both SARS-CoV-2 WA-1 and Omicron BA.1 compared to the buffer control and heat-inactivated (HI) PAI-1, achieving near-complete inhibition (Fig. 5A–D).

Conversely, adding an anti-PAI-1 antibody to deplete endogenous PAI-1 increased the multi-cycle growth of both SARS-CoV-2 WA-1 and Omicron BA.1, although this increase was only statistically significant for WA-1 (Fig. 5A–D). Of note, the effect of PAI-1 depletion was not as dramatic as the effect of recombinant PAI-1 addition, possibly because the depletion remained incomplete. Next, we assessed SARS-CoV-2 WA-1 infectious titers from the supernatants of Calu-3 cells infected in the presence of buffer control or recombinant PAI-1 at 48 h post-infection via $TCID_{50}$ assay and found that infectious titers were significantly reduced by 1000-fold (Fig. 5E).

To determine Spike maturation in the presence of PAI-1, we next co-transfected BHK cells with Spike WA-1 614G, TMPRSS2, and PAI-1 (Fig. 5F). While BHK cells express furin for the priming S1/2 cleavage, they lack proteases for the essential S2' cleavage, detected only with TMPRSS2 overexpression. The presence of PAI-1 reduced S2' cleavage. Additionally, in a parallel experiment, co-expression of Spike WA-1 614G with TMPRSS2 as the S2'-executing protease, along with the addition of recombinant PAI-1 (rPAI-1) to the cell supernatant, led to a reduction in Spike S2' cleavage (Fig. 5G). This suggests that PAI-1-mediated TMPRSS2 inhibition contributes to the observed reduction in SARS-CoV-2 multi-cycle replication.

Overall, our findings reveal a previously unappreciated PAI-1 target, CTSL, relevant to viral lifecycles and suggest a protective effect during SARS-CoV-2 infection, underscoring the expanding

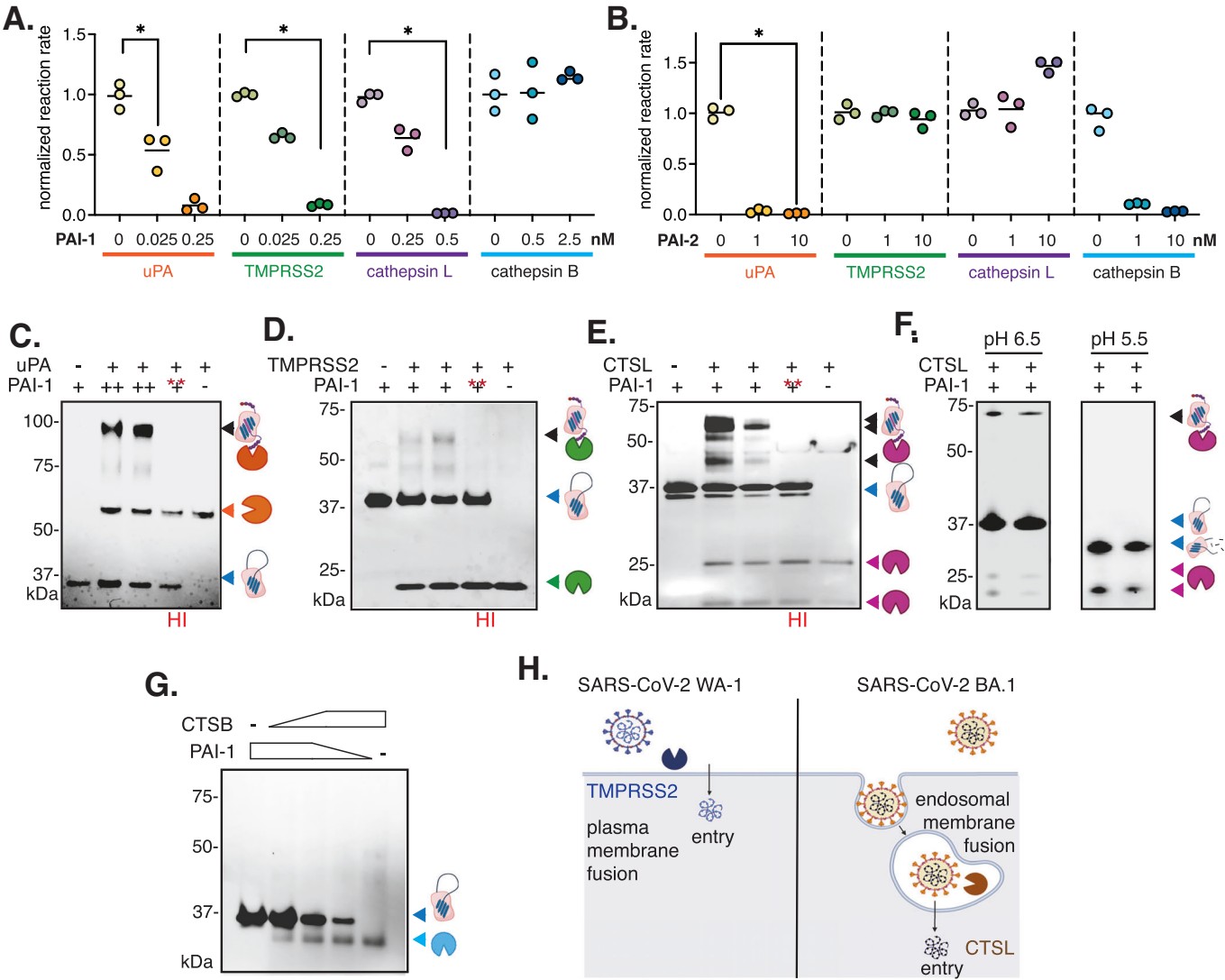

**Figure 4.    In vitro validation of select SERPIN-protease pairs from in-silico docking screen.**

(A, B) Normalized reaction rates for uPA, TMPRSS2, cathepsin L, and cathepsin B in the absence or presence of PAI-1 (A), or PAI-2 (B) from in vitro fluorometric assays. Statistical analysis by One-way ANOVA and Kruskal–Wallis test, *$P < 0.05$. Exact $P$ values in (A). uPA, 0 vs. 0.025 nM $P = 0.3594$, 0 vs. 0.25 nM $P = 0.0146$; TMPRSS2, 0 vs. 0.025 nM $P = 0.3594$, 0 vs. 0.25 nM $P = 0.0146$; CTSL, 0 vs. 0.025 nM $P = 0.3594$, 0 vs. 0.25 nM $P = 0.0146$; CTSB, 0 vs. 0.025 nM $P > 0.9999$, 0 vs. 0.25 nM $P = 0.5934$. Exact $P$ values in (B). uPA, 0 vs. 1 nM $P = 0.3594$, 0 vs. 10 nM $P = 0.0146$; TMPRSS2, 0 vs. 1 nM $P > 0.9999$, 0 vs. 10 nM $P = 0.9121$; CTSL, 0 vs. 1 nM $P > 0.9999$, 0 vs. 10 nM $P = 0.0722$; CTSB, 0 vs. 1 nM $P = 0.3594$, 0 vs. 10 nM $P = 0.0146$. Mean ± SD, $n = 3$ replicates. Raw data in Dataset EV3. (C) SDS-PAGE and silver stain of mixed recombinant active uPA (32 kDa, not visible) and PAI-1 (43 kDa). (D) SDS-PAGE and silver stain of mixed recombinant active TMPRSS2 (31 kDa) and PAI-1 (43 kDa). (E) SDS-PAGE and silver stain of mixed recombinant cathepsin L (32 kDa) and PAI-1 (43 kDa). *Denotes use of PAI-1 inhibitor triplaxinin and PAI-1 cleavage products. (F) SDS-PAGE and silver stain of mixed recombinant cathepsin L with PAI-1, reactions were conducted at pH 6.5 and pH 5.5. (G) SDS-PAGE and silver stain of mixed recombinant cathepsin B with PAI-1. (H) Schematic of importance of activity of TMPRSS2 and cathepsin L in SARS-CoV-2 entry by lineage. Source data are available online for this figure.

body of literature on SERPIN family members inhibiting viruses indirectly by blocking essential host proteases.

## SERPINB2 (encoding plasminogen activator inhibitor 2, PAI-2) is a direct-acting inhibitor of adenovirus protease

Both host- and virus-encoded (Urwyler et al, 2023) SERPINs can block essential host proteases to achieve antiviral function. However, no SERPIN to date has been shown to inhibit a virus-encoded protease. Small molecule inhibitors for virus-encoded proteases are successful therapeutics for HIV, HCV, and SARS-CoV-2. Adenovirus protease (AVP) is another attractive antiviral target, as it plays a crucial role in the viral life cycle by catalyzing viral maturation, a prerequisite for the infectivity of progeny particles (Mangel and San Martin, 2014). In addition, AVP is highly conserved among the numerous serotypes of human AdV. However, while inhibitors have been found that can inhibit the AVP enzyme in vitro, no inhibitor has yet been identified that can reduce viral titers during viral infections in cells, despite decades of research (Mac Sweeney et al, 2014).

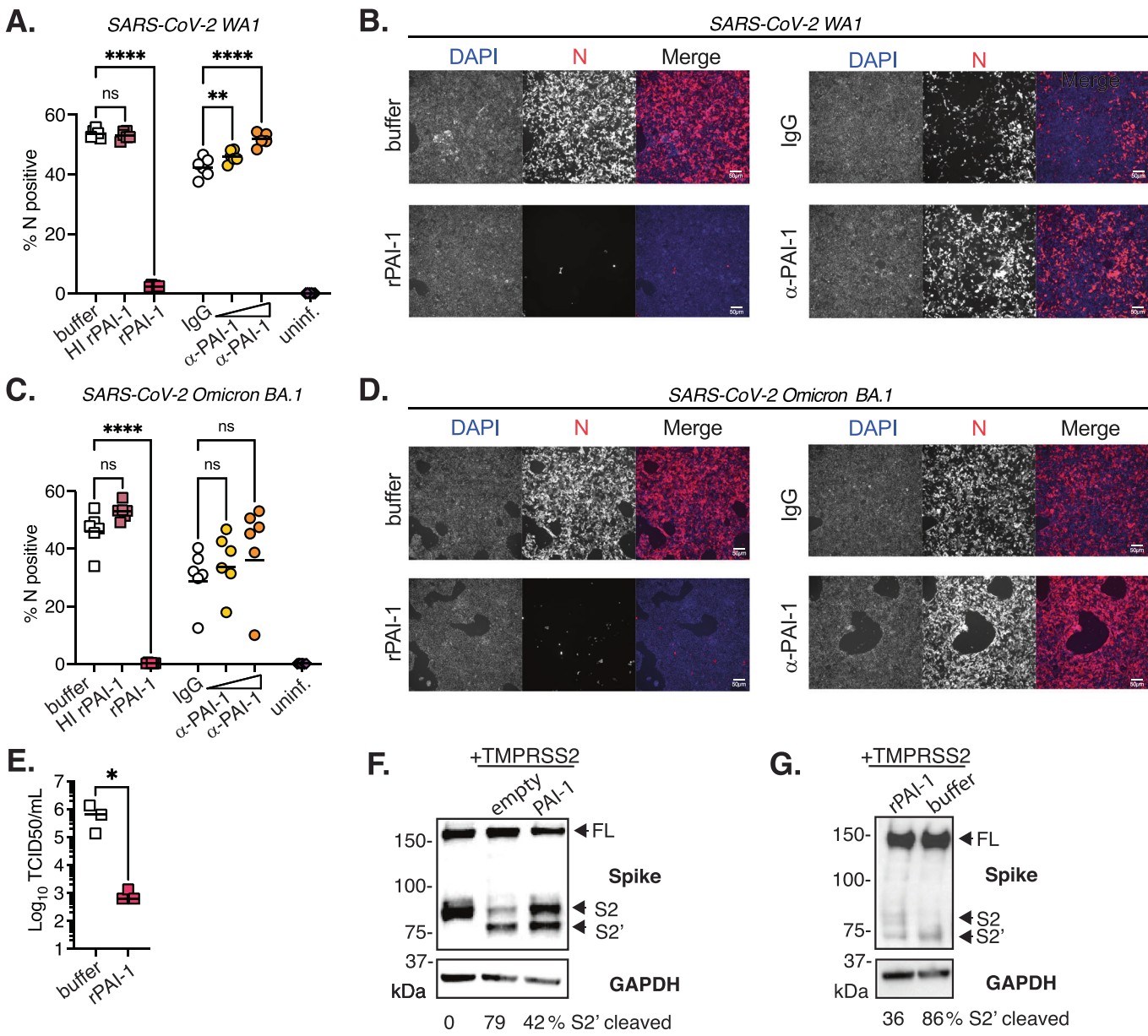

**Figure 5. Effect of PAI-1 on SARS-CoV-2 multi-cycle replication and spike maturation.**

(A–D) SARS-CoV-2 WA-1 (**A**, **B**) or Omicron BA.1 (**C**, **D**) multi-cycle infection in Calu-3 cells with extracellular addition of buffer, 12.5 or 25 ng/mL of active PAI-1 or of 25 ng/mL heat-inactivated (HI) PAI-1 (48 hpi); or 10 or 100 ng/mL anti-PAI-1 depleting antibody or 10 ng/mL isotype IgG control (48 hpi) by high-content microscopy. Statistical analysis by One-way ANOVA and Holm-Sidak's multiple comparison test, **$P < 0.005$. Exact P values in (**A**). buffer vs. HI rPAI-1 $P = 0.9407$; buffer vs. rPAI-1 $P < 0.0001$; anti-PAI-1 (1:100) vs. IgG control $P = 0.0059$; anti-PAI-1 (1:50) vs. IgG control $P < 0.0001$. Exact P values in (**C**). buffer vs. HI rPAI-1 $P = 0.5441$; buffer vs. rPAI-1 $P < 0.0001$; IgG vs. anti-PAI-1 (1:100) $P = 0.7968$; IgG vs. anti-PAI-1 (1:50) $P = 0.1473$. $n = 6$ biological replicates. Representative images, DAPI (blue, nuclei), SARS-CoV-2 nucleoprotein (red). (**E**) Extracellular titers of SARS-CoV-2 WA-1 grown at low MOI-infection on Calu-3 cells for 48 h. Infectious titers were determined by $TCID_{50}$ assay on Calu-3. Statistical analysis by one-sided t test, *$P < 0.05$. Exact P value $P = 0.05$, $n = 3$ biological replicates (**F**). BHK cells co-transfected to express SARS-CoV-2 spike, TMPRSS2, and PAI-1. Spike S2 and 2′ band intensities by western blot and densitometry. % S2′ cleaved determined by ratio of S2′ vs S2′ + S2 band intensity from the blot shown. FL full-length. (**G**) BHK cells co-transfected to express SARS-CoV-2 spike and TMPRSS2, and rPAI-1 or buffer control added to the cell supernatant. Spike S2 and 2′ band intensities by western blot and densitometry. S2′ % cleaved determined by the ratio of S2 vs S2′ band intensity from the blot shown. FL full-length. Source data are available online for this figure.

AVP is a 23 kDa cysteine proteinase, produced in the late stages of the AdV life cycle. It is packaged and activated inside nascent virus particles, where it cleaves viral precursor proteins. Of the 12 major AdV virion proteins, 7 are precursor proteins cleaved by AVP (Mangel and San Martin, 2014). Studies using an AdV mutant, *ts1*, incapable of packaging AVP, have shown that cleavage of AdV precursor proteins is essential for viral infectivity (Greber et al, 1996; Rancourt et al, 1995). AVP exists in three distinct forms

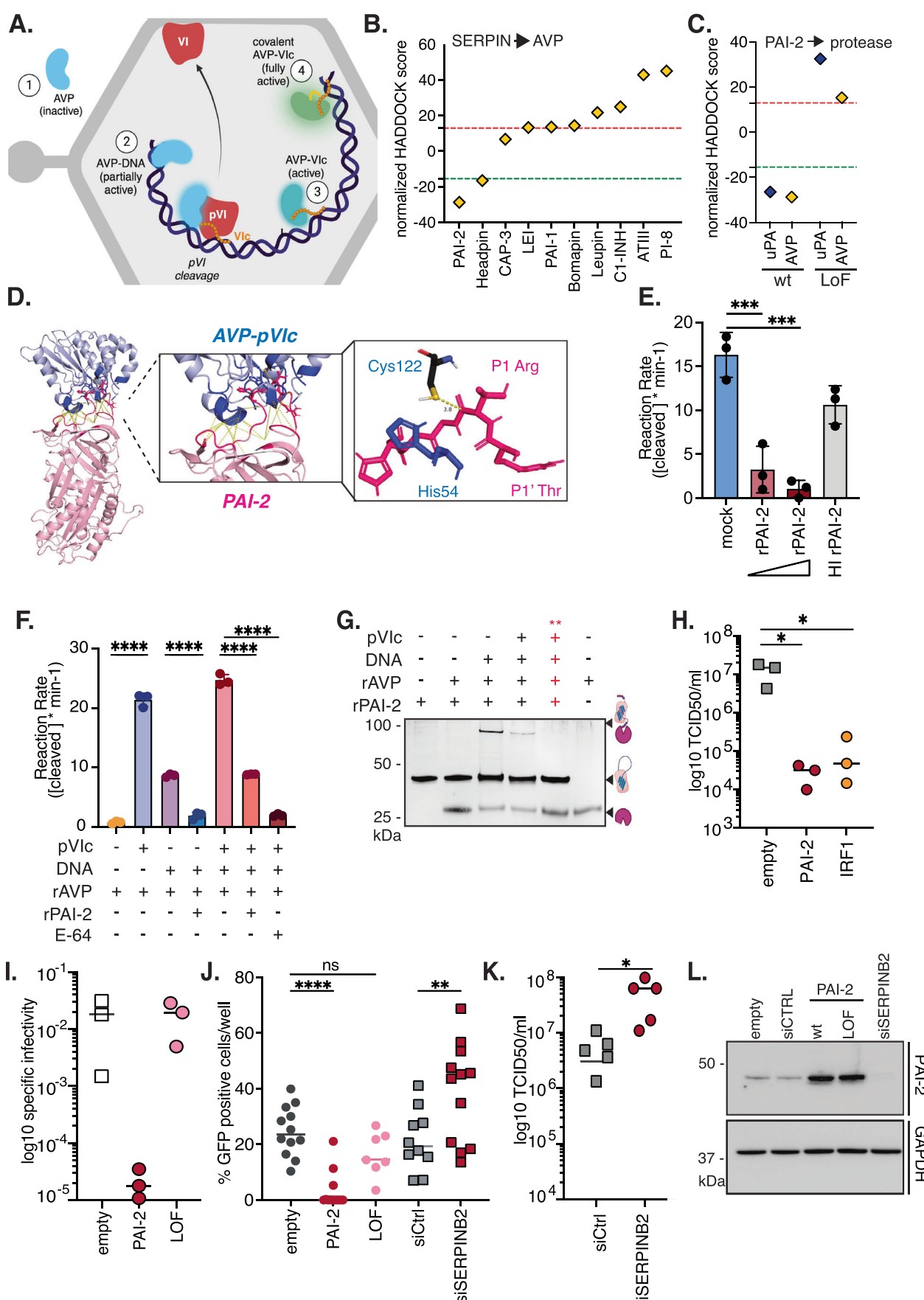

**Figure 6.  Characterization of PAI-2 and adenovirus protease (AVP).**

(A) AVP exists in different forms with varying catalytic activities. ① AVP itself is inactive. ② AVP becomes partially activated upon binding to the viral DNA in the immature AdV particle. ③ The precursor to protein VI, pVI, slides along the viral DNA into immobile AVP. AVP cleaves the 11-amino acid peptide pVIc from the C-terminus of pVI; the pVIc peptide binds covalently to AVP. The AVP-pVIc complex then slides along the DNA cleaving the viral precursor proteins also bound to the DNA. AVP-pVIc complexes not bound to DNA are active in vitro but do not exist in vivo. Relative enzyme activity ($k_{cat}/K_m$): AVP- 1, AVP-DNA- 110, AVP-pVIc- 1130 and AVP-pVIc-DNA- 34,100. (B) Normalized HADDOCK scores of AVP (conformation upon pVIc binding) with 10 SERPINs expressed during respiratory viral infections. Dotted line (mean of all positive controls in the in-silico screen) is the threshold for high-confidence binders. (C) Normalized HADDOCK scores of PAI-2 wild-type or loss-of-function (LOF, P4-P4′ alanine substitutions) with positive control uPA and AVP (conformation upon pVIc binding). Dotted line (mean of all positive controls in the in-silico screen) is the threshold for high-confidence binders. Blue, serine proteases; yellow, cysteine proteases. (D) Docking structure of top-scoring complex for PAI-2 with AVP (conformation upon pVIc binding). (E) Normalized reaction rates from in vitro fluorometric assays for sonicated AdV particles derived-AVP activity in the presence of rPAI-2, heat-inactivated PAI-2 control, or without the addition of rPAI-2 (mock). Statistical analysis by one-way ANOVA and Kruskal–Wallis test, *P < 0.05. Mean +/− SD, n = 3 replicates. (F) Normalized reaction rates from in vitro fluorometric assays for AVP activity in the presence of rAVP, pVIc, random DNA hexamers, PAI-2 or E-64 as an unspecific cysteine reducing agent. Statistical analysis by One-way ANOVA and Kruskal–Wallis test, *P < 0.05. Mean +/− SD, n = 3 replicates. (G) Silver-stained SDS-PAGE of 30 min reactions at 37 °C with recombinant AVP, pVIc, random DNA hexamers, and PAI-2; HI, heat-inactivated at 98 °C for 10 min. (H) AdV5-GFP TCID50 titers of A549 cells transduced with lentiviral constructs expressing either empty, PAI-2 or IRF-1 at 96 h.p.i. Statistical analysis by one-way ANOVA and Kruskal–Wallis test, *P < 0.05. Mean +/− SD, n = 3 biological replicates. (I) Specific infectivity of A549 cells infected with HAdV5-GFP transduced with either PAI-2, PAI-2 Loss of Function mutant (LoF), or empty vector control. Supernatants were titrated on A549 cells, calculated from genome copy numbers and titers from Fig. EV5D,E,J. GFP-positive HEK-293T cells per well after transfection with either empty, PAI-2 or LOF expression constructs, or control siRNA, SERPINB2 siRNA, followed by 72 h AdV5-GFP infection. % infected statistical analysis by ANOVA, *P < 0.05; ns, non-significant. Exact P values empty vs. PAI-2 P < 0.0001; empty vs. LOF P = 0.3789; siCtrl vs. siSERPINB2 P = 0.0017, n = 6. (K) Titration of supernatants in A549 cells derived from data shown in (J) at *P < 0.05. Exact P value P = 0.0004. (L) PAI-2 and GAPDH western blot of HEK-293T cells transfected with either empty, PAI-2, LoF, control siRNA or SERPINB2 siRNA at 48 h post-transfection. Source data are available online for this figure.

during the AdV life cycle, each displaying different catalytic activities and therefore, presumably, different structures (Fig. 6A): AVP alone, AVP bound randomly to DNA (AVP-DNA), and AVP covalently bound to the 11-amino acid peptide pVIc, which is in turn bound randomly to DNA (AVP-pVIc-DNA). The third form, which is the fully active AVP complex, slides by one-dimensional diffusion on AdV DNA to encounter and cleave DNA-bound AdV precursor proteins. This allows AVP to efficiently operate within the confines of the DNA mesh within the AdV particle (Baniecki et al, 2013; Mangel and San Martin, 2014; McGrath et al, 2001).

To determine whether a human SERPIN can inhibit AVP, we performed in-silico docking with AVP and each of the 10 SERPINs upregulated during viral infection (Figs. 1 and 2). We identified PAI-2 (*SERPINB2*) as a high-confidence binder of AVP (Fig. 6B). The HADDOCK score for wild-type PAI-2 with AVP was similar or lower to the scores of known PAI-2 targets, such as uPA; the introduction of a PAI-2 P1 alanine substitution rendered HADDOCK scores unfavorable (Fig. 6C). We note that in our host protease screen PAI-2 also scored well with host cathepsins (Fig. 3B), which are cysteine proteases, like AVP.

We visually assessed the interaction between PAI-2 and an active form AVP, the AVP-pVIc complex (Ding et al, 1996) (McGrath et al, 2003) and found that the PAI-2 RCL inserts deeply into the active site of AVP (Fig. 6D). AVP's nucleophilic C122 is positioned perfectly for attack on R380, suggesting possible cleavage at 380 R↓T381 by AVP, which is also where PAI-2's canonical target uPA cleaves. Notably, the fit is unfavorable for the inactive form of AVP (Baniecki et al, 2013) (Fig. EV5A). In addition to the RCL fit into the AVP-pVIc active site, we identified potential secondary binding sites (exosites) outside the PAI-2 RCL with AVP-pVIc (Fig. 6D), known from other SERPINs to stabilize SERPIN-protease interactions and to modulate SERPIN specificity (Gettins and Olson, 2009).

To test for inhibition of AVP proteolytic activity by PAI-2, we next conducted fluorometric protease activity assays, using (L-R-G-G-NH)2-Rhodamine as a synthetic substrate; cleavage at the glycine residue next to the rhodamine moiety causes a massive

increase in fluorescence (Mangel et al, 1993; McGrath et al, 1996). We first used AVP from sonicated AdV particles – in these particles, AVP is bound to both pIVc and DNA and is thus fully active. We observed a dose-dependent, significant reduction in AVP reaction rates in the presence of rPAI-2, showing that PAI-2 inhibits AVP activity (Fig. 6E). We then repeated this assay using active AVP reconstituted in vitro by combining recombinant AVP (rAVP), pVIc, and random DNA hexamers. We used E-64, a small molecule that non-specifically reacts with cysteines, as a control (Matsumoto et al, 1999). Consistent with previous reports (Mangel et al, 1993; McGrath et al, 1996), we found that AVP reaction rates were low with rAVP alone, but the addition of its co-factors DNA and pVIc significantly increased the reaction rate. The highest reaction rate was observed when both co-factors were present. Notably, we found that PAI-2 significantly inhibited both active AVP complexes, rAVP-DNA and rAVP-DNA-pVIc activity (Fig. 6F).

We next asked if PAI-2 inhibits AVP through the canonical covalent "mousetrap" mechanism, a process that requires active AVP. We thus used in vitro reconstituted active AVP complexes and incubated them with rPAI-2. SDS-PAGE and silver staining showed PAI-2 to be present in two bands, a 47 kDa band (corresponding to the PAI-2 monomer) and a 74 kDa band (corresponding to the predicted molecular weight of a PAI-2:AVP complex, Fig. 6G). We obtained the same result with active AVP from sonicated AdV particles (Fig. EV5B). The presence of this high-molecular-weight covalent complex was indicative of the SERPIN "mousetrap" mechanism.

Next, we evaluated the anti-AdV potential of PAI-2, which has not been shown for any AVP inhibitor before. We transduced A549 cells to express either PAI-2 or IRF-1, a validated anti-adenovirus antiviral effector control that is interferon-stimulated. The transduced cells were then infected with human Adenovirus, species C, serotype 5 (HAdV-C5), carrying a GFP reporter (AdV-GFP (Chikhalya et al, 2021)). At 72 h post-infection (hpi), we collected both intracellular and extracellular progeny virus particles

by freeze-thawing the cells in their infection media. We determined the progeny viral infectious titers using TCID$_{50}$ in A549 cells by scoring GFP-positive wells. Our results showed a significant reduction, greater than 100-fold, in infectious viral titer in the presence of PAI-2, on par with the IRF-1 positive control (Fig. 6H).

Experiments with a temperature-sensitive AVP mutant, *ts1*, which fails to incorporate AVP into progeny particles, demonstrated that the absence of AVP reduces the specific infectivity of progeny viral particles (Greber et al, 1996; Rancourt et al, 1995). Inhibition of AVP by PAI-2 is expected to replicate this effect. To determine specific infectivity, we assessed infectious titers of wild-type (wt) PAI-2 expressing cells, or a loss-of-function mutant (LoF) with an alanine substitution at the critical R at P1 of the RCL (PAI-2 R380A) expressing cells. We collected supernatants for titration via TCID$_{50}$ and quantified AdV viral genomes by qPCR. The ratio of infectious particles to genome copy numbers indicates specific infectivity. Our results showed that the PAI-2-mediated effect leads to a reduction in particle infectivity rather than a decrease in the total particle number, consistent with a defect in AVP-dependent maturation (Figs. 6I, and EV5). Previous research with the *ts1* mutant also demonstrated that reduced infectivity in the absence of AVP primarily results from the failure to process pVI, which is required for AdV release from the endosome, causing entrapment of immature virions within endosomes during entry (Greber et al, 1996; Rancourt et al, 1995).

AdV-based vectors are currently used in 20% of ongoing human gene therapy trials (Sallard et al, 2023). Generation of high-titer infectious AdV ensures the efficient transduction of target cells, which is crucial for therapeutic success. The action of viral restriction factors, such as PAI-2, in producer cells could potentially pose challenges during AdV manufacturing. Therefore, we evaluated whether overexpression of the AVP restriction factor PAI-2 reduces AdV infection and whether its knockdown enhances AdV infection in a cell line commonly used to produce AdV, HEK293T. We transfected HEK293T cells with expression constructs for PAI-2 wild-type (wt) or loss-of-function (LoF) or with siRNA non-targeting control or siSERPINB2 knockdown constructs for 48 h. We then infected the cells with HAdV-5C-GFP and, 72 h post-infection, determined the percentage of GFP-positive cells per well by high-content microscopy. We observed a significant decrease in the percentage of infected cells (GFP-positive, Fig. 6J, or stained for capsid, Fig. EV5f), with PAI-2 wt compared to LoF, and a significant increase of virus-positive cells with the siSERPINB2 construct compared to siCTRL. Titration of infectious virus from the supernatants resulted in a significant increase for siSERPINB2-transfected cells compared to controls (Fig. 6K). Silencing of PAI-2 expression was confirmed via western blot (Fig. 6L). This suggests that decreasing the amount of PAI-2 during stock production may increase AdV stock titers, which is highly relevant for manufacturing processes underlying AdV-mediated gene therapy.

Together, our findings establish PAI-2 as a potent host inhibitor of AdV through inhibition of its protease, and discover a direct-acting antiviral SERPIN.

## Discussion

SERPINs have been extensively investigated in various contexts of human disease, but our understanding of SERPIN expression has

mainly been limited to steady-state conditions. Recent sequencing efforts have uncovered the complexity of tissue-specific SERPIN expression (Law et al, 2006; Silverman et al, 2004), and emerging evidence indicates that SERPINs are induced in response to inflammatory cytokines (Dong et al, 2002; He et al, 2010; Lappin and Whaley, 1989; Lee and Ashkar, 2018; Rosendal et al, 2022). Consistent with these findings, our study demonstrates that SERPIN mRNA expression is elevated in response to viral infection in both individuals and in human airway epithelial (HAE) cultures. Notably, although key SERPINs were found to be elevated in both, we observed some differences between the two systems. These differences stem from the interplay between respiratory epithelial cells and immune cells, which are lacking in the HAE culture model. For example, it has been demonstrated that macrophage-derived proinflammatory cytokines can stimulate SERPIN expression (Schroder et al, 2019). Furthermore, there could be a bias in the epithelial cells that dissociate preferentially during BAL collection, which may be cells undergoing apoptosis or necroptosis. Future studies using HAE cultures with exogenously provided cytokines or co-culture systems incorporating immune cells will provide valuable experimental insights into SERPIN transcriptional regulation. Our findings further revealed diverse mRNA expression patterns of specific SERPINs during viral infection, with some SERPINs showing patterns similar to interferon-regulated genes, while others exhibited patterns that were distinct from interferon-stimulated genes and appeared virus-specific. This suggests the involvement of virus-specific regulatory mechanisms beyond interferon signaling, which will be part of future investigations.

Our knowledge of the full array of proteases targeted by the SERPIN family is limited, and thus, the comprehensive understanding of their regulatory potential remains incomplete. Historically, SERPIN target discovery has progressed incrementally, focusing on individual SERPINs or proteases rather than employing systematic discovery approaches. For example, tPA was identified as a PAI-1 target through SDS-PAGE with zymography in plasma samples, searching for SERPIN-protease high-molecular weight complexes (Booth et al, 1987). More systematic approaches include co-immunoprecipitation of SERPIN-protease complexes and subsequent partner identification by precipitating either the protease (Liu et al, 2023) or the SERPIN (Bhatia et al, 2011). Another strategy is the co-expression of SERPIN variants and select proteases in bacteria, followed by assaying residual protease activity to identify inhibitory variants (Polderdijk et al, 2017). In addition, a recent study generated a RCL mutant library on a known SERPIN background, α1-antitrypsin, tested their inhibitory potential against multiple proteases, and based on the obtained dataset, developed computational algorithms that predict binding denominators within the RCL (de Maat et al, 2019). This enabled the prediction of how RCLs contribute to target specificity in a comprehensive way. However, information on the 3D structure of both partners, i.e., of the protease's specificity pocket, and of the formation of exosites between residues outside of the RCL (Fig. EV1A) was not considered, and the algorithm relies on data from synthetic SERPINs rather than those occurring in nature. The use of in-silico protein-protein docking, which has demonstrated success in identifying compounds that modulate enzymatic reactions, including those involving proteases, could overcome these limitations. Yet, in-silico docking is constrained by the need for detailed knowledge on the binding directionality of the two partners, such

as guiding the docking of a small-molecule inhibitor into the catalytic center of a protease, where it is functionally relevant. Fortuitously, the well-understood and conserved nature of SERPIN-protease interactions has enabled us to leverage in-silico protein-protein docking for SERPIN target discovery in our study, by guiding the SERPIN RCL into a protease's catalytic center. Our approach provides a quantitative, numeric score for predicted SERPIN-protease interaction energies and, additionally, offers a structural assessment of bound complexes, thereby ensuring appropriate fit, visualization of potential exosites, and approximate prediction of P1 residues. Importantly, it transcends the reliance on protease motif similarities alone, as exemplified by our correct identification of CTSL but not CTSB as a PAI-1 target, despite those two CTS sharing the same minimal substrate motif. Another example is the case of TMPRSS2 – the molecule undergoes autocleavage to convert the inactive zymogen to the active enzyme. Our in-silico modeling suggests that PAI-1 only binds the active enzyme and not the zymogen, which was previously unknown, and which has important implications for designing recombinant SERPINs as therapeutics. Together, these findings show that while the protease recognition sequence is important, extrinsic factors that allow for proper 3D fit, such as zymogen cleavage, the presence of exosite interactions, and the composition of the surrounding environment (pH, co-factors), are additional key determinants for SERPIN protease selectivity. In all, our approach extends beyond the identification of SERPIN specificity patterns and allows for molecular predictions of how SERPINs bind to noncanonical proteases.

One limitation of our screening approach is that it relies on the accurate knowledge of both SERPIN and protease 3D structures. False predictions could occur when the input 3D structures do not account for physical and biochemical factors that influence their shape. For example, SERPINC1 requires heparin as a cofactor to mediate its interaction with thrombin, its canonical target, in a bridge-like manner (Law et al, 2006). To overcome this limitation, the crystal structure of the SERPINC1-heparin-thrombin complex can serve as a template for modeling (Jin et al, 1997). We used the same approach to model binding of active AVP (AVP-pVIc), whose active crystal structure had been solved previously (Ding et al, 1996), to PAI-2. Another limitation is that our docking methodology does not account for the subcellular localization of SERPINs and their targets, e.g., within specific compartments or the extracellular space. For instance, PAI-1, a secreted SERPIN, exhibits anti-protease activity in the extracellular space (Humphreys et al, 2023). The TMPRSS2 zymogen is membrane-bound initially and inactive, and its active form is released into the extracellular space upon autocleavage (Fraser et al, 2022). Thus, it is feasible that PAI-1 and the TMPRSS2 active form come into contact and interact in the extracellular space. Indeed, our previous research has demonstrated that the exit of PAI-1 from cells is crucial for providing antiviral protection against influenza virus spread (Dittmann et al, 2015). Although CTSL is typically found within endosomes (Anes et al, 2022), it can also be secreted (Dykes et al, 2019; Yadati et al, 2020), and the receptor-mediated uptake of extracellular PAI-1 via endocytosis and the described anti-protease activity of PAI-1 within endosomes have been shown (Cao et al, 2006; Underhill et al, 1992). This implies the potential for PAI-1 to interact with CTSL either extracellularly or within endosomes. Our

in vitro findings indicate PAI-1-CTSL complex formation at a pH resembling that of early endosomes, and we show inhibition of the SARS-CoV-2 variant Omicron BA.1, which relies on CTSL for maturation, by PAI-1.

We recognize that some previously published SERPIN:protease pairs did not meet our stringent HADDOCK score threshold for predicted high-confidence binders (Fig. 4B,C). An initial step in identifying true binders with scores in-between thresholds can be visual assessment of docked complexes to 1. evaluate the fit of the reactive center loop (RCL), 2. identify the presence or absence of exosites, and 3. determine the positioning of known SERPIN P4-P4' residues vis-à-vis the protease nucleophile. As a next step, introducing RCL mutations in-silico to reduce binding, as demonstrated with uPA, can be informative; for a true binder, these mutations should increase the HADDOCK score to levels similar to known negative controls (e.g., furin in the case of uPA). Furthermore, ranking HADDOCK scores in the context of closely related protease family members can be beneficial, as shown with cathepsins and PAI-1 (Fig. 4C). This analysis revealed that while the CTSB:PAI-1 HADDOCK score is only slightly higher than that of uPA:PAI-1, it is significantly higher than the scores of PAI-1 with most other cathepsin family members, suggesting that CTSB is likely a non-binder. Ultimately, as with any screening method, experimental validation is essential to confirm HADDOCK predictions and further explore binding modalities.

Our study marks a systematic exploration of SERPIN targets within a specific proteolytic context, by docking 10 SERPINs with a panel of 48 host proteases expressed in the respiratory epithelium. All tested SERPINs exhibited predicted additional protease binding partners to those previously described. Clustering of the data containing 480 modeled interactions unveiled previously unnoticed patterns in SERPIN selectivity as well as in protease sensitivity to SERPIN-mediated inhibition. For example, we observed that most members of the TMPRSS family were likely not targeted by any of the SERPINs tested, except for TMPRSS13 and 14, predicted for SERPING1 only; TMPRSS1, predicted for SERPINB1 only; and TMPRSS11D, TMPRSS4, and TMPRSS2, predicted to interact with multiple SERPINs. Conversely, certain host proteases such as trypsin, plasma kallikrein, and CTSA were predicted to bind nearly all tested SERPINs, indicating potential sensitivity to inhibition by multiple SERPINs. SERPINB1 was predicted as the most protease promiscuous SERPIN in our panel, whereas SERPINC1 was predicted to be the most selective. The vast majority of these predicted interactions (or predicted non-interactions) were previously unknown (Fig. EV3A). Another finding was cross-class inhibition by SERPINs that were previously thought to be protease class-specific. Such cross-class SERPIN interactions have been documented previously, particularly for viral-encoded SERPINs such as Serp-2 and CrmA, which bind to both cysteine and serine proteases(Viswanathan et al, 2012). This phenomenon is also observed in mammalian SCCA serpins (Masumoto et al, 2003). However, cross-class interactions for the well-studied SERPIN PAI-1 had not been previously shown. This discovery may have major implications due to PAI-1's role in many pathologies, including cancer, cardiovascular disease, inflammation, fibrosis, diseases of the central nervous system, and aging, all of which also rely on the critical action of numerous proteases (Sillen and Declerck, 2021).

Our findings highlight numerous potential candidates for antiviral SERPINs warranting further mechanistic exploration beyond this study. Given our interest in virology, we focused a part of the present study on PAI-1 targeting active TMPRSS2 and members of the CTS family, the latter challenging the prevailing understanding of this extensively studied SERPIN's specificity. CTS play pivotal roles in various viral diseases, regulating viral antigen presentation, facilitating viral release through extracellular matrix remodeling, modulating cell death, and contributing to viral glycoprotein maturation in viruses such as Ebola virus, Nipah virus, Hendra virus, and SARS-CoV-2 (Scarcella et al, 2022). Notably, both CTSB and CTSL are the main CTS known to provide viral glycoprotein maturation. The observation that PAI-1 exclusively inhibits CTSL and not CTSB suggests that another SERPIN with predicted specificity for CTSB (i.e., SERPINB2, encoding PAI-2) may be needed to bridge this gap. The discovery of PAI-1 targeting CTSL, previously unknown, along with our demonstration of PAI-1's direct binding of active TMPRSS2 in this study, implicates the inhibition of both primary (TMPRSS2) and auxiliary (CTSL) proteases in SARS-CoV-2 maturation. Our findings indicate a functional link between extracellular PAI-1 levels and the control of SARS-CoV-2 spread, potentially explaining the underlying mechanism behind other reports linking the presence of the PAI-1 single-nucleotide polymorphism rs6092 to increased disease severity in COVID-19 (Fricke-Galindo et al, 2022). Moreover, the discovery of PAI-1 targeting CTSL may have implications beyond infectious diseases, as this pair has been found to be upregulated concurrently in the metastasis of various cancers (Farinati et al, 1996; Harbeck et al, 2000; Herszenyi et al, 2008), suggesting a functional rather than merely correlative aspect of disease outcome.

Viral glycoproteins have evolved to exploit multiple host proteases redundantly for maturation, as seen in SARS-CoV-2 Spike's processing by TMPRSS2 and CTSL. Thus, targeting host proteases like TMPRSS2 with Camostat individually has not been fully effective in antiviral therapy (Bestle et al, 2020; Bottcher-Friebertshauser et al, 2010), (Gunst et al, 2021). Similarly, innate host defenses must block a combination of proteases to effectively inhibit viral glycoprotein maturation. Our screen predicts that most SERPINs inhibit multiple proteases. LEI (SERPINB1), a member of the B clade, demonstrated extensive predicted protease binding, encompassing various CTSs, including some previously identified (Law et al, 2006; Avril et al, 1994), as well as trypsin and furin. Another broad-spectrum candidate is Headpin (SERPINB13), which showed favorable scores throughout the tested CTS, in accordance with its known role as a CTS inhibitor (Jayakumar et al, 2003), but also scored favorably with furin, which was previously unknown. It must be noted that none of the tested SERPINs demonstrated comprehensive inhibitory activity against all known pro-viral host protease classes, suggesting a need for a combination of SERPINs to effectively inhibit viral replication by targeting complementary proteolytic pathways in the human airway epithelium.

Our study marks the discovery of a host SERPIN directly targeting a viral protease: PAI-2 (encoded by SERPINB2) inhibits Adenovirus protease (AVP). We propose a model where PAI-2 forms a covalent "mousetrap" bond with active AVP, irreversibly inhibiting its activity and reducing the specific infectivity of progeny AdV particles. Where do PAI-2 and AVP meet? Endocytosis of extracellular PAI-2 through the endosomal pathway has been previously shown (Croucher et al, 2006). Thus, it is theoretically feasible that extracellular PAI-2 enters cells alongside AdV particles and blocks endosomal escape or other stages of the replication cycle requiring AVP activity before the onset of viral replication. However, we consider this process unlikely, as PAI-2 would need to enter AdV particles to access AVP. A more plausible model is that intracellular PAI-2 binds and inhibits AVP during the late stage of infection. Within the cell, PAI-2 is present in both the cytoplasm and the nucleus, where AVP is also located, allowing for post-translational interaction. In the cytoplasm, AVP remains inactive due to the absence of its co-factors. It is possible that both partners bind in the cytoplasm, reversibly, before AVP reaches full activity, perhaps by inhibiting AVP cofactor binding through steric hindrance or by hindering nuclear translocation of AVP. Our data with reconstituted active AVP complex indicate that the irreversible mousetrap mechanism requires partially or fully active AVP, which requires AVP nuclear localization. It is possible that in the nucleus, irreversibly bound PAI-2 hinders AVP packaging into progeny particles or that the irreversible PAI-2:AVP-complex is packaged, with PAI-2 inhibiting AVP activity within the virion. Determining which of these scenarios occurs in nature will be the focus of future studies. Given the clinical relevance of human AdV infections, particularly in the elderly and immunocompromised, the lack of effective anti-AdV treatments, the critical importance of AVP within the AdV lifecycle, and the high conservation of AVP among all human AdV types, our findings are highly relevant for the development of anti-AdV therapeutics. In addition, we show that reducing the levels of PAI-2 in an AdV stock producer cell line significantly boosts viral replication, which could have significant implications for AdV gene therapy manufacturing.

The investigation of host and viral proteases as targets in antiviral treatments has been a key focus in drug development (Rahbar Saadat et al, 2021). Our foundational study reveals the antiviral potential of host-encoded protease inhibitors and suggests key proteolytic pathways that can be targeted for antiviral therapies by mirroring the action of antiviral SERPINs. Another possibility is the expression or administration of full-length recombinant SERPINs with redirected specificity (Maas and de Maat, 2021) or short SERPIN RCL peptides (Ambadapadi et al, 2016). An alternative option is the design of small molecules stabilizing SERPIN-protease binding (Gettins and Olson, 2016). These approaches aim to replicate the SERPIN mechanism while mitigating potential side effects associated with full-length wild-type SERPINs, which retain their impact on unwanted host proteases and can thus lead to adverse effects when overexpressed. Beyond SARS-CoV-2, similar approaches as in this study could be undertaken to discover antiviral SERPIN candidates relevant for proteolytic landscapes in other viral entry portals, such as the gut, and for emerging viruses with predicted protease reliance based on their glycoprotein cleavage sites.

Overall, our study broadens the understanding of SERPIN-protease interactions within the specific proteolytic context of the respiratory epithelium and discovers a direct-acting antiviral SERPIN, with significant implications for infectious disease research and beyond.

# Methods

### Reagents and tools table

| Reagent/resource | Reference or source | Identifier or catalog number |
|---|---|---|
| **Experimental models** | | |
| Bci-NS1.1 cells | Walters and Crystal labs, Cornell University | Cell Line #63-66 |
| A549 cells | ATCC | CCL-185 |
| Calu-3 cells | ATCC | HTB-55 |
| BHK-21 cells | ATCC | CCL-10 |
| SARS-CoV-2 BA.1 | Mehul Suthar Laboratory, Emory University | hCoV-19/USA/GA-EHC-2811C/2021 |
| HAdV5-GFP | Hearing Lab, Stony Brook University | Virus 25 |
| Influenza A/California/07/2009 (H1N1) | BEI Resources | NR-13663 |
| HPIV3-GFP | Gift from Dr. Peter Collins | N/A |
| Rhinovirus A16 | Gift from Dr. Ann Palmenberg | N/A |
| SARS-CoV-2 USA-WA1/2020 | BEI Resources | NR-52281 |
| Reovirus Type 3 Dearing (T3 SA) | Gift from Dr. Terence Dermody | N/A |
| **Recombinant DNA** | | |
| pSCRPSY SERPINE1 (PAI-1) | Dittmann Box 2/ Bieniasz lab | In-house #159 |
| pcDNA3.1 SARS-CoV-2 Spike | Sino Biological | VG40589 |
| pCAGGS TMPRSS2 | Dittmann Box 3/ Steinhauer lab | In-house #198 |
| pSCRPSY empty-RFP | Dittmann Box 2 | In-house #64 |
| **Antibodies** | | |
| Anti-SARS-CoV-2 Spike S2 (1A9), mouse monoclonal | ThermoFisher | MA5-35946 |
| Anti-SARS-CoV-1/2 Nucleocapsid, clone 1C7C7 | Cell Signaling Technology | 68344 |
| Anti-c-Myc, mouse monoclonal | ThermoFisher | MA1-980 |
| Anti-PAI-2 antibody | ThermoFisher | PA5-27857 |
| Anti-PAI-1 antibody | Proteintech | 13801-1-AP |
| Mouse IgG1 isotype control | Thermo Fisher Scientific | MA1-10406 |
| Goat anti-mouse IgG (HRP) | Invitrogen | G21040 |
| Goat anti-rabbit IgG (HRP) | Invitrogen | G21234 |
| Donkey anti-mouse IgG (Alexa Fluor 647) | Jackson ImmunoResearch | 715-605-151 |
| goat anti-Adenovirus | Antibodies-Online | ABIN236692 |
| Anti-goat IgG Alexa Fluor 647 | Jackson ImmunoResearch | 715-605-151 |
| GAPDH Mouse monoclonal, HRP-conjugated | Thermo Fisher Scientific | MA5-15738-HRP |

| Reagent/resource | Reference or source | Identifier or catalog number |
|---|---|---|
| **Oligonucleotides and other sequence-based reagents** | | |
| SERPIN qPCR Primers | This study | Table EV2 |
| Silencer Select siRNA SERPINB2 | Thermo Fisher Scientific | S10018 |
| Silencer Select Negative Control No. 1 siRNA | Thermo Fisher Scientific | 4390844 |
| random 36-mer DNA (AVP cofactor) | IDT (Integrated Device Technology) | ND |
| **Chemicals, enzymes, and other reagents** | | |
| Recombinant active human PAI-1 (N150K, K154T, Q319L, M354I) | Abcam | Not specified |
| Recombinant PAI-2 | This study | Purified recombinant protein |
| Recombinant TMPRSS2 | Biomatik | RPC24995 |
| Recombinant Cathepsin L | R&D Systems | 952-CY-010 |
| Recombinant Cathepsin B | R&D Systems | 953-CY-010 |
| Recombinant urokinase plasminogen activator (uPA) | Abcam | Not specified |
| PneumaCult™-Ex Plus Medium | STEMCELL Technologies | #05040 |
| PneumaCult™-ALI Medium | STEMCELL Technologies | #05001 |
| Triplaxtinin (PAI-039) | Axon Medchem | Not specified |
| DTT | Sigma-Aldrich | D9779 |
| EDTA | ThermoFisher | AM9260G |
| MES buffer | Sigma-Aldrich | M3671 |
| TRIS-HCl | ThermoFisher | BP152-1 |
| NaCl | Sigma-Aldrich | S3014 |
| Lipofectamine LTX + PLUS Reagent | Thermo Fisher Scientific | 15338100 |
| Rat-tail Collagen Type I | Corning | #354236 |
| 6.5 mm Transwell Inserts with 0.4 µm Pore Polyester Membranes | Corning | #3470 |
| Phosphate-buffered saline (PBS) | Various suppliers | Standard lab grade |
| Lipofectamine RNAiMAX Transfection Reagent | Thermo Fisher Scientific | 13778075 |
| Rhodamine-labeled peptide substrate | Biosynth | custom peptide order |
| E-64 | Sigma-Aldrich | E3132 |
| pVIc peptide (custom synthesis) | Thermo Fisher Scientific | PEP95UNMOD |
| Sonicated Adenovirus 5 particles | Internal preparation | N/A |
| NuPAGETM NOVEXTM 4-12% Bis-Tris | ThermoFisher | NP0323BOX |
| 0.25% Trypsin-EDTA (without Ca²⁺/Mg²⁺, HBSS) | Thermo Fisher Scientific | MT25053CI |
| DMEM 1X | Corning | 10-013-CV |

| Reagent/resource | Reference or source | Identifier or catalog number |
|---|---|---|
| iBlot™ PVDF Transfer Stacks | Thermo Fisher Scientific | IB24001 |
| NuPAGE™ LDS Sample Buffer (4X) | Fisher Scientific | B0007 |
| NuPAGE™ Sample Reducing Agent (10X) | Fisher Scientific | B0009 |
| NuPAGE™ Bis-Tris Gel, 4–12%, 10-well | Thermo Fisher Scientific | NP0322BOX |
| **Software** | | |
| ImageJ | NIH | https://imagej.net/ij/ |
| PyMOL v2.0 | Schrödinger | https://pymol.org |
| HADDOCK v2.4 | Bonvin Lab, Utrecht University | https://wenmr.science.uu.nl/haddock2.4 |
| Swiss-Model | SIB Bioinformatics Resource Center | https://swissmodel.expasy.org |
| GraphPad Prism v9 | GraphPad Software | https://www.graphpad.com |
| ComplexHeatmap (R package) | Bioconductor | https://bioconductor.org/packages/ComplexHeatmap |
| R v4.1.2 | R Project | https://www.r-project.org |
| Cellomics CX7 Software | Thermo Fisher Scientific | HCSDCX7LEDPRO |
| **Other** | | |
| TMPRSS2 Activity Assay Kit | BPS Bioscience | #78083 |
| Cathepsin L Activity Assay Kit | BPS Bioscience | #79591 |
| uPA Activity Assay Kit | Millipore | ECM600 |
| iBlot™ 2 Dry Blotting System | Thermo Fisher Scientific | IB21001 |
| SpectraMax M5 plate reader | Molecular Devices | Model M5 |
| EnVision plate reader | PerkinElmer | #1041590 |
| Thermocycler (PCR system) | Eppendorf | Mastercycler pro |
| High-content CX7 microscope | ThermoFisher | HCSDCX7LEDPRO |
| Centrifuge (for spinoculation) | Eppendorf | 5810R |

## COVID-19 sc-RNASeq data analysis

Publicly available COVID-19 patient sc-RNASeq data were obtained from the Gene Expression Omnibus database under the accession code GSE145926 (Liao et al, 2020). Aligned and normalized transcript per million (TPM) values were obtained from the integrated ToppCell web server (Ardini-Poleske et al, 2017a). Frequencies and log2 fold changes were calculated from the normalized TPM values by the healthy controls. Of note,

monocytes, innate lymphoid cells, neutrophils, and dendritic cells, although detected in COVID-19 individuals, were not detected in uninfected individuals' BALF, thereby preventing the determination of fold changes for these cell types. Upregulated SERPINs were defined as those expressed to levels higher than 2-fold change using housekeeping gene B2M (one-tailed Mann–Whitney test, nonparametric; each donor was one value). We further performed statistical analysis comparing the upregulation in different cell subsets by one-tailed nonparametric Mann–Whitney test, where each donor was one independent value ($n = 3$ for mild, $n = 5$ for severe COVID-19; $P$ values listed in Dataset EV1).

## Generation of polarized human airway epithelial cultures

Differentiation of hTert-immortalized bronchial basal progenitor cells into pseudo-stratified human airway epithelial cultures was performed as stated in the following publication (de Vries et al, 2021). Briefly, Bci-NS1.1 cells (obtained from the Laboratories of Drs. Matthew Walters and Ronald Crystal, Cornell University) were seeded into Pneumacult™-Ex Plus Medium (StemCell) and passaged at least two times before plating ($7 \times 10^4$ cells/well) on rat-tail collagen type-1 coated permeable transwell membrane supports (6.5 mm; Corning Inc) to generate human airway epithelial cultures (HAE). HAE were maintained in Pneumacult™-Ex Plus Medium until confluent, then grown at an air-liquid interface with Pneumacult™-ALI Medium in the basal chamber for approximately 3 weeks to form differentiated, polarized cultures that resemble in vivo pseudo-stratified mucociliary epithelium.

## Viral infection of human airway epithelial cultures and transcript determination by RT-qPCR

All viral infections were performed in a BSL-2 environment, except infection with SARS-CoV-2, which was performed in a BSL-3 environment. For infection with influenza A/California/07/2009 virus (BEI resources, NR-13663), HPIV3-GFP (kind gift of Dr. Peter Collins), Rhinovirus A16 (kind gift of Dr. Ann Palmenberg), and SARS-CoV-2 WA-1 (BEI resources) cultures were washed twice with room-temperature PBS, then virus was added apically ($5 \times 10^6$ PFU for IAV, $1.12 \times 10^7$ FFU for PIV, 0.39 TCID$_{50}$ for Rhinovirus, and 1.35E5 PFU for SARS-CoV-2) in 50 µl of PBS and incubated for one hour at 37 °C for IAV and PIV, and at 4 °C for 90 min for Rhinovirus. Viruses were aspirated, and HAE cultures were washed apically once with room-temperature PBS, then incubated at 37 °C until endpoint. For infection with human Reovirus T3 SA (kind gift of Dr. Terrence Dermody) and Adenovirus 5-GFP (kind gift of Dr. Patrick Hearing), transwells were placed inverted in a sterile 5-cm Petri dish bottom, and all remaining basal media was aspirated from the basal side of the transwells. In all, 25 µl of each virus was added to the basal side of the transwells ($1.3 \times 10^8$ PFU for Reovirus and $8.85 \times 10^5$ PFU for Adenovirus) and incubated for one hour at 37 °C for Adenovirus and room temperature for Reovirus. Viruses were aspirated from the basal side of the transwells, and transwells were placed back into ALI media and incubated at 37 °C until endpoint. At each endpoint, transwell membranes were submerged in 375 µl Buffer

RLT (Qiagen), and cell lysate was frozen at −80 °C for future RNA extraction.

RNA was isolated using the Qiagen RNeasy kit following the manufacturer's instructions. Extracted RNA was then quantified via NanoDrop, and diluted to correct for yield differences. Relative mRNA levels, normalized to housekeeping gene RPS11, were determined by RT-qPCR (SuperScript III First Strand Synthesis System, Life Technologies, and PowerUP SYBR Green Master Mix, Thermo Fisher Scientific). Individual *P* values, obtained by 2-way ANOVA and Fisher's LSD comparison, are listed in Table EV1. Primers used are listed in Table EV2.

## SARS-CoV-2 infection of Calu-3 cells under the addition of extracellular rPAI-1

We used 5E4 PFU SARS-CoV-2 WA-1 per culture in a black, clear-bottom 96-well plate from Corning. Target rPAI-1 levels were chosen to reflect physiological concentrations, at 25 or 12.5 ng/ml, respectively. Heat-inactivated rPAI-1 was achieved by incubating the protein at 95 °C for 10 min before dilution in medium. Virus was diluted in DMEM and supplemented with rPAI-1 to the concentrations mentioned above diluted in PBS, or buffer (50 mM $NaH_2PO_4$, 100 mM NaCl, 1 mM EDTA, pH 6.6). Cells were incubated with virus dilutions, rocking for 2 h at 37 °C, then removed. rPAI-1 or buffer dilutions to the same concentration were replenished in after, diluted in DMEM (DMEM, 10%FBS, 1× Penicillin/Streptomycin, 1× non-essential amino acids) for the course of the infection until fixing. Cells were fixed in 10% Formalin overnight and used for immunofluorescence for the detection of SARS-CoV-2 Nucleocapsid protein-positive cells.

## Spike cleavage assay in BHK cells

BHK-21 cells were seeded into 12-well plates and transfected at ~70% confluency using Lipofectamine™ LTX with PLUS™ Reagent (Thermo Fisher Scientific) according to the manufacturer's instructions. Cells were co-transfected with plasmids encoding SARS-CoV-2 WA-1 Spike, TMPRSS2, and human PAI-1 or an empty vector control. In parallel, BHK cells were transfected with SARS-CoV-2 Spike and TMPRSS2 expression plasmids, and recombinant PAI-1 (rPAI-1) or buffer control was added directly to the cell culture supernatant at a final concentration of 25 ng/mL. After 24 h, cells were lysed and analyzed by SDS-PAGE followed by Western blotting. SARS-CoV-2 Spike was detected using the anti–S2 (1A9) antibody. Band intensities for full-length Spike (FL), S2, and cleaved S2′ were quantified using ImageJ, and the percent cleavage was calculated as the ratio of the S2′ band to the combined intensity of S2 + S2′ bands.

## Biosafety and work with biohazardous material

All work with infectious SARS-CoV-2 isolates was performed in the Biosafety Level 3 (BSL3) facility of NYU Grossman School of Medicine (New York, NY). The facility is registered with its respective local Departments of Health and passed inspections by the Centers for Disease Control & Prevention (CDC) within the year of submission of this manuscript. The BSL3 facility is operated in accordance with its Biosafety Manual and Standard Operating Procedures, including for the containment of biohazardous aerosols using certified biosafety cabinets and the facility's sealed ventilation system that provides a sustained directional airflow, from clean towards potentially contaminated areas, and HEPA-filtered exhaust. Biohazardous waste generated in the facility is fully decontaminated using approved disinfectants, followed by autoclaving and incineration as Regulated Medical waste. Access to the BSL3 facility is restricted to certified and authorized personnel, enrolled in occupational health surveillance programs, and wearing adequate Personal Protective Equipment (PPE), including OSHA-approved respirators, eye protection, spill-resistant coveralls, and double-gloves. When analyzed outside of the BSL3 facility, infectious samples were thoroughly treated using vetted inactivation methods. All work with infectious SARS-CoV-2 isolates was performed with prior approval of the Institutional Biosafety Committee (IBC) of NYU Grossman School of Medicine. Import permits for SARS-CoV-2 variant isolates were approved by the CDC.

## Assembly of the protease and SERPIN database and structure modeling

Protein structures were obtained directly from the Protein Data Bank (PDB) or modeled by homology using Swiss PDB modeler (Bienert et al, 2017; Guex et al, 2009; Waterhouse et al, 2018). If obtained from the PDB, structures were refined by removing solvents, co-factors, small molecules, and co-crystallized atoms. Selection of proteases for later docking was based on the LungMAP database (Ardini-Poleske et al, 2017a) of human airway-expressed proteases and protease families, mainly prioritizing those expressed to >10 Log2TPM on epithelial cells of donors. For proteases expressed as zymogens, we derived their active forms from the UniProt server (Consortium, 2021) and modeled the active forms. Active sites and SERPIN RCLs were mapped onto refined structures using the UniProt server (Consortium, 2021). The mutated protease or SERPIN structures were generated using Swiss modeler 2.0 (Bienert et al, 2017; Guex et al, 2009; Waterhouse et al, 2018); Swiss modeler was preferred over AlphaFold for in-silico mutagenesis, as the AlphaFold algorithm disregards structural changes caused by introduced mutations.

## SERPIN-protease docking using HADDOCK v2.4

Docking of SERPIN-protease complexes were performed using the HADDOCK v2.4 server (de Vries et al, 2010; Honorato et al, 2021). HADDOCK predicts how two or more molecules (in this case, one SERPIN and one protease) interact with each other, creating a complex (in this case, the binding complex depicted in Fig. EV1A). The process starts with the full protein length 3D structures of protease and SERPIN, either solved experimentally or homology-modeled using Swiss modeler 2.0 (Bienert et al, 2017; Guex et al, 2009; Waterhouse et al, 2018). Next, we entered information about which parts of the molecules might interact based on experimental data or predictions; this information is called Ambiguous Interaction Restraints (AIRs). For a SERPIN-protease interaction, the AIRs are the SERPIN Reactive Center Loop (RCL) and the protease's active site. We thus mapped the protease active site and SERPIN RCL by examining individual entries on UniProt and then entered them as AIR. In the first step of the docking process, the molecules are treated as rigid bodies, and HADDOCK tries to find the best way to fit them together based on the AIRs. Next,

HADDOCK allows parts of the molecules near the interaction sites to move slightly to improve the fit. Finally, HADDOCK performs a more detailed refinement where the entire complex can move slightly to find the most stable configuration. HADDOCK evaluates many possible configurations and scores them based on how well they fit the AIRs and other physical and chemical properties. The HADDOCK score is calculated by a linear formula and is directly proportional to van der Vaals energy, de-solvation energy, AIR energy, and electrostatic energy of the interaction. It selects the top-scoring configurations as the most likely models of the interaction (Dominguez et al, 2003; Honorato et al, 2024).

Selection of primary residues to perform docking was based on the residues specified on UniProt as the reactive center loop's 8 critical residues found to dictate SERPIN specificity—P4-P4' for SERPINs and the catalytic triad within the protease active site for proteases. Secondary residues, also taken into account while guiding the docking, were automatically selected within the range of a 6.5 Å radius from each of the primary residues. Furthermore, the full RCL was given full flexibility, starting at the poly-alanine segment and ending at the PF loop end motif. Upon submission and finish of an entry on the HADDOCK v2.4 server, we evaluated the resulting complex position and specifically selected complexes that had RCLs fit inside the protease active site, if no such complex was found, the protease was removed from the screen. We then used the score of the visually fitting complexes and used it to guide our initial optimization and calculate thresholds for the z-scores. Structures of the complexes were then saved and visualized on PyMoL v2.0 for distance calculation and deeper examination of the positioned residues (Fig. 4).

PAI-2 was docked to the active structure of AVP, obtained from the Protein Data Bank (PDB) under the entry 1AVP, which represents AVP bound to the pVIc cofactor. The inactive form of AVP was retrieved from the PDB under the entry 4EKF, representing the unbound state of AVP.

## Statistical analysis of raw HADDOCK scores and conversion into normalized HADDOCK scores

To define the thresholds of SERPIN-protease docking, Haddock scores were determined from at least two positive and two negative control proteases for each SERPIN, known to bind or not bind SERPINs in vitro, respectively. For each SERPIN, average HADDOCK scores were calculated separately for the positive and negative controls, and the mean of both averaged scores served as the in-silico threshold of SERPIN-protease interaction. In cases where negative controls were not available, we generated in-silico mutants in the SERPIN reactive center loop (RCL). The mutated SERPIN structures were generated using Swiss modeler 2.0 (Bienert et al, 2017; Guex et al, 2009; Waterhouse et al, 2018), and then docked to positive control proteases. The Swiss modeler was preferred over AlphaFold for in-silico mutagenesis, as the AlphaFold algorithm disregards structural changes caused by introduced mutations. Positive controls were the following: SERPINB1, neutrophil elastase and cathepsin G; SERPINB2, uPA and tPA; SERPINB4, cathepsin G and chymase; SERPINB8, furin and subtilisin A; SERPINB9, granzyme B and caspase-1; SER-PINB10, cathepsin L and cathepsin V; SERPINB13, cathepsin L and cathepsin K; SERPINC1, thrombin and TMPRSS7; SERPINE1, uPA and tPA; SERPING1, C1 and plasma kallikrein. Negative controls

were the following: SERPINB1, thrombin and granzyme A; SERPINB2, full RCL mutant (converting the P4-P4' residues to alanines); SERPINB4, granzyme B; SERPINB8, uPA and subtilisin B; SERPINB9, trypsin and thrombin; SERPINB10, loss-of-function (LoF) point mutant (converting the P1 to alanine) and full RCL mutant (converting the P4-P4' residues to alanines); SERPINB13, LoF point mutant and full RCL mutant; SERPINC1, full RCL mutant; SERPINE1, furin; SERPING1, full RCL mutant.

The SERPIN-specific HADDOCK thresholds were used to normalize the full dataset of HADDOCK scores and to generate Z-scores, i.e., the thresholds were set to zero. Violin plots were generated using GraphPad Prism v.9, and clustered heatmaps of the normalized, Z-scored data were generated using the Complex-Heatmap package in program R v.4.1.2 (Team, 2014).

## Antibodies, recombinant proteins, and inhibitors

To treat cells with rPAI-1-, we used recombinant active human PAI-1 (carrying stabilizing mutations N150K, K154T, Q319L, and M354I, Abcam) in experiments on human airway epithelial cultures and in in vitro complex formation assays. To deplete PAI-1 in human airway epithelial cultures infected with SARS-CoV-2, we used anti-PAI-1 antibody (Proteintech, 13801-1-AP). A mouse IgG1 isotype control (clone MOPC-21; ThermoFisher, MA1-10406) was used as a negative control for antibody depletion experiments. rTMPRSS2 was obtained from Biomatik (RPC24995), rCTSL from R&D Systems (952-CY-010), rCTSB from R&D Systems (953-CY-010) and recombinant active urokinase plasmi-nogen activator from Abcam. Triplaxtinin (PAI-039, Axonmed-chem) was used as a PAI-1 inhibitor for in vitro assays at 10 μM. To visualize Spike in western blots, we used anti SARS-CoV-2 Spike S2 (1A9) mouse monoclonal antibody (ThermoFisher, MA5-35946) at 1:1000 dilution in 3% BSA in TBS-T (0.01% Tween). For visualization of SARS-CoV-2 infected cells via detection of N protein in Calu-3 cells, we used anti SARS-CoV-1/2 Nucleocapsid Mouse anti-N clone (Cell Signaling, 1C7C7) 1:1000 50 μL/well. For visualization of myc-tagged rAVP in western blots, we used c-Myc Antibody (ThermoFisher, MA1-980). For visualization of PAI-2 in western blots, we used anti-PAI-2 antibody (ThermoFisher, PA5-27857) at 1:1000 dilution. For detection, we used HRP-conjugated goat anti-mouse IgG (Invitrogen, G21040) and HRP-conjugated goat anti-rabbit IgG (Invitrogen, G21234) for western blot, and Alexa Fluor® 647-conjugated donkey anti-mouse IgG (Jackson ImmunoResearch, 715-605-151) or Alexa Fluor Plus 647-conjugated anti-goat IgG (Jackson ImmunoResearch, 715-605-151) for immunofluorescence at 1:1000 and 1:10,000, respectively.

## SERPIN-protease complex formation assays

Reactions were set up in PCR tubes containing a constant amount of 0.5 μg or 0.1 μg active recombinant PAI-1, and 0.5 μg of rPAI-2, co-incubated with a protease or not, respectively, per reaction. Reactions were conducted in the following buffers: TMPRSS2 and uPA reactions used 10 mM Tris-HCl, 200 mM NaCl, 10 mM EDTA, pH 7.4; Cathepsin L used 50 mM MES, 5 mM DTT, 1 mM EDTA, pH 6.5 and pH 5.5 for early and late endosome mimics, respectively; AVP-containing particles reactions used 10 mM TRIS-HCl, 10 mM EDTA, 20 mM NaCl, pH 8. Protease concentrations were used as follows: TMPRSS2, 0.1 μg; Cathepsin L, 0.25 μg, and 1 μg when incubated

alone; uPA, 1 μg and 2 μg. Reaction tubes were incubated in a thermocycler for 10 min at 25 °C and 30 min at 37 °C, consecutively. When applicable, Triplaxtinin was used at 10 μM.

## Fluorometric host protease activity assays

Commercially available recombinant protease activity kits were used to quantify activity and inhibition of the mentioned proteases in the presence of rPAI-1. The TMPRSS2 kit from BPS Bioscience (#78083) was used, and instructions were followed from the kit for kinetic measurements. rPAI-1 was used at 0 nM, 0.025 nM, and 0.25 nM. Cathepsin L kit from BPS Bioscience (#79591) was used, and instructions were followed, with the only modification of adjusting the pH of the 1× reaction buffer solution to pH 6.5 to maximize PAI-1 activity. This kit was also used for Cathepsin B screening, only by changing the enzyme. Both enzymes were used at a final concentration of 0.5 nM, and rPAI-1 was added to a final concentration of 0.25 nM and 0.5 nM, or 0.5 nM and 2.5 nM for Cathepsin L and B, respectively. uPA kit from Millipore (#ECM600) was used and instructions followed. Fluorescence or absorbance was measured over time using a SpectraMax M5 model by Molecular Devices (for TMPRSS2) and Cathepsins B and L, and uPA substrate emitted fluorescence or changes in absorbance were measured in kinetic mode using EnVision plate reader (#1041590 EnVision). Reaction slopes were calculated by fitting the raw time-course values curve to a simple linear regression model. Slopes were then normalized to the enzyme alone to obtain the relative rate.

## Fluorometric AVP activity assay

In a black-bottom (top-read) 96-well plate, the assay was prepared in triplicate for each condition. For the sonicated particles assay, rhodamine-linked AVP peptide was resuspended to achieve a final concentration of 5 μM. Sonicated particles at a concentration of 3.75E10 particles/μL were incubated with recombinant PAI-2 at low (0.1 nM) and high (10 nM) concentrations in assay buffer. Each reaction mixture was set with 0 or 10 μL of rPAI-2 (0.5 μg/μL), 0 or 20 μL of disrupted AdV particles.

For reactions with individual co-factors and rAVP, reaction components were incubated in the following final concentrations: rhodamine-linked AVP peptide at 5 μM, rAVP at 2.5 nM, rPAI-2 at 10 nM, pVIc at 3.5 nM, and DNA random hexamers at 7 μM.

Assay buffer (10 mM Tris-HCl, 20 mM NaCl, 10 mM EDTA, pH 8.0) was added to adjust the volume to 50 μL per well. Each component was pre-diluted to 10× the final concentration before being added to the reaction. The mixtures were incubated at 37 °C for 30 min before adding the peptide. Fluorescence was read kinetically for 60 min with an excitation wavelength of 492 nm and an emission wavelength of 523 nm as described above. Data were collected and analyzed based on the change in fluorescence. Reaction slopes were calculated by fitting the raw time-course values curve to a simple linear regression model. Slopes were then normalized to the enzyme alone to obtain the relative rate.

## SERPINB2 knockdown assay

HEK-293T cells were seeded at a density of $2 \times 10^{4+}$ cells/well in a 96-well plate. The following day, cells were transfected with either control siRNA (Silencer™ Negative Control No. 1 siRNA), or SERPINB2-targeting siRNAs (Thermo Fisher Scientific, S10018) at final amounts ranging from 7.5 pmol per well in Opti-MEM (Thermo Fisher Scientific), using RNAiMAX as the transfection reagent. Complexes were incubated for 15 min at room temperature before being added to the cells. Knockdown was validated by western blot as described above using anti-PAI-2 antibody (ThermoFisher, PA5-27857) at 1:1000 dilution.

At 48 h post-transfection, cells were infected with AdV5-GFP for 72 h. Following infection, cells were fixed with 50 μL of 8% paraformaldehyde in PBS for 30 min at room temperature. To visualize infection, cells were permeabilized with 0.1% Triton-X-100 in PBS for 4 min, blocked with 2% BSA/PBS, and stained with a primary goat anti-Adenovirus antibody (ABIN236692, 1:1000) diluted in 0.1% BSA/PBS for 2 h at 37 °C. After PBS washes, secondary detection was performed using Alexa Fluor Plus 647-conjugated anti-goat IgG (Jackson ImmunoResearch, 715-605-151, 1:2000) and DAPI (1:1000) for 1 h at 37 °C. Plates were imaged using the CellInsight CX7 high-content screening platform for both anti-Adenovirus and GFP signal.

## AdV spread assay

A549 cells at 80% confluency in a 48-well plate were transduced with pSCRPSY vectors encoding either empty vector, PAI-2, or IRF-1 using spinoculation at 1200 rpm for 45 min at 37 °C, followed by a 6-hour incubation at 37 °C. After 48 h, the cells were infected with HAdV5-GFP, using a viral stock at 3.54E7 FFU/mL, at a multiplicity of infection (MOI) of 1. After 96 h post-infection, supernatants were collected and titrated by TCID50 assay in A549 cells over 72 h in a 96-well plate. The cells' nuclei were stained with DAPI, and GFP was measured via high-content microscopy, as previously described. The TCID50 was determined by scoring the presence of GFP, indicating viral infection and replication.

## Cell line quality control

All cell lines are tested for Mycoplasma and authenticated when received from external sources. When purchased from a vendor, the lot-specific information is reviewed carefully.

## Graphics

Graphics were created with BioRender.com.

# Data availability

This study produced no data deposited in external repositories.

The source data of this paper are collected in the following database record: biostudies:S-SCDT-10_1038-S44318-025-00546-6.

# Peer review information

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

## Acknowledgements

We thank all members of the Dittmann lab for critical input on the data and manuscript. We further thank Benoit Roger, Jean-Baptiste Vergnes, and Harald Wodrich for fruitful discussions on PAI-2 and AVP. We are grateful to Michael Garabedian for mentoring and support during the preparation of this manuscript. We thank Dr. Terrence Dermody for the kind gift of Reovirus T3 SA, Dr. Ann Palmenberg for the kind gift of Rhinovirus A16, and Dr. Peter Collins for the kind gift of PIV3-GFP. Research was supported by the following grants from NIH/NIAID: R01AI143639 and R21AI139374 to MD, as well as T32AI100853. Work was further supported by The Vilcek Institute of Graduate Biomedical Sciences, and by NYU Grossman School of Medicine Startup funds. Walter F Mangel was supported by the Office of Biological and Environmental Research of the U.S. Department Energy under Prime Contract no DE-AC0298CH10866 with Brookhaven National Laboratory.

## Author contributions

**Joaquin Rodriguez Galvan**: Conceptualization; Resources; Data curation; Software; Formal analysis; Supervision; Validation; Investigation; Visualization; Methodology; Writing—original draft; Project administration; Writing—review and editing. **Maren de Vries**: Data curation; Writing—review and editing. **Shiraz Belblidia**: Data curation; Formal analysis; Writing—review and editing. **Ashley Fisher**: Data curation; Writing—review and editing. **Rachel A Prescott**: Data curation; Writing—review and editing. **Keaton M Crosse**: Data curation; Writing—review and editing. **Patrick Hearing**: Data curation; Writing—review and editing. **Walter F mangel**: Conceptualization; Writing—review and editing. **Ralf Duerr**: Data curation; Formal analysis; Visualization; Writing—original draft; Writing—review and editing. **Meike Dittmann**: Conceptualization; Resources; Software; Supervision; Funding acquisition; Validation; Investigation; Visualization; Writing—review and editing.

Source data underlying figure panels in this paper may have individual authorship assigned. Where available, figure panel/source data authorship is listed in the following database record: biostudies:S-SCDT-10_1038-S44318-025-00546-6.

## Disclosure and competing interests statement

The authors declare no competing interests.

# Expanded View Figures

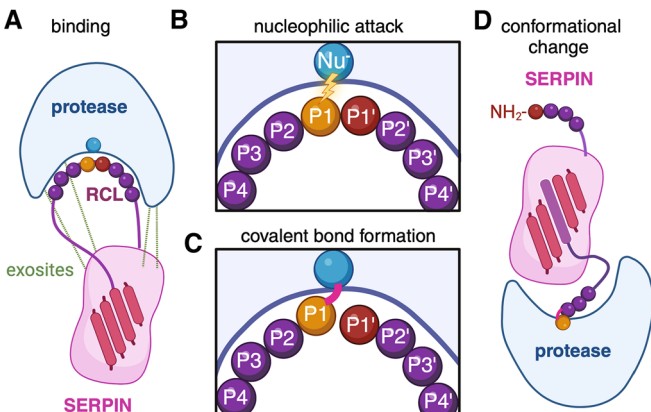

**Figure EV1. Molecular mode of action for inhibitory SERPINs.**

(**A**) A SERPIN binds to a target protease by inserting its reactive center loop (RCL, purple line) into the protease's catalytic center. This *binding step* is facilitated by the 3D fit of a given RCL into the catalytic center of the protease and can be enhanced by the formation of secondary binding sites ("exosites", green dashed lines). (**B**) The RCL core sequence (named P4-P4′) mimics the core sequence of the canonical protease substrate. *Nucleophilic attack* by the protease cleaves the SERPIN at the P1-P1′ bond. (**C**) Protease and SERPIN form a *covalent complex* (acyl-enzyme intermediate). (**D**) The ensuing rapid and significant *conformational change*, where the RCL attached to the protease inserts itself into a ß-sheet center, prompts the formation of a stable inhibitory complex between SERPIN and protease, akin to a "mousetrap".

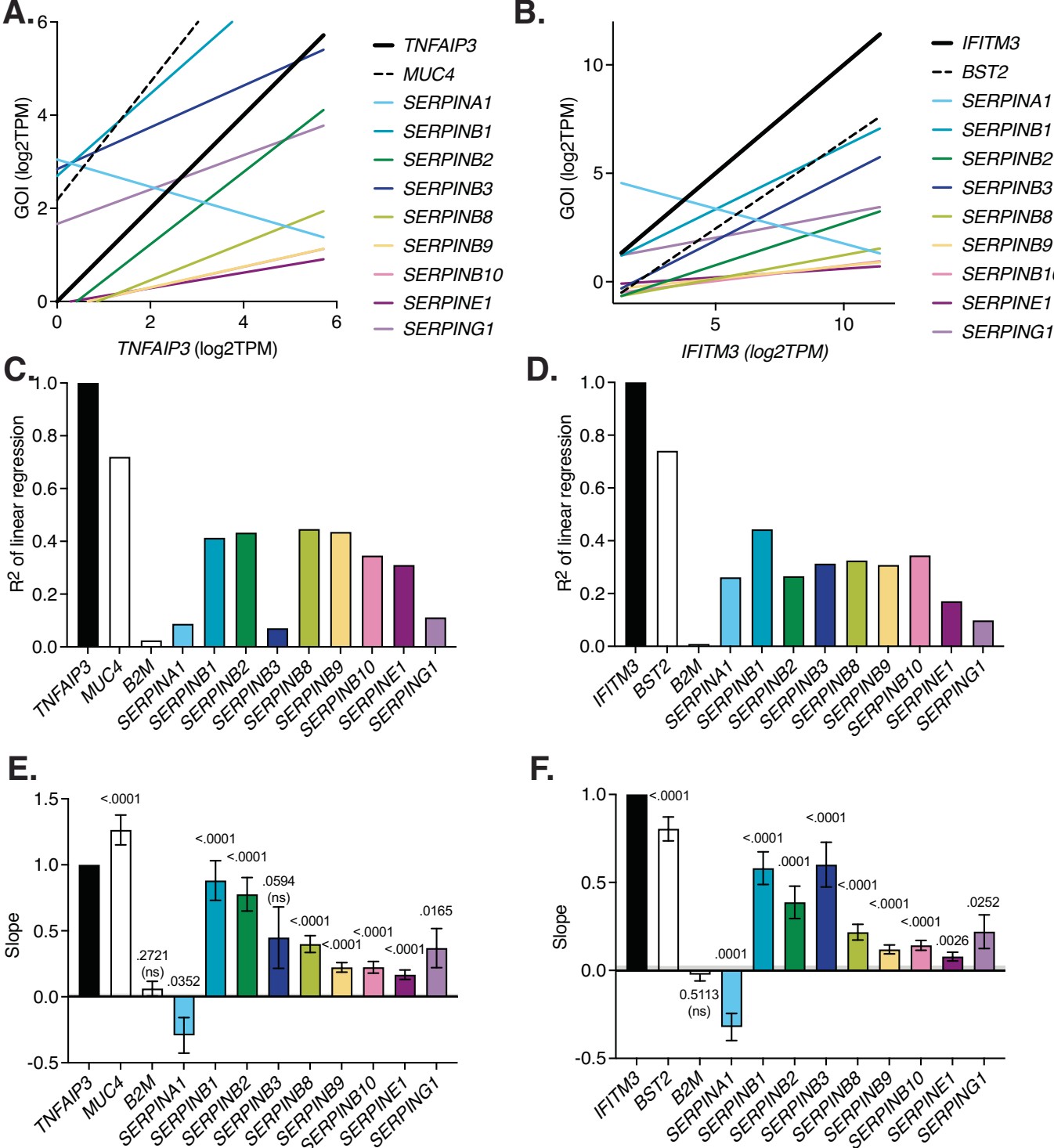

◀ **Figure EV2. Correlation analysis of SERPIN mRNA with TNF-alpha or type I interferon-induced genes.**

(A) Linear regression plot with correlation analysis of select mRNA levels in reference to canonical TNF-alpha-regulated gene TNFAIP3. MUC4, TNF-alpha-regulated gene positive control. (B) Linear regression plot with correlation analysis of select mRNA levels in reference to canonical type I-interferon-regulated gene IFITM3. BST-2, Interferon-regulated gene positive control. (C, D) R2 values of the linear regression plots from (A, B), respectively. (E) Magnitude of slopes of linear regression from (A) and linear regression statistics testing that the slope is significantly non-zero. Mean $+/-$ SD. Exact $P$ values MUC4 $P = 3.85 \times 10^{-15}$; B2M $P = 0.2721$; SERPINB8 $P = 8.4 \times 10^{-8}$; SERPINE1 $P = 2.21 \times 10^{-5}$; SERPINB2 $P = 1.55 \times 10^{-7}$; SERPINB1 $P = 3.63 \times 10^{-7}$; SERPING1 $P = 0.0165$; SERPINB3 $P = 0.0594$; SERPINA1 $P = 0.0352$; SERPINB9 $P = 1.38 \times 10^{-7}$; SERPINB10 $P = 5.65 \times 10^{-6}$; BST2 $P = 1.37 \times 10^{-7}$; MUC16 $P = 1.3 \times 10^{-10}$; IFITM3 $P = 1.19 \times 10^{-9}$, $n = 3$ technical replicates (F). Magnitude of slopes of linear regression from (B) and linear regression statistics testing that the slope is significantly non-zero. GOI, gene of interest; ns: not significant. Exact $P$ values SERPINB8 $P = 1.26 \times 10^{-5}$; SERPINE1 $P = 0.00258$; SERPINB2 $P = 1.1 \times 10^{-4}$; SERPINB1 $P = 9.85 \times 10^{-8}$; SERPING1 $P = 0.0252$; SERPINB3 $P = 1.98 \times 10^{-5}$; SERPINA1 $P = 1.26 \times 10^{-4}$; SERPINB9 $P = 2.39 \times 10^{-5}$; SERPINB10 $P = 6.05 \times 10^{-6}$; BST2 $P = 5.61 \times 10^{-16}$, $n = 3$ technical replicates. $*P < 0.05, **P < 0.005, ***P < 0.0001, ****P < 0.00001$.

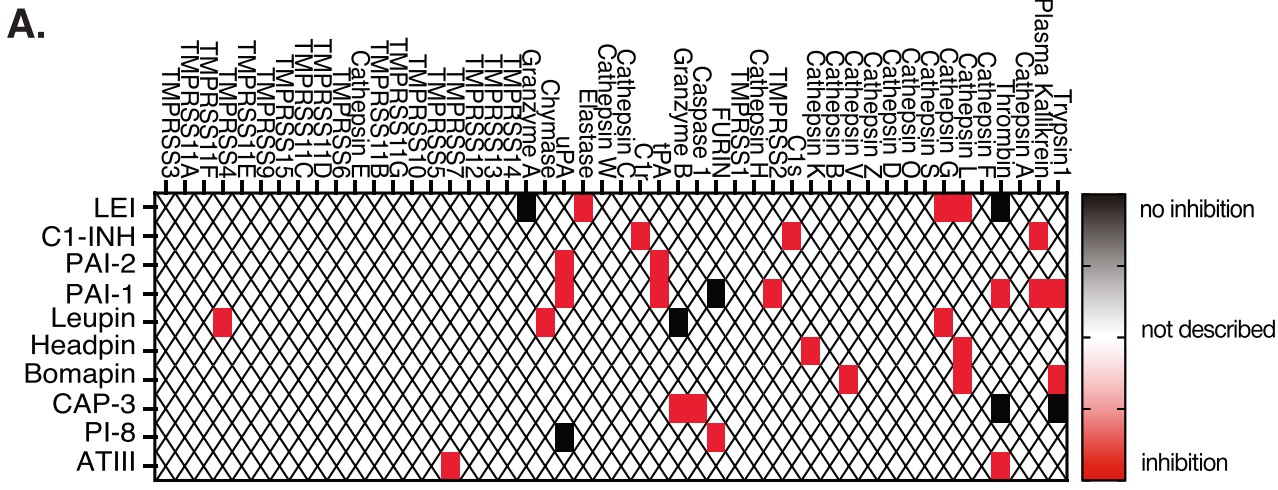

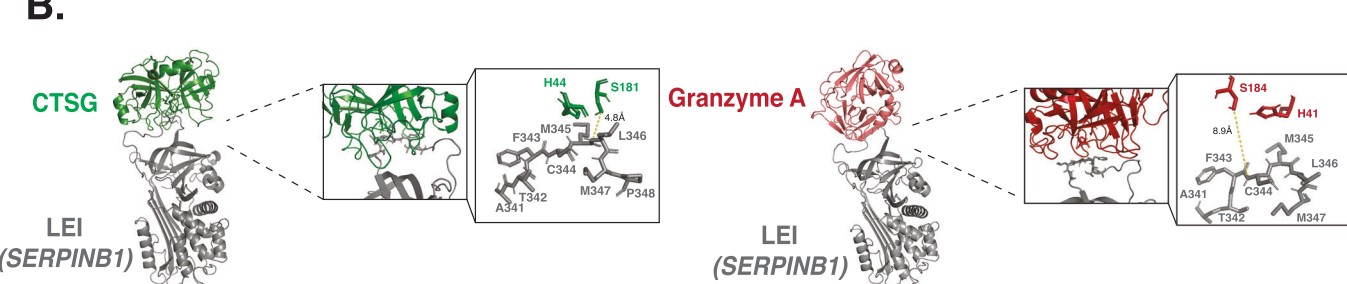

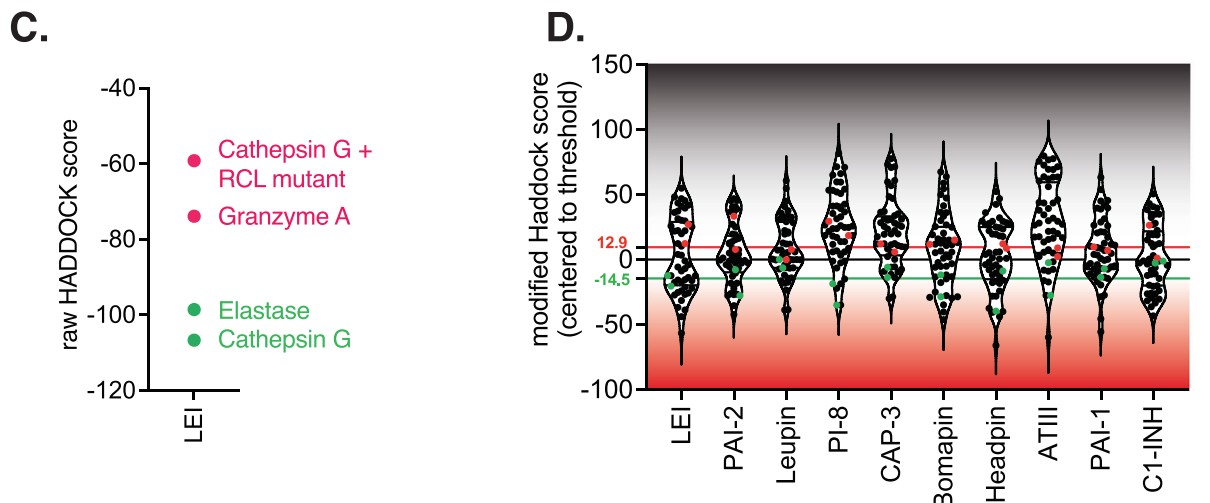

◀ **Figure EV3. Establishing the in-silico screen for identifying SERPIN-protease pairs.**

(A) Previously described SERPIN-protease pairs in the format of Fig. 3B. (B) 3D structure overview featuring leukocyte elastase inhibitor (LEI, encoded by *SERPINB1*, gray) docked to Cathepsin G (CTSG, green). Detailed interface view showing LEI RCL and protease active sites. Zoom-ins reveal residues involved in catalysis: nucleophile S181 and its general base H44 are positioned opposite LEI's M345↓L346 with S181 and M345 in close proximity (4.8 A). LEI (gray) docked to Granzyme A (red). Detailed interface view showing LEI RCL and protease active sites. Zoom-ins reveal residues involved in catalysis: nucleophile S184 and its general base H41 are positioned opposite LEI F343↓C344, with S184 and M344 further apart (8.9 A). (C) Raw HADDOCK scores for SERPINB1 with positive controls (Cathepsin G, Elastase) or negative controls (RCL mutant + Elastase, Granzyme A). More negative scores indicate favorable binding. (D) Violin Plots of normalized HADDOCK scores for $n = 10$ SERPINs docked to $n = 48$ proteases and to respective negative and positive controls (listed in "Methods"). LEI, leukocyte elastase inhibitor encoded by *SERPINB1*; PAI-2, plasminogen activator inhibitor 2 encoded by *SERPINB2*; Leupin, encoded by *SERPINB4*; PI-8, protease inhibitor 8 encoded by *SERPINB8*, CAP-3, cytoplasmic anti-protease 3 encoded by *SERPINB10*, Bomapin encoded by *SERPINB10*, Headpin encoded by *SERPINB13;* ATIII, antithrombin 3 encoded by *SERPINC1;* PAI-1, plasminogen activator inhibitor 1 encoded by *SERPINE1;* C1-INH, C1 inhibitor encoded by *SERPING1*. Black line, mean of all 480 values set to 0; red line, mean negative control; red background, range for predicted non-binders; green line, mean positive control; green background, range for predicted binders; red data points, negative controls; green data points, positive controls. Raw HADDOCK scores and normalization in Dataset EV2.

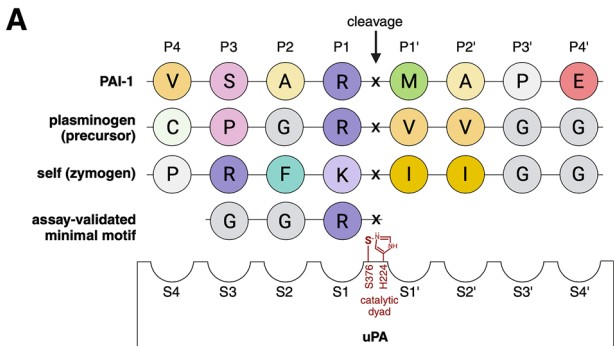

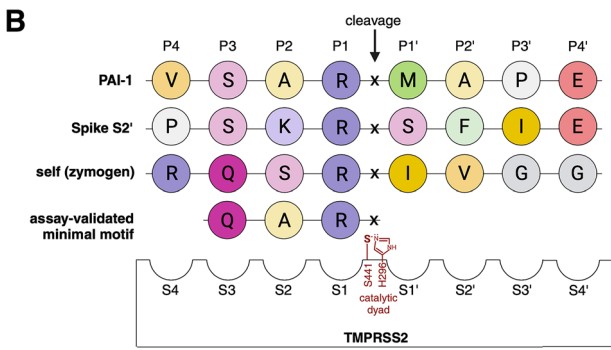

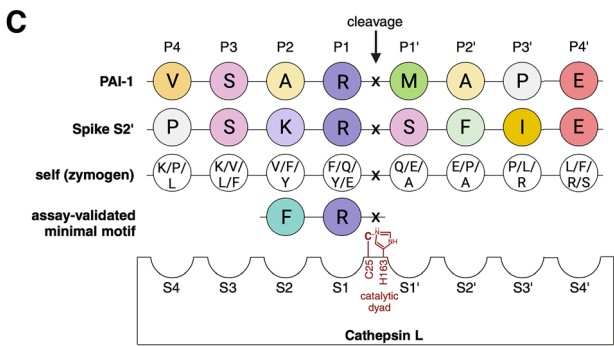

**Figure EV4. Schematic representation of substrate P4-P4′ motifs opposite proteases' catalytic pockets S4-S4′ for uPA, TMPRSS2 and Cathepsin L.**

(A–C) Schematic of fit of substrate residues into protease pockets for uPA, TMPRSS2, and cathepsin L, respectively. Substrates include zymogen self-cleavage region, canonical or previously described substrate, PAI-1 and fluorophore-linked peptide substrate used in this study. Amino acids are colored according to their side chain chemistry (Unipro UGENE): basic (R, K) litmus blue with R being more basic and darker; acidic (E, D) litmus red with more acidic being darker; hydrophobic (I, L, V, A), yellow with intensity corresponding to hydrophobic character; sulfur-containing (C, M) green; aromatic (F, Y, W) in teal; polar (N, Q, S, T) magenta/pink with darker coloring for more polarity; non-polar glycine (G) in dark gray and proline (P) in light gray. Assembled from references (Koga et al, 1990; Menard et al, 1998; Rossignol et al, 2004; Shrimp et al, 2020) and UNIPROT.

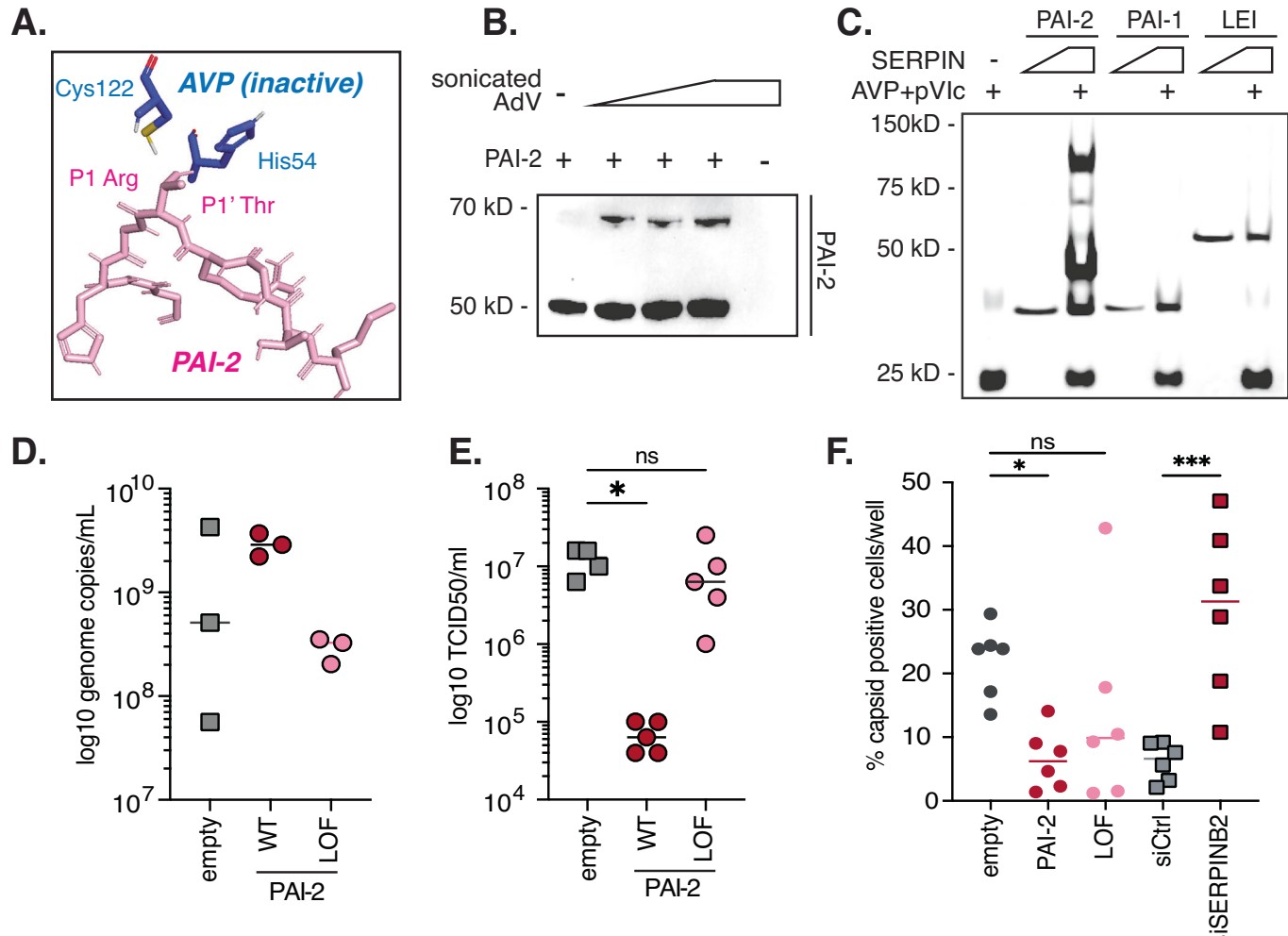

**Figure EV5. PAI-2 binding assays to inactive AVP and expression quality control.**

(A) Docking structure of top-scoring complex for PAI-2 with inactive AVP (structure solved without pVIc binding). (B) Western blot of AVP from sonicated adenovirus particles (23 kDa, not visible; bound to pVIc and DNA and thus in its most active form) mixed with rPAI-2 (47 kDa) and probed for PAI-2. (C) Silver-stained SDS-PAGE of AVP+pVIC incubated with PAI-2, PAI-1, or LEI in increasing concentrations (1-7.5 ng). (D) Genome copy numbers as quantified by qPCR of supernatants derived from HAdV5-infected A549s at 72hpi, transduced with either empty PAI-2 or LoF expression construct, $n = 3$. 6 h. (E) TICD50 of supernatants derived from HAdV5-infected A549s, transduced with either empty, PAI-2 or LoF expression constructs. Exact $P$ values: empty vs. PAI-2 $P = 0.0271$, empty vs. LOF $P = 0.7372$, $n = 5$. (F) AdV capsid protein-positive HEK-293T cells per well after transfection with either empty, PAI-2 or LOF expression constructs, or control siRNA, SERPINB2 siRNA, followed by 72 h AdV5-GFP infection. Western blot and % infected. Statistical analysis by ANOVA, *$P < 0.05$; ns, non-significant. Exact $P$ values: empty vs. PAI-2 $P = 0.0264$, empty vs. LOF $P = 0.4381$, siCtrl vs siSERPINB2 $P = 0.0004$, $n = 6$.

