## [Peer Review File · The EMBO Journal]

Host cell and viral protease targets of human SERPINS identified by *in silico* docking

Joaquin Rodriguez Galvan, Maren de Vries, Shiraz Belblidia, Ashley Fisher, Rachel Prescott, Keaton Crosse, Patrick Hearing, walter mangel, Ralf Duerr, and Meike Dittmann

Corresponding authors: Meike Dittmann (Meike.Dittmann@nyulangone.org) , Ralf Duerr (Ralf.Duerr@nyulangone.org)

Review Timeline:

Transferred from Review Commons:	14th Jun 24
Editorial Decision:	19th Jun 24
Appeal:	20th Jun 24
Editor's Correspondence:	26th Jun 24
Revision Received:	29th Oct 24
Editorial Decision:	25th Nov 24
Revision Received:	9th Mar 25
Editorial Decision:	20th May 25
Revision Received:	22nd Jul 25
Accepted:	5th Aug 25

Editor: Ieva Gailite

Transaction Report:

This manuscript was transferred to The EMBO Journal following peer review at Review Commons.

Review #1**1. Evidence, reproducibility and clarity:****Evidence, reproducibility and clarity (Required)**

Title - In-silico docking platform with serine protease inhibitor (SERPIN) structures identifies host cysteine protease targets with significance for SARS-CoV-2

Authors - Joaquín J Rodriguez Galvan, Maren de Vries, Shiraz Belblidia, Ashley Fisher, Rachel A Prescott, Keaton M Crosse, Walter F. Mangel, Ralf Duerr, Meike Dittmann

****Summary****

The finding that PAI-1 has cross class serpin functions is of definite interest given the roles of PAI-1 in regulation of physiological processes as well as in driving pathology. PAI-1 is generally considered to be a key regulator of thrombolysis and thus an effect on other pathways and even intracellular pathways is of interest. Examining airway epithelial proteases and serpins is of definite interest in respiratory viral infections. Broadening the targets for serpins is also of very definite interest. This study ranges from an overview of prior published work on bronchoalveolar lavage samples and serpin expression, a tissue culture analysis of lung epithelial cells and expression of proteases and serpins is assessed. In addition selective changes in serpin expression and protease targets are assessed by in silico analysis as well as proof of concept via Western blot and fluorometric analysis. This is an extensive study and of definite interest.

There are some limitations as with any study, albeit the study overall is excellent. The authors do not reference the prior work which has examined cross class serpins, viral and mammalian, - this should be noted as alternative protease targets are known. They do mention the modulation of protease targets by glycosaminoglycans in the discussion. Further, serpins are inhibitors, thus while the RCL provides a target for a protease, but the response may not be fully selective in vivo as the protease has to be present and active to complete the serpin protease interaction. The bronchiolar lavage analysis is excellent but cannot differentiate epithelial cell and associated immune cells and their roles in the response. PAI-1 does not seem to be present in the bronchoalveolar lavage samples. Further discussion of prior work with cross class serpins and also the limitations of the in silico analyses and the lavage specimens should be provided. Further analysis of the detected proteases that are reported here to bind to PAI-1 would be of great interest. The data from the bronchoalveolar lavage is published.

Overall this is a very simple, although extensive and excellent, study analyzing a wide range of data from patients with bronchoalveolar lavage and epithelial cell samples, human epithelial cell cultures after infection with a range of respiratory viruses as well as the development of a 3D in silico analysis of potential protease and serpin interactions. These correlations between changes in serpin and protease expression with viral infections and potential new interactions for serpins with previously non identified proteases is of clear interest. This shows an excellent correlation but as with big data sets this does not provide a true cause and effect - rather providing new potential directions for analysis of these interactions in viral infections in lung epithelium and this is valuable as a basis for ongoing studies. Prior work evidencing other cross class serpin protease targets as well as limitations related to the analyses as discussed in the critiques above should be noted and the abstract and title could better describe and define the studies as performed.

Critique

****Major****

1. Cross class serpin interactions are known and have been reported for at least two viral serpins Serp-1 and CrmA - both of which bind cysteine proteases as well as serine proteases as well as the mammalian SCCA serpins
2. The protease targets are reported to vary when interacting with glycosaminoglycans such as heparan sulfate - PAI-1 inhibits thrombin in the presence of heparin - thus while a canonical serpin suicide inhibition is considered specific - it can vary. This is noted in the discussion
3. What is the potential impact of the noted interactions of PAI-1 with other proteases such as cathepsin - PAI-1 is considered to have predominately extracellular functions, but prior work indicates internalization of PAI-1 when bound to the uPA/uPAR complex with alterations in intra cellular activation
4. This is supported by basic in vivo and in vitro serpin and protease interactions that are demonstrated confirming in silico analyses, eg. gel shift analyses or even Mass spectrometry analysis particularly for PAI-1
5. Per the authors "To date, three SERPINS have been studied in the context of innate antiviral defense: PAI- 1 (encoded by SERPINE1) against influenza viruses encoding hemagglutinin H1 and SARS-CoV-2, by impeding the proteolytic maturation of H1 or spike, respectively^{19,20}; alpha-1-antitrypsin (encoded by SERPINA1) and antithrombin (encoded by SERPINC1) against SARS-CoV-2, likely through the inhibition of TMPRSS2, by reducing maturation of spike, although direct inhibition of TMPRSS2 by either SERPIN was not

shown". This is partially complete however other serpins such as C1Inh and one virus derived serpin that have been analyzed for efficacy in treating SARS

6. While TMPRSS2 is indeed a serine protease - Beneficial effects of some serpins may be due to modulation of the immune response as opposed to selective anti-viral responses. The immune / cytokine storm and coagulopathies (with clotting and even hemorrhage) seen in the excess inflammatory response that causes respiratory vascular leak and severe viral sepsis. PAI-1 targets tPA and uPA - uPA has marked proinflammatory actions when bound to the uPA receptor (uPAR) and can activate growth factors and MMPs which can enhance immune cell invasion - PAI-1 binds to the uPA / uPAR complex which can thus also alter inflammatory cell responses and cell activation when internalized.

7. The RCL does in general incorporate P4 to P4' but can vary from this specific P4 to P4' sequence

8. How accurately does in silico protease serpin analysis predict real interactions? - this should be discussed as HADDOCK may have some limitations - This is outside my field of expertise

9. The data from a published study examining bronchoalveolar lavage fluid single cell transcriptional analysis from patients with and without COVID - mild and severe - and with comparison to patients without COVID does demonstrate altered protease and serpin activity - but does not indicate specific interactions

10. What is the significance for changes in gene expression in epithelial cells versus macrophage T and B cells looks - This looks like a small change like a small change in the mean values Figure 1b

11. The more common names for the SERPINS as detected in COVID alveolar lavage samples would be helpful in figure 1 - and specifically labelling PAI-1 as this is a focus for this study - together with the known SERPIN nomenclature or under abbreviations - For example SERPINB2 is PAI-2 and SERPING1 is C1INH and SERPINA1 is alpha 1 antitrypsin

12. Of interest - is the bronchoalveolar lavage fluid likely to contain both epithelial cells as well as immune response macrophage, T cells and NK cells etc - one assumes single cells were identified and isolated- Is this defined?

13. The known previously reported target proteases for PAI-1 should be noted

14. SERPINE1 is not noted in figure 1 - this is PAI-1 - but is seen in the HAEC infection model data

15. "To overcome this limitation, we developed a computational method to predict 3D interactions between SERPINS and proteases, simulating the binding process depicted in Supplemental Figure 1a. Specifically, we employed High Ambiguity Driven protein- protein Docking (HADDOCK), a tool that predicts complex structures, integrating experimental and computational data^{35,36}." This analysis looks to be extensive however this is a correlation - not a true analysis of cause and effect This does however have the potential to identify

significant interactions - In future it might be of interest to assess PAI-1 given to infected cultures to assess viral replication and titers or perhaps examine a knock out cell model?

16. Why does supplemental figure 2 show SERPINB1 and not PAI-1

17. As PAI-1 was identified as having new cathepsin protease binding in addition to TMPRSS2 - the authors did demonstrate inhibition of the new targets on fluorometric analysis and also demonstrated interaction by gel shift - This is excellent

18. The title and the abstract could be better written and more clearly indicate the extent of the analyses performed and the discovery of alternate protease targets for PAI-1

19. Was the SARS CoV2 lung epithelial cell culture analysis performed in BSL3?

****Minor critiques****

1. Results section heading "SERPINS are differentially expressed individuals with COVID-19 and in response to respiratory virus infection in a model of the human airway epithelium."

The word in needs to be inserted between expressed and individuals

2. Significance:

Significance (Required)

Overall this is a very simple, although extensive and excellent, study analyzing a wide range of data from patients with bronchoalveolar lavage and epithelial cell samples, human epithelial cell cultures after infection with a range of respiratory viruses as well as the development of a 3D in silico analysis of potential protease and serpin interactions. These correlations between changes in serpin and protease expression with viral infections and potential new interactions for serpins with previously non identified proteases is of clear interest. This shows an excellent correlation but as with big data sets this does not provide a true cause and effect - rather providing new potential directions for analysis of these interactions in viral infections in lung epithelium and this is valuable as a basis for ongoing studies. Prior work evidencing other cross class serpin protease targets as well as limitations related to the analyses as discussed in the critiques above should be noted and the abstract and title could better describe and define the studies as performed.

3. How much time do you estimate the authors will need to complete the suggested revisions:

Estimated time to Complete Revisions (Required)

(Decision Recommendation)

Less than 1 month

4. Review Commons values the work of reviewers and encourages them to get credit for their work. Select 'Yes' below to register your reviewing activity at Web of Science Reviewer Recognition Service (formerly Publons); note that the content of your review will not be visible on Web of Science.

No

Review #2

1. Evidence, reproducibility and clarity:

Evidence, reproducibility and clarity (Required)

Summary:

Rodriguez Galvan et al. use a combined computational and functional approach to identify a novel target for the protease inhibitor, PAI-1 (SERPINE1), and show that exogenous PAI-1 can inhibit SARS-CoV-2 replication. They first use a COVID-19 dataset to identify SERPINS that are differentially expression in individuals with mild and severe COVID-19. They further use experimental infections of a model of human airway epithelium to identify SERPINS that are upregulated in response to several viruses, as well as treatment with interferon. Using this panel of SERPINS and a panel of host proteases, they use computational docking to predict SERPINS that may inhibit human proteases that may be relevant for viral infection. Using these predictions, they show that PAI-1 inhibits TMPRSS2 (previously shown) and CTSL (newly shown in this study), two proteases with relevance for SARS-CoV-2 infection. They finally show that extracellular addition of PAI-1 inhibits multicycle replication of SARS-CoV-2.

Major comments:

- 1) The rigor of the results presented in Figure 1 are unclear. For the COVID-19 analyses (Figure 1), only one dataset is used, and no statistical analyses are performed to determine to what degree any of the changes they observe are significant relative to variation in the dataset. This makes it difficult to determine how much can be extrapolated from these data.
- 2) Similarly, the qPCR data presented in Figure 2 are presented with no statistical analyses. Results should not only be presented with fold change but also p-values that are adjusted for multiple testing.

3) How is the dotted line drawn in Figure 3C and D? It would appear there is very little in terms of HADDOCK score to distinguish a predicted "binder" from "non-binder". Also, they later show that CTSB is non inhibited, and yet in Figure 3C it is below the dotted line. Can the authors more clearly delineate how one might use their dataset shown in Figure 3B to accurately predict targets of SERPINs?

4) Based on this, it would be preferable for the authors to tone down their claims about the broad applicability of this approach to predict SERPIN-protease interactions. It is true that they have used it to accurately predict PAI-1-CTSL interactions, but to make such a broad claim about the generalizable nature of this approach would require testing several more SERPIN-protease pairs (both binders and non-binders) to clearly define the scores and parameters that can be used to robustly predict interactions.

5) In Figure 3D, the authors mutate all eight modeled RCL residues to alanine to create a LOF mutant that has a higher HADDOCK score. Single residue mutations would be more convincing for their model, and would be more informative in terms of their predicted models of interactions.

6) Figures 4F and 4G are rather confusing. First, in Figure 4F, amount of PAI-1 in lane 1 is not the same as in the lanes with CTSL. The biggest concern with this is that there is a second, higher MW band that is present in lane 1 (also in Figure 4G lane 1) that runs near the band in lanes 2&3 that is marked as the PAI-1-CTSL complex. Although it does appear that the band in lane 1 and lanes 2&3 are slightly different sizes, it is hard to say that conclusively when the amounts of PAI-1 are different. Can the authors repeat this assay to load consistent amounts PAI-1 across all conditions and even potentially separate the top bands to more convincingly show that the band in lanes 2&3 is not in the PAI-1 alone control?

7) Further, in Figure 4G lanes 2-4, the PAI-1 band at ~38kDa is not present. Can the authors explain this?

8) The authors show that exogenous PAI-1 can inhibit SARS-CoV-2 in a multicycle infection in Figure 4H. However, this could be acting at multiple points during the viral infection cycle. A clearer virology experiment to support their model would be to perform single-cycle infections to show that the virus fails to productively infect the cell. For instance, have the authors attempted a high MOI, single-cycle infection to see whether they can detect uncleaved spike protein to show inhibition of cleavage? Or show that no early products of viral infection are produced? While this type of experiment is optional in that it is not required to support the claim that PAI-1 inhibits multicycle SARS-CoV-2 infection, it would support the conclusion that PAI-1 is inhibiting viral entry.

9) In Figure 4I, the authors claim that the addition of PAI-1 is inhibiting cleavage of the SARS-CoV-2 spike protein (S2) based on densitometry quantifications. However, it is unclear how the authors are normalizing their data, nor whether the experiments (and

therefore quantification) are from a single experiment or multiple replicates. Could the authors explain the quantification further and provide replicate information (including statistical support) if those experiments were performed? Further, to strengthen the conclusions of this data the authors should include additional controls. One would be to use trixplanin as they did in previous panels to show that PAI-1 is necessary. Further, if the authors generate mutant PAI-1 that is unable to inhibit TMPRSS2 (see comment 11 below), they could also use this as a control to show the necessity of functional PAI-1.

10) For Figures 4I-J, is it possible to also blot for S1 cleavage? If possible, this optional data would be helpful to understand whether the entire cleavage process is disrupted or only S2 to S2' especially given that visually it appears as if the full length is more depleted in the condition with PAI-1 suggesting that it is cleaving spike better into S1 and S2. Could also suggest that the dynamics of cleavage are shifted rather than impaired?

11) One (optional) way to extend these data and support their molecular model would be to mutate residues in PAI-1 that they predict are important for protease inhibition. As their source of PAI-1 currently is commercial, this would require purification of WT and variant PAI-1, which is clearly an undertaking. However, these data would strongly support their modeling and the importance of these residues in engaging with the proteases and springing the mousetrap for their in-vitro/in-vivo experiments (as suggested by data shown in Figure 3F and explained in text). Further, the authors can use these mutants to do some of the functional experiments in Figure 4 as a negative control, and potentially even separate the role of PAI-1 in inhibition of CTSL and TMPRSS2 in terms of SARS-CoV-2 inhibition.

Minor comments:

1) The authors speculate about SERPINA1 regulation during viral infection, suggesting an active process of "viral evasion". However, it would appear that even upon interferon treatment in Figure 2C, SERPINA1 expression is decreased. Based on that, the authors should soften their claims about the cause of downregulation of SERPINA1.

2) In Figure 2C, do the authors have an explanation or hypothesis for why SERPINE1 is less upregulated at 72hrs when compared to 24hr infection of SARS-CoV-2?

3) Can the authors demonstrate how the docking structure of the TMPRSS2 zymogen differs from the active version (especially zooming in on the interface of PAI-1 and the protease)? This could be supplemental data but can the authors show a panel like that in Figure 3F to show how the interface between PAI-1 and TMPRSS2 zymogen looks. Does the inactive TMPRSS2 not interface well with the RCL? Or what is leading to the decreased HADDOCK score?

4) In methods, uPA fluorometric protease assay information is missing. Please add this

information.

5) It is a bit confusing that Figure 4K is the quantification of assays shown in Figure 4A-C, rather than quantification of any of the intervening figure panels. It might be clearer to move this quantification next to 4A-C so that it is clearer.

6) In Figure 4H, the authors show that addition of recombinant PAI-1 decreases the number of SARS-CoV-2 nucleoprotein positive cells. Have the authors examined whether this decreases the viral titers as well?

7) As a supplemental figure, can the authors show a complex blot (similar to Figure 4F) for CTSL to show that it does not complex with PAI-1.

8) In the text, the authors suggest that PAI-1 inhibition of CTSL is surprising/novel. The authors should reconsider phrasing this since there are several other SERPINS that have been shown to inhibit other cathepsins, making this appear less surprising than the authors are suggesting.

2. Significance:

Significance (Required)

Assessment and impact:

This paper brings attention to the potential role of SERPINS in viral pathogenesis. The datasets shown in Figure 1 and 2, with the statistical caveats described above, are interesting demonstrations of the regulation of SERPINS during viral infection. In particular, the comparison of different viruses, and viruses compared to interferon alone, in Figure 2B is intriguing. These data are the strongest points of the paper.

The impact of the computational modeling is difficult to assess. While they have used this dataset to predict one novel interaction (CTSL) with PAI-1, the generalizable nature of this approach to broadly predict SERPIN-protease interactions is unclear since they have not tested or validated any other SERPIN-protease pairs. One major concern is the one raised in Major comments 3&4 above, which is that the score difference between a "non-binder" (CTSB) and a "binder" (uPa) is very small. It is exciting that they predicted CTSL as a target of PAI-1, but it is not obvious that this is a generalizable approach without further hypothesis testing.

The claim of novelty about TMPRSS2 is confusing. In their previous paper (reference 19) they show that PAI-1 inhibits TMPRSS2 activity. These data are clearly shown in Figure 4C & 4D of that paper and are summarized in their sentence in the discussion: "Here, we find three new PAI-1 protease targets: human tryptase (tryptase Clara; club cell secretory

protein), HAT, and TMPRSS2 ...". In this current paper, although they characterize the PAI-1-TMPRSS2 interaction in more detail than in their previous paper, they have truly only discovered one new target for PAI-1, which is CTSL.

Finally, the data on SARS-CoV-2 are intriguing and contribute to an emerging field on antiviral SERPINS. This reveals an additional virus that is inhibited by PAI-1, to add to their previous discoveries (reference 19) of influenza virus and Sendai virus inhibition by PAI-1. Future virology experiments, and experiments with mutants that ideally separate the ability of PAI-1 to inhibit TMPRSS2 versus CTSL, will further reveal the step of viral replication that is inhibited, and reveal the contribution of inhibition of TMPRSS2, CTSL, or any other PAI-1 targets, on SARS-CoV-2 replication.

Audience:

Overall, this paper will be interesting to a specialized audience that is interested in SERPIN function. The SERPIN expression data during viral infection, discovery of CTSL as a target of PAI-1, and evidence that PAI-1 can inhibit SARS-CoV-2 replication, will move that field forward.

Field of expertise:

Biochemistry, host-virus interactions

3. How much time do you estimate the authors will need to complete the suggested revisions:

Estimated time to Complete Revisions (Required)

(Decision Recommendation)

Between 1 and 3 months

No

Revision Plan

Manuscript number: RC-2024-02448

Corresponding author(s): Meike Dittmann

1. General Statements

We thank the reviewers for the overwhelmingly positive feedback on our initial submission.

Reviewer 1: "Overall this is a very simple, although extensive and excellent, study analyzing a wide range of data from (*sic*) patients with bronchoalveolar lavage and epithelial cell samples, human epithelial cell cultures after infection with a range of respiratory viruses as well as the development of a 3D *in silico* analysis of potential protease and serpin interactions."

Reviewer 2: "Overall, this paper will be interesting to a specialized audience that is interested in SERPIN function. The SERPIN expression data during viral infection, discovery of CTSL as a target of PAI-1, and evidence that PAI-1 can inhibit SARS-CoV-2 replication, will move that field forward."

*No new experiments were requested, but some were either suggested or explicitly marked optional. We thus focused the initial 4-week-revision on performing new experiments aimed to enhance our study's significance and impact by validating the heart of our study: the data from the *in-silico* docking screen.*

Reviewer 1: "Further analysis of the detected proteases that are reported here to bind to PAI-1 would be of great interest."

Reviewer 2: "It is exciting that they predicted CTSL as a target of PAI-1, but it is not obvious that this is a generalizable approach without further hypothesis testing."

*Thus, we performed additional protease activity assays to validate SERPIN-protease pairs from the *in-silico*-screen. The results elevate our study above the proof-of-principle state. Beyond their described roles in infectious disease, the two SERPINs that are now tested in more detail (SERPINB2, plasminogen activator inhibitor 2 and SERPINE1, plasminogen activator inhibitor 1) also play critical roles in cancer, neurodegeneration, aging, and cardiovascular disease. (Bouton et al., EMBO Mol Med 2023 Vol. 15 No. 6; Zhang et al., EMBO Mol Med 2023 Vol. 15 No. 9; Bode et al., EMBO Journal 1986 Vol. 5 No. 10; Uhl et al., EMBO Mol Med 2021 Vol. 13 No. 6). **Given these multifaceted roles, we anticipate that our discovery of new SERPIN-protease binders and non-binders will advance various areas of human disease driven by SERPIN biology.***

2. Description of the planned revisions

We believe that the planned revisions outlined below can be finalized within 1-2 months.

Reviewer 1: “Further analysis of the detected proteases that are reported here to bind to PAI-1 would be of great interest.”

At the time of the 30-day revision, recombinant SERPINB1 (LEI) and SERPING1 (C1-INH) were still backordered with an estimated shipping date the week of resubmission. Once delivered, we will perform protease activity assays with LEI or C1-INH and uPA, TMPRSS2, Cathepsin L, and Cathepsin B to bring up the number of validated SERPIN:protease interactions from 8 to 16.

Reviewer 2, major points:

9) Further, to strengthen the conclusions of this data the authors should include additional controls. One would be to use triplaxin as they did in previous panels to show that PAI-1 is necessary. Further, if the authors generate mutant PAI-1 that is unable to inhibit TMPRSS2 (see comment 11 below), they could also use this as a control to show the necessity of functional PAI-1.

We agree that these optional experiments would increase rigor. We generated plasmids containing mutated PAI-1 that we can use in spike cleavage assays as suggested and can perform this experiment.

We can unfortunately not use triplaxin on cells, as our preliminary data show that it is quite cytotoxic at the concentrations required to inhibit PAI-1.

10) For Figures 4I-J, is it possible to also blot for S1 cleavage? If possible, this optional data would be helpful to understand whether the entire cleavage process is disrupted or only S2 to S2' especially given that visually it appears as if the full length is more depleted in the condition with PAI-1 suggesting that it is cleaving spike better into S1 and S2. Could also suggest that the dynamics of cleavage are shifted rather than impaired?

S1 cleavage is shown indirectly in (now) Figure 5f,g – the main product of S1 cleavage is the fragment annotated as S2. Due to high levels of endogenous furin in BHK cells, this cleavage always occurs in this experimental setting. It is true that we have not shown the effects of PAI-1 inhibits on S1 cleavage– we can include that control in the above optional experiment (point 9). We do not expect PAI-1 to have an effect on S1 cleavage, as it is well-established that it does not inhibit furin.

Reviewer 2, minor points

7) As a supplemental figure, can the authors show a complex blot (similar to Figure 4F) for CTSB to show that it does not complex with PAI-1.

Purified active CTSB is not commercially available, but we can attempt to perform gel shift analysis on the samples from the in vitro protease assay. Due to the presence of proteinaceous substrate in these samples, we have previously observed lot of

Revision Plan

background on the gel, but we can attempt it and include it in a revised manuscript if reviewer/editor find it useful.

Revision Plan

3. Description of the revisions that have already been incorporated in the transferred manuscript

Reviewer 1:

Summary:

The authors do not reference the prior work which has examined cross class serpins, viral and mammalian, - this should be noted as alternative protease targets are known. *Thank you – please see our response in point 1 below.*

The bronchiolar lavage analysis is excellent but cannot differentiate epithelial cell and associated immune cells and their roles in the response.

We apologize for not making this clear – scRNAseq can indeed differentiate between different cell types using cell-type specific expression markers for each individual cell. This is how we were able to retrieve expression data specific for individual cell types. The reviewer is correct in that an expression analysis cannot show the role of individual cell types in the antiviral response. However, as epithelial cells are the primary cell type infected by SARS-CoV-2 gene expression patterns in these epithelial cells may show us cell-intrinsic effectors that are upregulated in response to viral infection. We now revised language in this paragraph to make this clearer (lines 156-162).

PAI-1 does not seem to be present in the bronchoalveolar lavage samples.

We do not know if PAI-1 is present, as we did not analyze protein levels in these samples. The gene expression data suggests that SERPINE1, the gene encoding PAI-1, is expressed at low levels in the epithelial cell subset at baseline, and expressed at slightly higher levels in individuals with severe COVID-19 (Figure 1c). This is consistent with previously published data on SERPINE1 gene expression upon viral infection (Dittmann et al., Cell, 2015).

Further discussion of prior work with cross class serpins and also the limitations of the in-silico analyses and the lavage specimens should be provided.

Prior work evidencing other cross class serpin protease targets as well as limitations related to the analyses as discussed in the critiques above should be noted and the abstract and title could better describe and define the studies as performed.

Thank you for raising these important points. For cross-class SERPINS, please see our response to point 1. The limitations of in silico analyses are discussed in-depth in a paragraph of the discussion (lines 608-631). We also discuss discrepancies observed between SERPIN expression in lavage specimens and in HAEC – please advise whether this is sufficient or needs bolstering (lines 546-564). We revised both title and abstract to better describe and define the studies as performed.

These correlations between changes in serpin and protease expression with viral infections and potential new interactions for serpins with previously non identified proteases is of clear interest. This shows an excellent correlation but as with big data sets this does not provide a true cause and effect - rather providing new potential

Revision Plan

directions for analysis of these interactions in viral infections in lung epithelium and this is valuable as a basis for ongoing studies.

We are in agreement with the lack of cause and effect –to our knowledge, we make no such claim from the gene expression data. We state that we used the expression data to guide the selection of SERPINs for our in-silico screen (lines 317-319). We then validated select data from our in-silico screen in vitro, which provides true cause and effect (Figures 4 and 5).

Major points:

- 1) Cross class serpin interactions are known and have been reported for at least two viral serpins Serp-1 and CrmA - both of which bind cysteine proteases as well as serine proteases as well as the mammalian SCCA serpins.

Thank you for bringing these two examples to our attention – we added them to the discussion (lines 648-652). We now also emphasized throughout the manuscript that the novelty of our findings is in PAI-1 cross-class inhibition, specifically, which has not been previously reported despite PAI-1 being an extremely well-studied SERPIN.

We also would like to mention that in our opinion the scientific advance provided by our in-silico screen is not limited to the identification of new PAI-1 targets, but also provides a birds-eye view on SERPIN selectivity in a specific proteolytic landscape. For example, to our knowledge, it was unknown that SERPINB1 is promiscuous and that SERPINC1 is more selective, which our docking predicted. It was unknown that most TMPRSSs are unlikely SERPIN targets and that those that are SERPIN targets need to be in their active state to bind. The unsupervised clustering in Figure 4b (both on the SERPIN and on the protease side) predicts such unrecognized patterns in SERPIN selectivity.

- 2) The protease targets are reported to vary when interacting with glycosaminoglycans such as heparan sulfate - PAI-1 inhibits thrombin in the presence of heparin - thus while a canonical serpin suicide inhibition is considered specific - it can vary. This is noted in the discussion

Yes, we agree (lines 608-610).

- 3) What is the potential impact of the noted interactions of PAI-1 with other proteases such as cathepsin - PAI-1 is considered to have predominately extracellular functions, but prior work indicates internalization of PAI-1 when bound to the uPA/uPAR complex with alterations in intra cellular activation

This is correct and PAI-1 internalization is cited and mentioned in discussion (lines 620-624). We now also added data on SARS-CoV-2 variant Omicron BA.1, which predominantly uses CTSL for maturation, and we show is also inhibited by PAI-1 (new Figure 5).

Revision Plan

- 4) This is supported by basic in vivo and in vitro serpin and protease interactions that are demonstrated confirming in silico analyses, eg. gel shift analyses or even Mass spectrometry analysis particularly for PAI-1
Yes, this is the data shown in Figure 4. We now also added protease activity assays for other SERPIN-protease pairs, thereby elevating our study above the proof-of-principle state. This was also a suggestion raised by reviewer 2.
- 5) Per the authors "To date, three SERPINS have been studied in the context of innate antiviral defense: PAI- 1 (encoded by SERPINE1) against influenza viruses encoding hemagglutinin H1 and SARS-CoV-2, by impeding the proteolytic maturation of H1 or spike, respectively^{19,20}; alpha-1-antitrypsin (encoded by SERPINA1) and antithrombin (encoded by SERPINC1) against SARS-CoV-2, likely through the inhibition of TMPRSS2, by reducing maturation of spike, although direct inhibition of TMPRSS2 by either SERPIN was not shown". This is partially complete however other serpins such as C1Inh and one virus derived serpin that have been analyzed for efficacy in treating SARS
Thank you for mentioning this, we added the information to the introduction (lines 106-111).
- 6) While TMPRSS2 is indeed a serine protease - Beneficial effects of some serpins may be due to modulation of the immune response as opposed to selective anti-viral responses. The immune / cytokine storm and coagulopathies (with clotting and even hemorrhage) seen in the excess inflammatory response that causes respiratory vascular leak and severe viral sepsis. PAI-1 targets tPA and uPA - uPA has marked proinflammatory actions when bound to the uPA receptor (uPAR) and can activate growth factors and MMPs which can enhance immune cell invasion - PAI-1 binds to the uPA / uPAR complex which can thus also alter inflammatory cell responses and cell activation when internalized.
Thank you for bringing up this point. The role of SERPINS in inflammation and anti-viral immune responses is indeed well-established. While our study focuses on cell-intrinsic antiviral roles of SERPINS by shutting down pro-viral proteases, which is much less established, we now added this to the results section for clarification (line 153-156).
- 7) The RCL does in general incorporate P4 to P4' but can vary from this specific P4 to P4' sequence
Yes, we agree.
- 8) How accurately does in silico protease serpin analysis predict real interactions? - this should be discussed as HADDOCK may have some limitations - This is outside my field of expertise
We added an in-depth paragraph on how HADDOCK operates to the results section to help readers not familiar with the technique (lines 248-290). We discuss the limitations of HADDOCK in depth in the discussion section (lines 608-631)– please advise whether this needs additional information.

Revision Plan

*We argue that, with the limitations stated in the discussion, our in-silico method predicts interactions well, as shown by the correct prediction of known binders and non-binders, as well as of new binders (PAI-1 to *active* TMPRSS2 and CTSL) and a new non-binder (CTSB).*

As with any screening method, results require validation via another method, which we performed for select SERPINs and proteases. In fact, the revised manuscript now features in vitro validation of 8 SERPIN-protease pairs (Figure 4a, b), with 8 additional planned (see “planned revisions” section).

- 9) The data from a published study examining bronchoalveolar lavage fluid single cell transcriptional analysis from patients with and without COVID - mild and severe - and with comparison to patients without COVID does demonstrate altered protease and serpin activity - but does not indicate specific interactions

We agree with this statement partially. We disagree in that the data does not demonstrate altered protease and SERPIN activity; it instead demonstrates changes in gene expression levels. We agree in that this does indeed not indicate specific interactions.

- 10) What is the significance for changes in gene expression in epithelial cells versus macrophage T and B cells looks - This looks like a small change like a small change in the mean values Figure 1b

We performed additional statistical analyses on the Figure 1 data – please refer to Reviewer 2 point 1.

- 12) Of interest - is the bronchoalveolar lavage fluid likely to contain both epithelial cells as well as immune response macrophage, T cells and NK cells etc - one assumes single cells were identified and isolated- Is this defined?

Apologies if this was unclear. Yes, the BALF contains all of these cell types. We now added some sentences to the results section explaining scRNAseq and analyses in more detail (lines 147-162).

- 13) The known previously reported target proteases for PAI-1 should be noted

Agreed; it is noted in the results section where we first speak about PAI-1 target specificity (line 379-382).

SERPINE1 is not noted in figure 1 - this is PAI-1 - but is seen in the HAEC infection model data

SERPINE1 is indeed not significantly upregulated in Figure 1, but is significantly upregulated in HAEC upon infection with Reovirus and parainfluenzavirus 3, and upon IFN stimulation (new Supplemental Tables S1 and S2). The possible reasons for discrepancies between the BALF and HAEC data are discussed in lines 546-564.

- 14) “To overcome this limitation, we developed a computational method to predict 3D interactions between SERPINs and proteases, simulating the binding process depicted in Supplemental Figure 1a. Specifically, we employed High Ambiguity

Revision Plan

Driven protein- protein Docking (HADDOCK), a tool that predicts complex structures, integrating experimental and computational data^{35,36}. " This analysis looks to be extensive however this is a correlation - not a true analysis of cause and effect.

We agree on the first point – to our knowledge, our study provides the most extensive SERPIN target discovery process (testing 480 SERPIN-protease interactions).

We disagree on the point that our results provide a mere correlation. If you will, we performed a computer-modeled interaction experiment that yields predicted binding energies between each SERPIN with each tested protease. We added a paragraph on how HADDOCK operates to the results section to help readers unfamiliar with the technique. As with any screening method, results need to be validated with another method, which we did for select SERPINs and proteases (Figure 4a, b).

This does however have the potential to identify significant interactions

We certainly agree on this point.

In future it might be of interest to assess PAI-1 given to infected cultures to assess viral replication and titers or perhaps examine a knock out cell model?

We did exactly the former in Figure 4 (now 5).

- 17) As PAI-1 was identified as having new cathepsin protease binding in addition to TMPRSS2 - the authors did demonstrate inhibition of the new targets on fluorometric analysis and also demonstrated interaction by gel shift - This is excellent

Thank you.

- 18) The title and the abstract could be better written and more clearly indicate the extent of the analyses performed and the discovery of alternate protease targets for PAI-1

We modified both title and abstract.

- 19) Was the SARS CoV2 lung epithelial cell culture analysis performed in BSL3?

Yes. All SARS-CoV-2 infection experiments were performed in a BSL3 environment. We added this information throughout the Methods section, and also generated a new Methods section on Biohazards (lines 779-797).

Minor critiques

- 1) Results section heading "SERPINs are differentially expressed individuals with COVID-19 and in response to respiratory virus infection in a model of the human airway epithelium." The word in needs to be inserted between expressed and individuals

Thank you for catching this – we fixed the sentence (lines 128-129).

Revision Plan

Reviewer 2:

Major points:

1) The rigor of the results presented in Figure 1 are unclear. For the COVID-19 analyses (Figure 1), only one dataset is used, and no statistical analyses are performed to determine to what degree any of the changes they observe are significant relative to variation in the dataset. This makes it difficult to determine how much can be extrapolated from these data.

We agree that performing statistics on the BALF dataset would be ideal. However, the BALF contains only two non-infected individuals (intubated gun-shot victims), limiting our possibilities for statistical analysis.

For Figure 1b, we overcame this limitation by adding statistical analysis of upregulated expression values between cell types (i.e. by analyzing differences of upregulation of given SERPIN in epithelial cells compared to macrophages; Supplemental Table S1).

We also performed statistical analysis on upregulation for individual SERPINs compared to housekeeping gene B2M (Supplemental Table S1). This revealed that SERPINs statistically significantly upregulated in severe COVID-19 in most cell types, including epithelial cells, in which SERPIN function has not been broadly studied.

Upregulation was not statistically significant in mild COVID-19 samples, likely due to the n=3 (as compared to n=6 in the severe COVID-19 group).

As for analysis of Figure 1c, we could theoretically perform analysis of differential levels between mild and severe COVID-19, but this is not the question we are trying to answer. The question is whether epithelial cells express SERPINs and proteases, and whether there is an upregulation of either in infected individuals. We now state the limitation of lacking statistical power in the figure legend and the text (lines 176-177).

2) Similarly, the qPCR data presented in Figure 2 are presented with no statistical analyses. Results should not only be presented with fold change but also p-values that are adjusted for multiple testing.

We now present p-values in Supplemental Table S2. Of note, data obtained with the experimental system of polarized airway epithelial cultures, differentiated over several weeks, tends to be noisier than that obtained with cell lines. Despite this, a number of SERPINs reach statistical significance.

3) How is the dotted line drawn in Figure 3C and D? It would appear there is very little in terms of HADDOCK score to distinguish a predicted "binder" from "non-binder". Also, they later show that CTSB is non inhibited, and yet in Figure 3C it is below the dotted line. Can the authors more clearly delineate how one might use their dataset shown in Figure 3B to accurately predict targets of SERPINs?

This is a valid point. We added a more in-depth description to the results section on how we define "binders" and "non-binders" (lines 324-331 and Figure 3 legend). We added raw data graph with the thresholds in Supplemental Figure 3d. We further added and defined a threshold line to the PAI-1:CTSS graph (Figure 3c). It is now evident that CTSL, A, F, K score as high-confidence "binders", while CTSB and others do not. We also added the normalization process and the visual assessment of top-scoring

Revision Plan

complexes to the in silico docking screen schematic in Figure 3a and the respective figure legend to guide readers.

4) Based on this, it would be preferable for the authors to tone down their claims about the broad applicability of this approach to predict SERPIN-protease interactions. It is true that they have used it to accurately predict PAI-1-CTSL interactions, but to make such a broad claim about the generalizable nature of this approach would require testing several more SERPIN-protease pairs (both binders and non-binders) to clearly define the scores and parameters that can be used to robustly predict interactions.

We thank the reviewer for this criticism. We now address this in the text as outlined in our response to point 3 above. As with any screening method, the results require to be validated via an alternative approach, which we did in the initial submission for TMPRSS2 and CTSL as binders and CTSB as a non-binder. The revised manuscript now features additional in vitro validation of binders and non-binders for a total of 8 SERPIN-protease combinations (Figure 4a, b), which were all correctly predicted by our in-silico method. Two more SERPINs will be added in the final revision (see "planned revisions" section). Our study provides ample data for future studies validating additional predicted pairs and characterizing their biological function, in infectious disease and beyond.

5) In Figure 3D, the authors mutate all eight modeled RCL residues to alanine to create a LOF mutant that has a higher HADDOCK score. Single residue mutations would be more convincing for their model, and would be more informative in terms of their predicted models of interactions.

We now performed the docking with the single mutant, please see new Figure 3c.

7) Further, in Figure 4G lanes 2-4, the PAI-1 band at ~38kDa is not present. Can the authors explain this?

This is likely because CTSL digests PAI-1 working at its optimum pH (aka "the protease wins"). We removed the panel from the manuscript.

9) In Figure 4I, the authors claim that the addition of PAI-1 is inhibiting cleavage of the SARS-CoV-2 spike protein (S2) based on densitometry quantifications. However, it is unclear how the authors are normalizing their data, nor whether the experiments (and therefore quantification) are from a single experiment or multiple replicates. Could the authors explain the quantification further and provide replicate information (including statistical support) if those experiments were performed?

Thank you for pointing this out. An explanation has now been added to the Figure 5 legend.

Minor comments:

1) The authors speculate about SERPINA1 regulation during viral infection, suggesting an active process of "viral evasion". However, it would appear that even upon interferon

Revision Plan

treatment in Figure 2C, SERPINA1 expression is decreased. Based on that, the authors should soften their claims about the cause of downregulation of SERPINA1.

Thank you for pointing this out – we softened the language on this point (lines 225-228).

2) In Figure 2C, do the authors have an explanation or hypothesis for why SERPINE1 is less upregulated at 72hrs when compared to 24hr infection of SARS-CoV-2?

We can only speculate on this point. It is possible that one or several of the SARS-CoV-2 accessory proteins modulate SERPINE1 expression in a time-dependent manner.

3) Can the authors demonstrate how the docking structure of the TMPRSS2 zymogen differs from the active version (especially zooming in on the interface of PAI-1 and the protease)? This could be supplemental data but can the authors show a panel like that in Figure 3F to show how the interface between PAI-1 and TMPRSS2 zymogen looks. Does the inactive TMPRSS2 not interface well with the RCL? Or what is leading to the decreased HADDOCK score?

We added an extensive paragraph on how HADDOCK operates to the results section to introduce how the HADDOCK score is calculated (lines 248-290). We also added a visual of the top-scoring docking complex of PAI-1 and the TMPRSS2 zymogen (Figure 3d) to illustrate the differences in binding.

4) In methods, uPA fluorometric protease assay information is missing. Please add this information.

Thank you for catching this – we added the information (line 890).

5) It is a bit confusing that Figure 4K is the quantification of assays shown in Figure 4A-C, rather than quantification of any of the intervening figure panels. It might be clearer to move this quantification next to 4A-C so that it is clearer.

Thank you for the suggestion – Figure 4 has been restructured.

6) In Figure 4H, the authors show that addition of recombinant PAI-1 decreases the number of SARS-CoV-2 nucleoprotein positive cells. Have the authors examined whether this decreases the viral titers as well?

Yes, this is now part of the (new) Figure 5.

8) In the text, the authors suggest that PAI-1 inhibition of CTSL is surprising/novel. The authors should reconsider phrasing this since there are several other SERPINS that have been shown to inhibit other cathepsins, making this appear less surprising than the authors are suggesting.

Thank you for pointing this out. We have now clarified throughout the manuscript that while other SERPINS indeed are known to inhibit cathepsins, this had not been previously shown for the extremely well-studied SERPIN PAI-1 with over 15,000 pubmed entries. We also added the implications of this PAI-1-specific finding to the discussion section.

Revision Plan

Significance:

The claim of novelty about TMPRSS2 is confusing. In their previous paper (reference 19) they show that PAI-1 inhibits TMPRSS2 activity. These data are clearly shown in Figure 4C & 4D of that paper and are summarized in their sentence in the discussion: "Here, we find three new PAI-1 protease targets: human tryptase (tryptase Clara; club cell secretory protein), HAT, and TMPRSS2 ...". In this current paper, although they characterize the PAI-1-TMPRSS2 interaction in more detail than in their previous paper, they have truly only discovered one new target for PAI-1, which is CTSL.

Thank you for pointing this out – we softened language on the novelty of TMPRSS2 as a PAI-1 target throughout the manuscript. We further clarify that the novelty is that TMPRSS2 has to be in its active form to be inhibited by PAI-1, which was previously unknown (lines 392, 432). The revised manuscript now also provides validation of total 8 predicted binders and non-binders for 2 (Figure 4 b,c), with 8 more pending (see “planned revisions” section). As those two (future four) SERPINs have various roles in cancer, cardiovascular disease, neurodegeneration, and immunity, our findings have impact beyond their antiviral potential, thereby increasing the overall significance of the manuscript.

4. Description of analyses that authors prefer not to carry out

Reviewer 1 major point:

11. The more common names for the SERPINS as detected in COVID alveolar lavage samples would be helpful in figure 1 - and specifically labelling PAI-1 as this is a focus for this study - together with the known SERPIN nomenclature or under abbreviations - For example SERPINB2 is PAI-2 and SERPING1 is C1INH and SERPINA1 is alpha 1 antitrypsin

Thank you for this suggestion. We tried to keep the SERPIN nomenclature consistent throughout the manuscript, in that the SERPIN genes are referred to by their gene name (i.e., SERPINE1), while the proteins are referred to by their protein name (i.e., PAI-1). Editor and/or Reviewer 1, please advise whether this is acceptable or should be changed. We also added the protein corresponding names in the figure legend.

16. Why does supplemental figure 2 show SERPINB1 and not PAI-1.

We performed this computer-modeled experiment (docking SERPINs to known binders and known non-binders) for each SERPIN tested in the study. This was needed to obtain thresholds to define likely binders and likely non-binders. We chose to show SERPINB1 in this supplemental figure because it is well-described with regards to binders and non-binders (the latter, as “negative result”, is not always published for a given SERPIN). We also did not want to narrow the study immediately to PAI-1, as we believe our screen is a generalizable method and our findings are valid beyond PAI-1. We can easily show any other SERPIN here - editor and/or Reviewer 1, please advise.

Reviewer 2 major point:

6) Figures 4F and 4G are rather confusing. First, in Figure 4F, amount of PAI-1 in lane 1 is not the same as in the lanes with CTSL. The biggest concern with this is that there is a second, higher MW band that is present in lane 1 (also in Figure 4G lane 1) that runs near the band in lanes 2&3 that is marked as the PAI-1-CTSL complex. Although it does appear that the band in lane 1 and lanes 2&3 are slightly different sizes, it is hard to say that conclusively when the amounts of PAI-1 are different. Can the authors repeat this assay to load consistent amounts PAI-1 across all conditions and even potentially separate the top bands to more convincingly show that the band in lanes 2&3 is not in the PAI-1 alone control?

The upper band is an impurity that disappears upon addition of a protease to the reaction. We confirmed that this band is neither PAI-1 nor CTSL via western blot with PAI-1- or CTSL-specific antibodies. Should reviewer 2 and/or the editor feel that we should repeat the experiment with more loading in the first lane, we can certainly do so. Please advise.

8) The authors show that exogenous PAI-1 can inhibit SARS-CoV-2 in a multicycle infection in Figure 4H. However, this could be acting at multiple points during the viral infection cycle. A clearer virology experiment to support their model would be to perform single-cycle infections to show that the virus fails to productively infect the cell. For instance, have the authors attempted a high MOI, single-cycle infection to see whether they can detect uncleaved spike protein to show inhibition of cleavage? Or show that no early products of viral infection are produced? While this type of experiment is optional in that it is not required to support the claim that PAI-1 inhibits multicycle SARS-CoV-2 infection, it would support the conclusion that PAI-1 is inhibiting viral entry.

We agree with the reviewer. We did expand on the virology by using now two strains of SARS-CoV-2 with different proteolytic needs, ancestral WA-1 and Omicron BA.1. We also performed titer analysis (all in Figure 5).

However, the other suggested experiments would represent a substantial amount of work in a BSL3 environment. We thus would prefer not do these experiments (as the reviewer states, it is optional), and instead tone down the manuscript to make clear we make no claims on viral entry.

Reviewer 2 minor point:

11) One (optional) way to extend these data and support their molecular model would be to mutate residues in PAI-1 that they predict are important for protease inhibition. As their source of PAI-1 currently is commercial, this would require purification of WT and variant PAI-1, which is clearly an undertaking. However, these data would strongly support their modeling and the importance of these residues in engaging with the proteases and springing the mousetrap for their in-vitro/in-vivo experiments (as suggested by data shown in Figure 3F and explained in text). Further, the authors can use these mutants to do some of the functional experiments in Figure 4 as a negative

Revision Plan

control, and potentially even separate the role of PAI-1 in inhibition of CTSL and TMPRSS2 in terms of SARS-CoV-2 inhibition.

We agree that these (optional) experiments would be beautiful and are indeed part of future studies on the subject. We feel that they exceed the scope of this current manuscript.

Dear Dr. Dittmann,

Thank you for submitting your Review Commons manuscript "In-silico docking with serine protease inhibitors (SERPINs) identifies alternate protease targets with significance for SARS-CoV-2" to The EMBO Journal together with the revision plan. I have now read your manuscript, the reviewer comments and your response to them. I am sorry to say that we have decided not to pursue the publication of the manuscript at The EMBO Journal.

I appreciate that your study reports on increased expression of various SERPINs upon infection with four respiratory viruses in human airway epithelial cultures and in bronchioalveolar lavage samples from COVID-19 patients. Further in-silico docking screen of SERPIN/protease pairs using HADDOCK identifies potential novel targets of specific SERPINs, including cathepsin L as a target for SERPINE1/PAI-1. Finally, addition of recombinant PAI-1 inhibits replication of the ancestral SARS-CoV-2 WA-1 strain (mainly relies on the known SERPINE1/PAI-1 target TMPRSS2) and the variant Omicron BA.1, which mainly requires cathepsin L for maturation.

We appreciate the high quality of the study and realise that the findings will be of interest to the more immediate research field, as also indicated in the reviewer comments from Review Commons. However, we also noted that the role of SERPINE1/PAI-1 in suppression of SARS-CoV-2 infection via TMPRSS2 inhibition has been shown in previous work. Furthermore, SERPINs have been shown to inhibit activity of various cathepsins in the context of inflammation, and the relevance of the new SERPIN-protease pairs reported here for viral infection suppression remains to be shown. Therefore, I am afraid we concluded that your study is not a sufficiently strong candidate for publication in The EMBO Journal.

Thank you for giving us the opportunity to consider this manuscript. I regret that I could not offer more positive news at this time, and I sincerely hope for rapid publication of your manuscript at another Review Commons partner journal.

Yours sincerely,

Ieva Gailite

Rev_Com_number: RC-2024-02448

New_manu_number: EMBOJ-2024-118196-T

Corr_author: Dittmann

Title: In-silico docking with serine protease inhibitors (SERPINs) identifies alternate protease targets with significance for SARS-CoV-2

Dear Dr. Gailite,

thank you so much for your detailed response.

I wanted to investigate whether you would be open to reconsider if we added another angle to the manuscript - the finding that we identified the first direct-antiviral SERPIN in our in-silico screen. We had initially planned to make this a separate manuscript, but it would actually fit the current manuscript well and would also address the reviewers' criticism of novelty and significance. I attached a preliminary Figure 6 to this email for your perusal. A revised title could be "In-silico docking with full-length 3D serine protease inhibitors (SERPINs) identifies alternate protease targets of human and viral origin".

Based on the longstanding history of SERPINs at The EMBO Journal I think that our manuscript would be a great fit. Please let me know if there is a possibility to move forward.

Best regards,

Meike

Dear Meike,

Thank you for contacting me for a further discussion of your manuscript. I find the new data interesting, and I think that their inclusion would make the study a more suitable candidate for a resource manuscript at our journal. It would be great if you could supplement the recombinant PAI-2 treatment shown in Figure 6f with some evidence that the inhibitory effect is linked to AVP degradation, I think such evidence would strengthen the study significantly.

Best wishes,

leva

*We thank the reviewers for the overwhelmingly positive feedback on our initial submission. **No new experiments were requested**, but some were either suggested or explicitly marked optional. In addition to performing suggested experiments, we focused on adding new data aimed to further **enhance our study's novelty and impact**.*

Reviewer #1

Summary:

The finding that PAI-1 has cross class serpin functions is of definite interest given the roles of PAI-1 in regulation of physiological processes as well as in driving pathology. PAI-1 is generally considered to be a key regulator of thrombolysis and thus an effect on other pathways and even intracellular pathways is of interest. Examining airway epithelial proteases and serpins is of definite interest in respiratory viral infections. Broadening the targets for serpins is also of very definite interest. This study ranges from an overview of prior published work on bronchoalveolar lavage samples and serpin expression, a tissue culture analysis of lung epithelial cells and expression of proteases and serpins is assessed. In addition selective changes in serpin expression and protease targets are assessed by in silico analysis as well as proof of concept via Western blot and fluorometric analysis. This is an extensive study and of definite interest.

There are some limitations as with any study, albeit the study overall is excellent. The authors do not reference the prior work which has examined cross class serpins, viral and mammalian, - this should be noted as alternative protease targets are known. *Thank you – please see our response in point 1 below.* They do mention the modulation of protease targets by glycosaminoglycans in the discussion. Further, serpins are inhibitors, thus while the RCL provides a target for a protease, but the response may not be fully selective in vivo as the protease has to be present and active to complete the serpin protease interaction. The bronchiolar lavage analysis is excellent but cannot differentiate epithelial cell and associated immune cells and their roles in the response. *We apologize for not making this clear – scRNAseq can indeed differentiate between different cell types using cell-type specific expression markers for each individual cell. This is how we were able to retrieve expression data specific for individual cell types. The reviewer is correct in that an expression analysis cannot show the role of individual cell types in the antiviral response. However, as epithelial cells are the primary cell type infected by SARS-CoV-2 gene expression patterns in these epithelial cells may show us cell-intrinsic effectors that are upregulated in response to viral infection. We now revised language in this paragraph to make this clearer (line 126).* PAI-1 does not seem to be present in the bronchoalveolar lavage samples. *We do not know if PAI-1 is present, as we did not analyze protein levels in these samples. The gene expression data suggests that SERPINE1, the gene encoding PAI-1, is expressed at low levels in the epithelial cell subset at baseline, and expressed at slightly higher levels in individuals with severe COVID-19 (Figure 1c). This is consistent with previously published data on SERPINE1 gene expression upon viral infection (Dittmann et al., Cell, 2015).* Further discussion of prior work with cross class serpins and also the limitations of the in silico analyses and the lavage specimens should be provided. *For cross-class SERPINS, please see our response to point 1. The limitations of in silico analyses are discussed in-depth in a paragraph of the discussion (line 543). We also discuss discrepancies observed between SERPIN expression in lavage specimens and in HAEC – please advise whether this is sufficient or needs bolstering (line 600).* Further analysis of the detected proteases that are reported here to bind to PAI-1 would be of great interest. *We now provide a total of 8 in vitro validation experiments for SERPIN-host protease binders and non-binders, showing that the in-silico analysis is a generalizable approach. The two SERPINS (SERPINB2, plasminogen activator inhibitor 2 and SERPINE1, plasminogen activator inhibitor*

1) that are now tested for host protease inhibition with four proteases also play critical roles in cancer, neurodegeneration, aging, and cardiovascular disease. Given these multifaceted roles, we anticipate that our discovery of new SERPIN-host protease binders and non-binders will advance various areas of human disease driven by SERPIN biology (Bouton et al., *EMBO Mol Med* 2023 Vol. 15 No. 6; Zhang et al., *EMBO Mol Med* 2023 Vol. 15 No. 9; Bode et al., *EMBO Journal* 1986 Vol. 5 No. 10; Uhl et al., *EMBO Mol Med* 2021 Vol. 13 No. 6). In addition, we now provide experimental evidence of the **first direct-acting antiviral SERPIN**. Using our in-silico method, we identify PAI-2 as an inhibitor of adenovirus protease, which we validate via protease assays and viral replication experiments. The data from the bronchoalveolar lavage is published. That is correct, and we here mine this valuable dataset for SERPIN or protease expression.

Overall this is a very simple, although extensive and excellent, study analyzing a wide range of data from patients with bronchoalveolar lavage and epithelial cell samples, human epithelial cell cultures after infection with a range of respiratory viruses as well as the development of a 3D in silico analysis of potential protease and serpin interactions. These correlations between changes in serpin and protease expression with viral infections and potential new interactions for serpins with previously non identified proteases is of clear interest. This shows an excellent correlation but as with big data sets this does not provide a true cause and effect - rather providing new potential directions for analysis of these interactions in viral infections in lung epithelium and this is valuable as a basis for ongoing studies. *We are in agreement with the lack of cause and effect –to our knowledge, we make no such claim from the gene expression data. We state that we used the expression data to guide the selection of SERPINs for our in-silico screen (line 240). We then validated select data from our in-silico screen in vitro, which provides true cause and effect (Figures 4 and 5).* Prior work evidencing other cross class serpin protease targets as well as limitations related to the analyses as discussed in the critiques above should be noted and the abstract and title could better describe and define the studies as performed. *For cross-class SERPINs, please see our response to point 1. The limitations of in silico analyses are discussed in-depth in a paragraph of the discussion (line 601). We also discuss discrepancies observed between SERPIN expression in lavage specimens and in HAEC – please advise whether this is sufficient or needs bolstering (line 543). We revised both title and abstract to better describe and define the studies as performed.*

Critique, Major

1) Cross class serpin interactions are known and have been reported for at least two viral serpins Serp-1 and CrmA - both of which bind cysteine proteases as well as serine proteases as well as the mammalian SCCA serpins.

Thank you for bringing these two examples to our attention – we added them to the discussion (line 647). We now also emphasized throughout the manuscript that the novelty of our findings is in PAI-1 cross-class inhibition, specifically, which has not been previously reported despite PAI-1 being an extremely well-studied SERPIN.

2) The protease targets are reported to vary when interacting with glycosaminoglycans such as heparan sulfate - PAI-1 inhibits thrombin in the presence of heparin - thus while a canonical serpin suicide inhibition is considered specific - it can vary. This is noted in the discussion *Yes, we agree.*

3) What is the potential impact of the noted interactions of PAI-1 with other proteases such as cathepsin - PAI-1 is considered to have predominately extracellular functions, but prior work

indicates internalization of PAI-1 when bound to the uPA/uPAR complex with alterations in intra cellular activation.

PAI-1 internalization is cited and mentioned in discussion (line 618). We also discuss the potential impact of PAI-1-cathepsin interaction in cancer and viral infections. Finally, we added data on SARS-CoV-2 variant Omicron BA.1, which predominantly uses CTSL for maturation, and we show is also inhibited by PAI-1 (Fig 5).

4) This is supported by basic in vivo and in vitro serpin and protease interactions that are demonstrated confirming in silico analyses, eg. gel shift analyses or even Mass spectrometry analysis particularly for PAI-1

Yes, this is the data shown in Figure 4. We now also added protease activity assays to achieve a total of 8 SERPIN-host protease pairs, thereby elevating our study above the proof-of-principle state. This was also a suggestion raised by reviewer 2.

5) Per the authors "To date, three SERPINS have been studied in the context of innate antiviral defense: PAI- 1 (encoded by SERPINE1) against influenza viruses encoding hemagglutinin H1 and SARS-CoV-2, by impeding the proteolytic maturation of H1 or spike, respectively^{19,20}; alpha-1-antitrypsin (encoded by SERPINA1) and antithrombin (encoded by SERPINC1) against SARS-CoV-2, likely through the inhibition of TMPRSS2, by reducing maturation of spike, although direct inhibition of TMPRSS2 by either SERPIN was not shown". This is partially complete however other serpins such as C1Inh and one virus derived serpin that have been analyzed for efficacy in treating SARS

Thank you for mentioning this, we added the information to the introduction (line 96)

6) While TMPRSS2 is indeed a serine protease - Beneficial effects of some serpins may be due to modulation of the immune response as opposed to selective anti-viral responses. The immune / cytokine storm and coagulopathies (with clotting and even hemorrhage) seen in the excess inflammatory response that causes respiratory vascular leak and severe viral sepsis. PAI-1 targets tPA and uPA - uPA has marked proinflammatory actions when bound to the uPA receptor (uPAR) and can activate growth factors and MMPs which can enhance immune cell invasion - PAI-1 binds to the uPA / uPAR complex which can thus also alter inflammatory cell responses and cell activation when internalized.

Thank you for bringing up this point. The role of SERPINS in inflammation and anti-viral immune responses is indeed well-established. While our study focuses on cell-intrinsic antiviral roles of SERPINS by shutting down pro-viral proteases, which is much less established, we now added this to the results section for clarification.

7) The RCL does in general incorporate P4 to P4' but can vary from this specific P4 to P4' sequence

We agree.

8) How accurately does in silico protease serpin analysis predict real interactions? - this should be discussed as HADDOCK may have some limitations - This is outside my field of expertise

We added an in-depth paragraph on how HADDOCK operates to the results section to help readers not familiar with the technique (line 852). We discuss the limitations of HADDOCK in depth in the discussion section (line 601)– please advise whether this needs additional information.

*With the limitations stated in the discussion, we argue that our in-silico method predicts interactions well, as shown by the correct prediction of known binders and non-binders, as well as of new binders (PAI-1 to *active* TMPRSS2 and CTSL) and a new non-binder (CTSB). As with any screening method, results require validation via another method, which we performed*

for select SERPINS and proteases. In fact, the revised manuscript now features in vitro validation of 8 SERPIN-host protease pairs (Figure 4a, b). In addition, we now show inhibition of a viral protease, AVP, by PAI-2, which was predicted by our in-silico screen and is also validated by in vitro and viral infection experiments.

9) The data from a published study examining bronchoalveolar lavage fluid single cell transcriptional analysis from patients with and without COVID - mild and severe - and with comparison to patients without COVID does demonstrate altered protease and serpin activity - but does not indicate specific interactions

We agree with this statement partially. We disagree in that the data does not demonstrate altered protease and SERPIN activity; it instead demonstrates changes in gene expression levels. We agree in that this does indeed not indicate specific interactions.

10) What is the significance for changes in gene expression in epithelial cells versus macrophage T and B cells looks - This looks like a small change like a small change in the mean values Figure 1b

*We performed additional statistical analyses on the Figure 1 data – please refer to **Reviewer 2 point 1.***

11) The more common names for the SERPINS as detected in COVID alveolar lavage samples would be helpful in figure 1 - and specifically labelling PAI-1 as this is a focus for this study - together with the known SERPIN nomenclature or under abbreviations - For example SERPINB2 is PAI-2 and SERPING1 is C1INH and SERPINA1 is alpha 1 antitrypsin

Thank you for this suggestion. We tried to keep the SERPIN nomenclature consistent throughout the manuscript, in that the SERPIN genes are referred to by their gene name (i.e., SERPINE1), while the proteins are referred to by their protein name (i.e., PAI-1). Editor and/or Reviewer 1, please advise whether this is acceptable or should be changed. We also added the protein corresponding names in the figure legend.

12) Of interest - is the bronchoalveolar lavage fluid likely to contain both epithelial cells as well as immune response macrophage, T cells and NK cells etc - one assumes single cells were identified and isolated- Is this defined?

Apologies if this was unclear. Yes, the BALF contains all of these cell types. We now added some sentences to the results section explaining scRNAseq and analyses in more detail (line 126).

13) The known previously reported target proteases for PAI-1 should be noted

*Agreed; it is noted in the results section where we first speak about PAI-1 target specificity (line 388-392). We also attached a heatmap visualizing the state of previous knowledge on SERPIN-protease interactions, in the same format as the heatmap in Figure 3 (Figure EV3d). We further denote with an * which interactions and non-interactions have been in vitro validated in this current study (Figure 3 heatmap). This will allow the reader to easily access the contribution of this manuscript to the SERPIN field in general.*

14) SERPINE1 is not noted in figure 1 - this is PAI-1 - but is seen in the HAEC infection model data

SERPINE1 is indeed not significantly upregulated in Figure 1, but is significantly upregulated in HAEC upon infection with Reovirus and parainfluenzavirus 3, and upon IFN stimulation (Table 1 and Table 2). The possible reasons for discrepancies between the BALF and HAEC data are discussed in line 543

15) "To overcome this limitation, we developed a computational method to predict 3D interactions between SERPINs and proteases, simulating the binding process depicted in Supplemental Figure 1a. Specifically, we employed High Ambiguity Driven protein- protein Docking (HADDOCK), a tool that predicts complex structures, integrating experimental and computational data^{35,36}." This analysis looks to be extensive however this is a correlation - not a true analysis of cause and effect

We agree on the first point – to our knowledge, our study provides the most extensive SERPIN target discovery process (testing 480 SERPIN-protease interactions). We disagree on the point that our results provide a mere correlation. If you will, we performed a computer-modeled interaction experiment that yields predicted binding energies between each SERPIN with each tested protease. We added a paragraph on how HADDOCK operates to the results section to help readers unfamiliar with the technique. As with any screening method, results need to be validated with another method, which we now present for 2 SERPINs with 5 proteases (Figure 4a, b, and new Figure 6).

This does however have the potential to identify significant interactions.

We certainly agree on this point. We now show for the first time inhibition of a viral protease, AVP, by PAI-2, which was predicted by our in-silico screen and is validated by in vitro and viral infection experiments.

In future it might be of interest to assess PAI-1 given to infected cultures to assess viral replication and titers or perhaps examine a knock out cell model?

We did exactly the former in Figure 4 (now Figure 5).

16) Why does supplemental figure 2 show SERPINB1 and not PAI-1

We performed this computer-modeled experiment (docking SERPINs to known binders and known non-binders) for each SERPIN tested in the study. This was needed to obtain thresholds to define likely binders and likely non-binders. We chose to show SERPINB1 in this supplemental figure because it is well-described with regards to binders and non-binders (the latter, as “negative result”, is not always published for a given SERPIN). We also did not want to narrow the study immediately to PAI-1, as we believe our screen is a generalizable method and our findings are valid beyond PAI-1. We can easily show any other SERPIN here - editor and/or Reviewer 1, please advise.

17) As PAI-1 was identified as having new cathepsin protease binding in addition to TMPRSS2 - the authors did demonstrate inhibition of the new targets on fluorometric analysis and also demonstrated interaction by gel shift - This is excellent

Thank you.

18) The title and the abstract could be better written and more clearly indicate the extent of the analyses performed and the discovery of alternate protease targets for PAI-1

We modified both title and abstract.

19) Was the SARS CoV2 lung epithelial cell culture analysis performed in BSL3?

Yes. All SARS-CoV-2 infection experiments were performed in a BSL3 environment. We added this information throughout the Methods section, and also generated a new Methods section on Biohazards (line 816)

Minor critiques:

1) Results section heading "SERPINs are differentially expressed individuals with COVID-19 and in response to respiratory virus infection in a model of the human airway epithelium. "The word in needs to be inserted between expressed and individuals

Thank you for catching this – we fixed the sentence (Line 142).

Significance:

Overall this is a very simple, although extensive and excellent, study analyzing a wide range of data from patients with bronchoalveolar lavage and epithelial cell samples, human epithelial cell cultures after infection with a range of respiratory viruses as well as the development of a 3D in silico analysis of potential protease and serpin interactions. These correlations between changes in serpin and protease expression with viral infections and potential new interactions for serpins with previously non identified proteases is of clear interest. This shows an excellent correlation but as with big data sets this does not provide a true cause and effect - rather providing new potential directions for analysis of these interactions in viral infections in lung epithelium and this is valuable as a basis for ongoing studies.

We are in agreement with the lack of cause and effect –to our knowledge, we make no such claim from the gene expression data. We state that we used the expression data to guide the selection of SERPINs for our in-silico screen (line 239). We then validated select data from our in-silico screen in vitro, which provides true cause and effect (Figures 4 and 5).

Prior work evidencing other cross class serpin protease targets as well as limitations related to the analyses as discussed in the critiques above should be noted and the abstract and title could better describe and define the studies as performed.

For cross-class SERPINs, please see our response to point 1. The limitations of in silico analyses are discussed in-depth in a paragraph of the discussion as mentioned above. We also discuss discrepancies observed between SERPIN expression in lavage specimens and in HAEC – please advise whether this is sufficient or needs bolstering (line 543). We revised both title and abstract to better describe and define the studies as performed.

Reviewer #2:

Summary:

Rodriguez Galvan et al. use a combined computational and functional approach to identify a novel target for the protease inhibitor, PAI-1 (SERPINE1), and show that exogenous PAI-1 can inhibit SARS-CoV-2 replication. They first use a COVID-19 dataset to identify SERPINS that are differentially expression in individuals with mild and severe COVID-19. They further use experimental infections of a model of human airway epithelium to identify SERPINS that are upregulated in response to several viruses, as well as treatment with interferon. Using this panel of SERPINS and a panel of host proteases, they use computational docking to predict SERPINS that may inhibit human proteases that may be relevant for viral infection. Using these predictions, they show that PAI-1 inhibits TMPRSS2 (previously shown) and CTSL (newly shown in this study), two proteases with relevance for SARS-CoV-2 infection. They finally show that extracellular addition of PAI-1 inhibits multicycle replication of SARS-CoV-2.

Major comments:

1) The rigor of the results presented in Figure 1 are unclear. For the COVID-19 analyses (Figure 1), only one dataset is used, and no statistical analyses are performed to determine to what degree any of the changes they observe are significant relative to variation in the dataset. This makes it difficult to determine how much can be extrapolated from these data.

We agree that performing statistics on the BALF dataset would be ideal. However, the BALF contains only two non-infected individuals (intubated gun-shot victims), limiting our possibilities for statistical analysis.

For Figure 1b, we overcame this limitation by adding statistical analysis of upregulated expression values between cell types (i.e. by analyzing differences of upregulation of given SERPIN in epithelial cells compared to macrophages; Table 1). We also performed statistical analysis on upregulation for individual SERPINS compared to housekeeping gene B2M (Table 1). This revealed that SERPINS statistically significantly upregulated in severe COVID-19 in most cell types, including epithelial cells, in which SERPIN function has not been broadly studied. Upregulation was not statistically significant in mild COVID-19 samples, likely due to the n=3 (as compared to n=6 in the severe COVID-19 group).

As for analysis of Figure 1c, we could theoretically perform analysis of differential levels between mild and severe COVID-19, but this is not the question we are trying to answer. The question is whether epithelial cells express SERPINS and proteases, and whether there is an upregulation of either in infected individuals. We now state the limitation of lacking statistical power in the figure legend and the text.

2) Similarly, the qPCR data presented in Figure 2 are presented with no statistical analyses. Results should not only be presented with fold change but also p-values that are adjusted for multiple testing.

We now present p-values in Table 2. Of note, data obtained with the experimental system of polarized airway epithelial cultures, differentiated over several weeks, tends to be noisier than that obtained with cell lines. Despite this, a number of SERPINS reach statistical significance.

3) How is the dotted line drawn in Figure 3C and D? It would appear there is very little in terms of HADDOCK score to distinguish a predicted "binder" from "non-binder". Also, they later show that CTSL is non inhibited, and yet in Figure 3C it is below the dotted line. Can the authors more clearly delineate how one might use their dataset shown in Figure 3B to accurately predict targets of SERPINS?

This is a valid point. We added a more in-depth description to the results section on how we define “binders” and “non-binders” (Figure 3 legend). We added raw data graph with the thresholds in Figure EV3d. We further added and defined a threshold line to the PAI-1:CTSLs graph (Figure 3c). It is now evident that CTSL, A, F, K score as high-confidence “binders”, while CTSLB and others do not. We also added the normalization process and the visual assessment of top-scoring complexes to the *in silico* docking screen schematic in Figure 3a and the respective figure legend to guide readers.

4) Based on this, it would be preferable for the authors to tone down their claims about the broad applicability of this approach to predict SERPIN-protease interactions. It is true that they have used it to accurately predict PAI-1-CTSL interactions, but to make such a broad claim about the generalizable nature of this approach would require testing several more SERPIN-protease pairs (both binders and non-binders) to clearly define the scores and parameters that can be used to robustly predict interactions.

We thank the reviewer for this criticism. We now address this in the text as outlined in our response to point 3 above. As with any screening method, the results require to be validated via an alternative approach, which we did in the initial submission for TMPRSS2 and CTSL as binders and CTSLB as a non-binder. The revised manuscript now features additional *in vitro* validation of binders and non-binders for a total of 8 SERPIN-host protease combinations (Figure 4a, b), which were all correctly predicted by our *in-silico* method. We also added a heatmap visualizing the state of previous knowledge on SERPIN-protease interactions, in the same format as the heatmap in Figure 3 (new Figure EV3d).

We further denote with an * which interactions and non-interactions have been *in vitro* validated in this current study (Figure 3 heatmap). This will easily depict the contributions of this manuscript. We now also show for the first time inhibition of a viral protease, AVP, by PAI-2, which was predicted by our *in-silico* screen and is validated by *in vitro* and viral infection experiments. Our study provides ample data for future studies validating additional predicted pairs and characterizing their biological function, in infectious disease and beyond.

5) In Figure 3D, the authors mutate all eight modeled RCL residues to alanine to create a LOF mutant that has a higher HADDOCK score. Single residue mutations would be more convincing for their model, and would be more informative in terms of their predicted models of interactions.

We now performed the docking with the single mutant, please see new Figure 3c (PAI-1 R346A).

6) Figures 4F and 4G are rather confusing. First, in Figure 4F, amount of PAI-1 in lane 1 is not the same as in the lanes with CTSL. The biggest concern with this is that there is a second, higher MW band that is present in lane 1 (also in Figure 4G lane 1) that runs near the band in lanes 2&3 that is marked as the PAI-1-CTSL complex. Although it does appear that the band in lane 1 and lanes 2&3 are slightly different sizes, it is hard to say that conclusively when the amounts of PAI-1 are different. Can the authors repeat this assay to load consistent amounts PAI-1 across all conditions and even potentially separate the top bands to more convincingly show that the band in lanes 2&3 is not in the PAI-1 alone control?

The upper band is an impurity that disappears upon addition of a protease to the reaction. We confirmed that this band is neither PAI-1 nor CTSL via western blot with PAI-1- or CTSL-specific antibodies. Should reviewer 2 and/or the editor feel that we should repeat the experiment with more loading in the first lane, we can certainly do so. Please advise.

7) Further, in Figure 4G lanes 2-4, the PAI-1 band at ~38kDa is not present. Can the authors explain this?

This is likely because CTSL digests PAI-1 working at its optimum pH (aka "the protease wins"). We removed the panel from the manuscript.

8) The authors show that exogenous PAI-1 can inhibit SARS-CoV-2 in a multicycle infection in Figure 4H. However, this could be acting at multiple points during the viral infection cycle. A clearer virology experiment to support their model would be to perform single-cycle infections to show that the virus fails to productively infect the cell. For instance, have the authors attempted a high MOI, single-cycle infection to see whether they can detect uncleaved spike protein to show inhibition of cleavage? Or show that no early products of viral infection are produced? While this type of experiment is optional in that it is not required to support the claim that PAI-1 inhibits multicycle SARS-CoV-2 infection, it would support the conclusion that PAI-1 is inhibiting viral entry.

We agree with the reviewer. We did expand on the virology by using now two strains of SARS-CoV-2 with different proteolytic needs, ancestral WA-1 and Omicron BA.1. We also performed titer analysis (Figure 5).

However, the other suggested experiments would represent a substantial amount of work in a BSL3 environment. We thus would prefer not do these experiments (as the reviewer states, it is optional), and instead tone down the manuscript to make clear we make no claims on viral entry.

9) In Figure 4I, the authors claim that the addition of PAI-1 is inhibiting cleavage of the SARS-CoV-2 spike protein (S2) based on densitometry quantifications. However, it is unclear how the authors are normalizing their data, nor whether the experiments (and therefore quantification) are from a single experiment or multiple replicates. Could the authors explain the quantification further and provide replicate information (including statistical support) if those experiments were performed?

Thank you for pointing this out. An explanation has now been added to the (now) Figure 5 legend.

Further, to strengthen the conclusions of this data the authors should include additional controls. One would be to use trixplanin as they did in previous panels to show that PAI-1 is necessary. Further, if the authors generate mutant PAI-1 that is unable to inhibit TMPRSS2 (see comment 11 below), they could also use this as a control to show the necessity of functional PAI-1.

We agree that these optional experiments would increase rigor. We now performed spike assays with mutated LoF PAI-1 (below), but we don't believe that the blot contributes significantly to the manuscript. Please let us know if you feel differently and we can add it. We

can unfortunately not use triplaxinin on cells, as our preliminary data show that it is quite cytotoxic at the concentrations required to inhibit PAI-1.

10) For Figures 4I-J, is it possible to also blot for S1 cleavage? If possible, this optional data would be helpful to understand whether the entire cleavage process is disrupted or only S2 to

S2' especially given that visually it appears as if the full length is more depleted in the condition with PAI-1 suggesting that it is cleaving spike better into S1 and S2. Could also suggest that the dynamics of cleavage are shifted rather than impaired?

S1 cleavage is shown indirectly in (now) Figure 5f,g – the main product of S1 cleavage is the fragment annotated as S2. Due to high levels of endogenous furin in BHK cells, this cleavage always occurs in this experimental setting. It is true that we have not shown the effects of PAI-1 inhibits on S1 cleavage– we have included the control in the experiment mentioned above (point 9). We do not expect PAI-1 to have an effect on S1 cleavage, as it is well-established that it does not inhibit furin.

11) One (optional) way to extend these data and support their molecular model would be to mutate residues in PAI-1 that they predict are important for protease inhibition. As their source of PAI-1 currently is commercial, this would require purification of WT and variant PAI-1, which is clearly an undertaking. However, these data would strongly support their modeling and the importance of these residues in engaging with the proteases and springing the mousetrap for their in-vitro/in-vivo experiments (as suggested by data shown in Figure 3F and explained in text). Further, the authors can use these mutants to do some of the functional experiments in Figure 4 as a negative control, and potentially even separate the role of PAI-1 in inhibition of CTSL and TMPRSS2 in terms of SARS-CoV-2 inhibition.

We agree that these (optional) experiments would be beautiful and are indeed part of future studies on the subject. We feel that they exceed the scope of this current manuscript.

Minor comments:

1) The authors speculate about SERPINA1 regulation during viral infection, suggesting an active process of "viral evasion". However, it would appear that even upon interferon treatment in Figure 2C, SERPINA1 expression is decreased. Based on that, the authors should soften their claims about the cause of downregulation of SERPINA1.

Thank you for pointing this out – we softened the language (line 156)

2) In Figure 2C, do the authors have an explanation or hypothesis for why SERPINE1 is less upregulated at 72hrs when compared to 24hr infection of SARS-CoV-2?

We can only speculate on this point. It is possible that one or several of the SARS-CoV-2 accessory proteins modulate SERPINE1 expression in a time-dependent manner.

3) Can the authors demonstrate how the docking structure of the TMPRSS2 zymogen differs from the active version (especially zooming in on the interface of PAI-1 and the protease)? This could be supplemental data but can the authors show a panel like that in Figure 3F to show how the interface between PAI-1 and TMPRSS2 zymogen looks. Does the inactive TMPRSS2 not interface well with the RCL? Or what is leading to the decreased HADDOCK score?

We added an extensive paragraph on how HADDOCK operates to the results section to introduce how the HADDOCK score is calculated (line 212). We also added a visual of the top-scoring docking complex of PAI-1 and the TMPRSS2 zymogen (Figure 3d) to illustrate the predicted differences in binding.

4) In methods, uPA fluorometric protease assay information is missing. Please add this information.

Thank you for catching this – we added the information.

5) It is a bit confusing that Figure 4K is the quantification of assays shown in Figure 4A-C, rather than quantification of any of the intervening figure panels. It might be clearer to move this quantification next to 4A-C so that it is clearer.

Thank you for the suggestion – Figure 4 has been restructured.

6) In Figure 4H, the authors show that addition of recombinant PAI-1 decreases the number of SARS-CoV-2 nucleoprotein positive cells. Have the authors examined whether this decreases the viral titers as well?

Yes, this is now part of the (new) Figure 5.

7) As a supplemental figure, can the authors show a complex blot (similar to Figure 4F) for CTSB to show that it does not complex with PAI-1.

Purified active CTSB is not commercially available, but we attempted to perform gel shift analysis on the samples from the in vitro protease assay. Due to the presence of proteinaceous substrate in these samples, we unfortunately observed a lot of background on the gel and did thus not deem it a useful addition to the manuscript.

8) In the text, the authors suggest that PAI-1 inhibition of CTSL is surprising/novel. The authors should reconsider phrasing this since there are several other SERPINS that have been shown to inhibit other cathepsins, making this appear less surprising than the authors are suggesting.

*Thank you for pointing this out. We have now clarified throughout the manuscript that while other SERPINS indeed are known to inhibit cathepsins, this had not been previously shown for the extremely well-studied SERPIN PAI-1 with over 15,000 pubmed entries. We also added the implications of this PAI-1-specific finding to the discussion section. We also added a heatmap visualizing the state of previous knowledge on SERPIN-protease interactions, in the same format as the heatmap in Figure 3 (Figure EV3d). We further denote with an * which interactions and non-interactions have been in vitro validated in this current study (Figure 3b heatmap). This will easily depict the contributions of this manuscript.*

Significance:

This paper brings attention to the potential role of SERPINS in viral pathogenesis. The datasets shown in Figure 1 and 2, with the statistical caveats described above, are interesting demonstrations of the regulation of SERPINS during viral infection. In particular, the comparison of different viruses, and viruses compared to interferon alone, in Figure 2B is intriguing. These data are the strongest points of the paper.

The impact of the computational modeling is difficult to assess. While they have used this dataset to predict one novel interaction (CTSL) with PAI-1, the generalizable nature of this approach to broadly predict SERPIN-protease interactions is unclear since they have not tested or validated any other SERPIN-protease pairs. One major concern is the one raised in Major comments 3&4 above, which is that the score difference between a "non-binder" (CTSB) and a "binder" (uPa) is very small. It is exciting that they predicted CTSL as a target of PAI-1, but it is not obvious that this is a generalizable approach without further hypothesis testing.

We hope that our better explanation of the method, including calculation of thresholds for high-confidence binders and high-confidence non-binders, and the in vitro validation of a total of 8 SERPIN-protease pairs with different specificities alleviates these concerns.

The claim of novelty about TMPRSS2 is confusing. In their previous paper (reference 19) they show that PAI-1 inhibits TMPRSS2 activity. These data are clearly shown in Figure 4C & 4D of

that paper and are summarized in their sentence in the discussion: "Here, we find three new PAI-1 protease targets: human tryptase (tryptase Clara; club cell secretory protein), HAT, and TMPRSS2 ...". In this current paper, although they characterize the PAI-1-TMPRSS2 interaction in more detail than in their previous paper, they have truly only discovered one new target for PAI-1, which is CTSL.

Thank you for pointing this out – we softened language on the novelty of TMPRSS2 as a PAI-1 target throughout the manuscript. We further clarify that the novelty is that TMPRSS2 has to be in its active form to be inhibited by PAI-1, which was previously unknown (line 278). Furthermore, this is the first kinetic biochemical evidence showing that PAI-1 directly inhibits TMPRSS2, rather than TMPRSS2-mediated HA cleavage (like in our previous manuscript). The revised manuscript now also provides validation of total of 8 predicted binders and non-binders for 4 SERPINs (Figure 4 b,c), As those SERPINs have various roles in cancer, cardiovascular disease, neurodegeneration, and immunity, our findings have impact beyond their antiviral potential, thereby increasing the overall significance of the manuscript.

To further enhance significance, we now also show for the first time inhibition of a viral protease, AVP, by PAI-2, which was predicted by our in-silico screen and is validated by in vitro and viral infection experiments.

Finally, the data on SARS-CoV-2 are intriguing and contribute to an emerging field on antiviral SERPINs. This reveals an additional virus that is inhibited by PAI-1, to add to their previous discoveries (reference 19) of influenza virus and Sendai virus inhibition by PAI-1.

We thank the reviewer for this comment. We would like to add that we believe the scientific advance provided by our in-silico screen is not limited to the identification of new PAI-1 targets, but also provides a birds-eye view on SERPIN selectivity in a specific proteolytic landscape. For example, to our knowledge, it was unknown that SERPINB1 is promiscuous and that SERPINC1 is more selective, which our docking predicted. It was unknown that most TMPRSSs are unlikely SERPIN targets and that those that are SERPIN targets need to be in their active state to bind. The unsupervised clustering in Figure 3b (both on the SERPIN and on the protease side) predicts such previously unrecognized patterns in SERPIN selectivity.

Future virology experiments, and experiments with mutants that ideally separate the ability of PAI-1 to inhibit TMPRSS2 versus CTSL, will further reveal the step of viral replication that is inhibited, and reveal the contribution of inhibition of TMPRSS2, CTSL, or any other PAI-1 targets, on SARS-CoV-2 replication.

We agree with those suggestions.

In addition, while viral SERPINs targeting host proteases have been described, a host SERPIN targeting a viral protease had to date not been found. We now close that gap by showing that PAI-2 inhibits Adenovirus protease (AVP) – identified by in-silico docking, and validated in vitro and in viral infection experiments.

Audience:

Overall, this paper will be interesting to a specialized audience that is interested in SERPIN function. The SERPIN expression data during viral infection, discovery of CTSL as a target of PAI-1, and evidence that PAI-1 can inhibit SARS-CoV-2 replication, will move that field forward.

*We thank the reviewer for this praise. We would like to add that the revised manuscript contains experimental evidence of the **first direct-acting antiviral SERPIN**.*

Finally, we would like to add that the two SERPINs that are now tested in detail with 4 different host proteases play critical roles in cancer, neurodegeneration, aging, and cardiovascular disease. Given these multifaceted roles, we anticipate that our discovery of new SERPIN-host protease binders and non-binders will advance various areas of human disease driven by SERPIN biology (Bouton et al., EMBO Mol Med 2023 Vol. 15 No. 6; Zhang et al., EMBO Mol

Med 2023 Vol. 15 No. 9; Bode et al., EMBO Journal 1986 Vol. 5 No. 10; Uhl et al., EMBO Mol Med 2021 Vol. 13 No. 6).

Correspondence with the Editor:

*I wanted to investigate whether you would be open to reconsider if we added another angle to the manuscript – the finding that we identified the first direct-antiviral SERPIN in our in-silico screen. We had initially planned to make this a separate manuscript, but it would actually fit the current manuscript well and would also address the reviewers' criticism of novelty and significance. I attached a preliminary Figure 6 to this email for your perusal. A revised title could be **“In-silico docking with full-length 3D serine protease inhibitors (SERPINs) identifies alternate protease targets of human and viral origin”**.*

I find the new data interesting, and I think that their inclusion would make the study a more suitable candidate for a resource manuscript at our journal.

We would like to thank the Editor for her guidance.

The revised Figure 6 contains in silico data showing the predicted fit of AVP with PAI-2 and controls. In vitro assays with PAI-2 and active AVP (AVP-pVlc-DNA, from cracked Adenovirus particles), prove that PAI-2 inhibits AVP activity. We further show that PAI-2 binds AVP using two different assays – by mixing commercially available recombinant AVP (which is inactive due to the absence of co-factors) with PAI-2 and performing pulldown experiments (Figure EV5), and by mixing active AVP (AVP-pVlc-DNA, from cracked Adenovirus particles) with PAI-2 and detecting high-molecular weight complexes. (Fig 6f, 6g)

It would be great if you could supplement the recombinant PAI-2 treatment shown in Figure 6f with some evidence that the inhibitory effect is linked to AVP degradation, I think such evidence would strengthen the study significantly. I indeed think it would be great to show experimentally that PAI-2 reduces adenovirus infectivity due to AVP inhibition.

We to date do not know whether PAI-2 leads to AVP degradation – this would be the topic of future studies. Our in vitro results show, however, that PAI-2 inhibits AVP proteolytic activity (Figure 6e and 6f). To link this to an antiviral phenotype, we now provide multi-cycle Adenovirus growth kinetics, which show that expression of PAI-2 significantly reduces the production of infectious viral progeny. Further, we assess specific infectivity of these progeny particles, and find that expression of PAI-2 reduces their infectivity (Fig 6i). This shows that not the quantity of the particles is changed, but their quality, which is exactly what we would expect with AVP inhibition by PAI-2. In agreement with that data, our immunofluorescence images point towards that in PAI-2 expressing cells after multicycle infection of AdV, progeny virions are unable to escape endolysosomal pathways. Moreover, when knocking down PAI-2 in 293T cells we see an increase in % infected cells per well (Fig 6j).

Dear Meike,

Thank you for submitting a revised version of your Review Commons manuscript. The study has now been seen by both original referees, who appreciate the added information, but also find that several of their initial points were not sufficiently addressed or clarified. Therefore, I would like to invite you to address the remaining referee comments either by textual changes or additional experimentation (reviewer #1, major points 4,5 and minor points regarding data). Regarding point 17 by reviewer #2, As previously discussed, from the editorial point of view it is important to include PAI-2 data in the manuscript. Regarding point 3 by reviewer #1, I would be happy to discuss whether addition of further controls to substantiate this interesting finding is feasible.

We generally allow three months as standard revision time. Should you foresee a problem in meeting this deadline, please let us know in advance to discuss an extension.

As a matter of policy, competing manuscripts published during this period will not negatively impact on our assessment of the conceptual advance presented by your study. However, please contact me as soon as possible upon publication of any related work to discuss the appropriate course of action.

When preparing your letter of response to the referees' comments, please bear in mind that this will form part of the Review Process File and will therefore be available online to the community. For more details on our Transparent Editorial Process, please visit our website: <https://www.embopress.org/page/journal/14602075/authorguide#transparentprocess>. Please also see the attached instructions for further guidelines on preparation of the revised manuscript.

Please feel free to contact me if have any questions regarding this final revision. Thank you again for giving us the chance to consider your manuscript for The EMBO Journal. I look forward to receiving the revised version.

With best regards,

Ieva

Ieva Gailite, PhD
Senior Scientific Editor
The EMBO Journal
Meyerohofstrasse 1
D-69117 Heidelberg
Tel: +4962218891309
i.gailite@embojournal.org

We realize that it is difficult to revise to a specific deadline. In the interest of protecting the conceptual advance provided by the work, we recommend a revision within 3 months (23rd Feb 2025). Please discuss the revision progress ahead of this time with the editor if you require more time to complete the revisions.

Referee #1:

Summary:

Rodriguez Galvan et al. use a combined computational and functional approach to identify a new host target for the protease inhibitor, PAI-1 (SERPINE1), and show that exogenous PAI-1 can inhibit SARS-CoV-2 replication. They also identify that PAI-2 (SERPINB2) can inhibit adenovirus protease and adenovirus replication. They first use a COVID-19 dataset to identify SERPINS that are differentially expression in individuals with mild and severe COVID-19. They further use experimental infections of a model of human airway epithelium to identify SERPINS that are upregulated in response to several viruses, as well as treatment with interferon. Using this panel of SERPINS and a panel of host proteases, they use computational docking to predict SERPINS that may inhibit human proteases that may be relevant for viral infection. Using these predictions, they show that PAI-1 inhibits TMPRSS2 (previously shown) and CTSL (newly shown in this study), two proteases with relevance for SARS-CoV-2 infection. They show that extracellular addition of PAI-1 inhibits multicycle replication of SARS-CoV-2. In a second aspect of their work, they computationally predict and validate that PAI-2 inhibits the protease activity of adenovirus protease. They further show that cellular overexpression can inhibit adenovirus replication whereas knockdown of endogenous PAI-2 can increase the number of adenovirus-positive cells in culture.

This work is a substantial advance in the field. The data on SERPIN regulation during viral infection and host protease inhibition by SERPINS was already going to be an important addition to the field. The new data on adenovirus protease inhibition makes this manuscript even more interesting and exciting. The functional conclusions are for the most part well supported by data. Below are a few comments, both major and minor, that could be addressed with additional data. The majority of the comments below, however, are in regards to strengthening the manuscript by filling in critical missing information, softening some conclusions (mostly regarding the generalizability of the computational method), and cleaning up errors in the manuscript.

Major Comments

- 1) The generalizability of the computational modeling is still not full supported. This is not a problem that needs to be addressed with data necessarily, but rather a less black-and-white interpretation of their data. In response to our previous comments, the authors have now included definitions of likely binder and likely non-binder based on the new normalization methods utilized, and have provided additional test cases. However, it is still unclear, as a reader, how one would use HADDOCK to reliably predict SERPIN-protease interactions. For instance, beginning on line 600, the text states that HADDOCK modeling correctly identified CTSL as a binder and CTSB as a non-binder. However, the HADDOCK scores for CTSB are little different than uPA, which is a known binder but still above their dotted threshold line in Figure 3c. As a reader, how would I interpret the uPA data? Or what would I predict for CTSO, CTSG, etc? It would appear that the HADDOCK data can make nice predictions for new interactions such as PAI-2 and adenovirus protease, but there needs to be more discussion of the limitations of the predictions in the 'grey area' that is occupied by both binders and non-binders (presumably falling approximately between the two dotted lines in Figure 3c).
- 2) Likewise, the authors present a black-and-white interpretation of the interaction predictions with regards to 3D fit and specific sequence recognition. On line 329, they state "3D fit, rather than the recognition sequence, is crucial for SERPIN target protease recognition". This is demonstrably false based on their HADDOCK models and functional data that show that changing the recognition sequence impacts SERPIN target protease recognition. Their data do show that 3D recognition is important, and that should be rightly highlighted, but a more nuanced interpretation about the importance of both of these types of molecular interactions is warranted.
- 3) The last figure panel (EV5f) and interpretations are entirely too preliminary and speculative. Using singular microscopy fields, the authors suggest that PAI-2 expressing cells may cause AdV to become trapped in the endosome. While it is true that there is colocalization of the AdV capsid proteins with endolysosomal markers, there are multiple instances of colocalization in the

empty vector cells. Further, to truly say that the particles are trapped would require live imaging that tracks these particles through time since the particles are in the endosome at some point during their lifecycle. Overall, these data are too preliminary and require several more controls to suggest this mechanism, even with their caveat that these are preliminary. It would be better to remove this from the manuscript as it weakens the overall rigor of the otherwise well-supported conclusions.

4) The blots in Figure 4c-e are not properly controlled and are therefore difficult to interpret. We previously raised a concern about Figure 4e, to which the authors responded that they know that the band that runs near their presumed SERPIN-protease complex is not PAI-1 or CTSL. It would be useful to see the data that support their statement. But the fact remains that in Figure 4d and 4e, the appearance of an upper band that the authors suggest is SERPIN-protease complex is only in lanes that contain more PAI-1 than the control lanes (compare lanes 1 and 4 in Figure 4d and lanes 1 and 2 in Figure 4e). Even in Figure 4c, the lanes (1 versus 4) are marked as having the same PAI-1, but the abundance by gel is visually quite different. In answer to their response to our previous concern, I would prefer they re-run these experiments with the proper controls if they wish to include these data to complement the already convincing functional data about SERPIN-mediated protease inhibition.

5) Was PAI-1, or any other SERPIN, tested against adenovirus protease? The interpretation that PAI-2 is a specific inhibitor of adenovirus protease would be strengthened by testing at least one other SERPIN.

6) There are quite a few claims of "firsts" throughout this manuscript. I do not know how EMBO considers these, but most journals discourage this in the way that discourage "novel" and "new". The manuscript is strong without making primacy claims about things like "the first comprehensive exploration of SERPIN targets ... (Lines 643-645)", which is subject to interpretation of the word "comprehensive" anyway.

7) Overall, there are many more mistakes in this manuscript than one would normally expect for a revised manuscript. Many of those are listed below. Individually, each is a minor comment, but together they become a major comment.

Minor Comments regarding data:

1) In response to author's response to major comment 10, while we do agree that the S2 is indirectly showing S1 cleavage, the intensity of the S2 is unreliable for quantifying how much of the original cleavage is occurring since it is subsequently cleaved. For the purposes of clarity of why the upper FL band is decreased in the PAI-1 condition, can the authors provide a loading control if the FL band difference is explained by loading difference?

2) Please include a similar graph as depicted in Figure 3C for PAI-2 since data is shown for it in Figure 4. This can easily be a supplement but would be helpful for readers to draw conclusions on the utility of HADDOCK predictions for these interactions, which is relevant for major comment 1 above.

3) Line 392, the authors state spike is not detectable from supernatant without showing any data to support that. It would be better to either include those data or omit this sentence.

4) For figure EV5b, is there a reason why the protease band is not shown in this gel images? It would help the reader to interpret the data in much the same way that all other gel images in the paper show both SERPIN and protease.

5) Similarly, in figure 6G, the cropping of the gel is too close to the bands of interest, please provide a blot that is not cropped as much.

6) Do the authors have titrating or genome copy measurements that correspond to with data in figure 6J? This would strengthen claims about utility for adenovirus production, since it is not clear how much of an impact siSERPINB2 has on adenovirus production practically.

Minor Comments regarding text and figures:

1) In line 151, the authors say with seeming certainty that, because SERPINS are upregulated in epithelial cells, they are involved in cell-intrinsic antiviral response. The authors should tone down the claim and write it as a speculative statement since not all SERPINS that are upregulated are examined in this manuscript. Similarly, line 173 makes a similar claim about the importance of the SERPINS produced in the airway during SARS-CoV-2 infection, tone down the claims as well. As a suggestion, the language in line 184 seems more appropriate.

2) In line 488, the authors say cracked but earlier in the text they say sonicated Adv particles. If they are the same, they should use consistent terminology throughout.

3) In Figure 1 and 2, it is strange to use ** for $p < 0.05$ as this is not the standard, nor is it what they use elsewhere in the paper.

4) The data in Figure 6i are confusing in many ways. First, how is it possible to have a specific infectivity, which is described as infectious particles/genome copy, that is greater than 1 (e.g. 2^{10} in Figure 6i)? I understand that the math from data shown in EV5c&d gives that answer, but it seems difficult to have more infectious particles than genomes. Second, why are there three data points in figure 6i and EV5c, but five in EV5d? Were only three of the five points from EV5d used to calculate data in Figure 6i, and if so, how were those chosen?

Errors or omissions:

1) In Figure 3c, the legend does not describe what the different shapes for the symbols represent (e.g. diamond, square, triangle). Also, the legend describes one dotted line. Which one is it (presumably green)? What is the other dotted line?

2) In Figure 6d, there are extra lines and a "ns".

3) Legend for 5f. It would appear that the ratios are $S2' / S2+S2'$, not what is written in the legend currently. Please clarify.

4) In figure legend 6i, the samples are described as 72 hours-post-infection, In line 495 of the text, it says 96 hpi. Please clarify.

5) Figure legends for Figure 5A+C lack info about the anti-PAI-1 treatment.

6) It would appear that Figure 6h does not have a legend and that the current legend marked (h) actually refers to panel i.

7) For figure 6F, the authors utilize E-64 but do not define what it is in the text or figure legend. This information should be added. Further, do the authors have a speculation on why E-64 is more effective than PAI-2 in the experiment?

- 8) For figure 6l and EV5C, please provide statistical tests and provide p-values.
- 9) The authors have figure EV5E that is not reference at all in the text. Can the authors incorporate this data into their text and explain how it differs from figure 6j?
- 10) Legend for 6j (currently labeled as 6i) indicates that it is GFP positive cells per well, but the y-axis is labeled % GFP positive/well. Please clarify.
- 11) Line 1433 has a typo.
- 12) Line 452 has a typo.
- 13) Line 504 should say "infectivity" rather than "activity".
- 14) Lines 519-520 refers to Figure EV5f.
- 15) Line 299, Figure 3e is the appropriate figure callout.

Referee #2:

Re-Review

Prior Title - In-silico docking platform with serine protease inhibitor (SERPIN) structures identifies host cysteine protease targets with significance for SARS-CoV-2 - Now modified to the following title

"Current title

Ref EMBOJ-2024-118196R-Q

Title In silico docking with full length 3D SERPINs identifies protease targets of human and viral origin

Authors - Joaquín J Rodriguez Galvan, Maren de Vries, Shiraz Belblidia, Ashley Fisher, Rachel A Prescott, Keaton M Crosse, Walter F. Mangel, Ralf Duerr, Meike Dittmann

Summary

The finding that PAI-1 has cross class serpin functions is of definite interest given the roles of PAI-1 in regulation of physiological processes, as a marker for pathology and disease progression as well as potentially in causing or driving pathology. Examining airway epithelial proteases and serpins is of definite interest in respiratory viral infections. Broadening the targets for serpins is also of very definite interest. This study ranges from an overview examining prior published work and further analysis of bronchoalveolar lavage samples from COVID infected patients and associated rtPCR analysis of serpin gene expression, a tissue culture analysis of lung epithelial cells and expression of proteases and serpins is assessed. In addition selective changes in serpin expression and protease targets are assessed by in silico analysis followed by proof of concept - Western blot and fluorometric analysis. This is an extensive study and of definite interest. Prior work has been extended to include further analysis of protease serpin interactions with greater focus on PAI-1

There are some limitations as with any study, albeit the study overall is excellent.

The bronchiolar lavage analysis has been extended to differentiate epithelial and immune cell responses.

Overall this is a very simple, extensive and excellent study analyzing a wide range of data from patients with bronchoalveolar lavage and epithelial cell samples, human epithelial cell cultures after infection with a range of respiratory viruses as well as the development of a 3D in silico analysis of potential protease and serpin interactions. These correlations between changes in serpin and protease expression with viral infections and potential new interactions for serpins with previously non identified proteases is of clear interest. This shows an excellent correlation but as with big data sets this does not provide a true cause and effect. - rather provides new potential directions for analysis of these interactions in viral infections in lung epithelium and this is valuable as a basis for ongoing studies.

Summary

This is an excellent and innovative analysis of serpins and specifically PAI-1 role in respiratory disease and virus infections. There is extensive experimental data well supported by the studies reported here.

The responses to a prior review are generally well prepared for each critique and provide very good responses with additional work both in silico and in vitro. I would support this paper for publication if the following critiques are carefully and clearly addressed.

Critiques

- 1) It would be very helpful if the modifications made in the text were marked by highlighting or text colour - The lines referred to in the review response do not always correlate with the lines in the merged pdf file
- 2) Abstract - The references to the SERPIN nomenclature are consistent - However, the commonly used serpin names such as PAI-1 PAI-2 etc are only noted in the abstract - Most readers likely are more familiar with the biologic names such as PAI-1 - I

would use both throughout to make reading the paper more accessible to a reader from outside the SERPIN field

3) Abstract lines 34, 35 "Additionally, we identify PAI-2 as the first direct-acting antiviral SERPIN, inhibiting adenovirus protease and reducing adenovirus infectivity." I do believe there was work examining benefits of the serpins C1Inh (C 1 esterase inhibitor) and A1AT (alpha 1 antitrypsin) as beneficial for COVID - Perhaps not direct acting? As noted by the authors in the Introduction lines 94-103 "To date, three SERPINs have been studied in the context of innate antiviral defense: PAI-1 (encoded by SERPINE1) against influenza viruses encoding hemagglutinin H1, Sendai virus, and SARS-CoV-2, through the inhibition of trypsin-like proteases by impeding proteolytic maturation of H1, F, or Spike, respectively (Dittmann et al, 2015a; Rosendal et al, 2022); alpha-1-antitrypsin (encoded by SERPINA1) and antithrombin (encoded by SERPINC1) against SARS-CoV-2, likely through the inhibition of TMPRSS2, by reducing maturation of Spike, although direct inhibition of TMPRSS2 protease activity or formation of "mousetrap" complexes by either SERPIN was not shown (Rosendal et al., 2022). ..."

Perhaps the authors wish to refer only to Adenovirus but this appears to ignore the other anti-viral effects - This does explain the focus in the abstract on potentially improving Adenovirus mediated therapy

4) Figure 2- the letters in panel a, Figure 1 - specifically - A1,B1, B9, F1,G1 etc are present labelling the graph but are not defined in the Figure legend nor in the Methods sections - ? Do these represent the individual lung lavage samples? This needs to be defined

5) The figure legends would be much improved if there was a statement at the beginning describing what is illustrated, and specifically what are the findings in each figure

6) How is the score for each in silico interaction derived?

7) The authors response re. cause and effect is somewhat simple - To actually prove a role for PAI-1 as protective in disease would require an animal model - however the further work with epithelial cell cultures provides a good basis. for example lines 390-392 - "PAI-1 given to virus infected cells in culture to assess response "Addition of active PAI-1 to the culture medium of Calu-3 cells significantly reduced the multi-cycle growth of both SARS-CoV-2 WA-1 and Omicron BA.1 compared to the buffer control and heat-inactivated (HI) PAI-1, achieving near-complete inhibition (Figure 5a-d)."

The current study provides the basis for ongoing work and the observation that PAI-1 has the capacity to function as a cross class serpin is intriguing. Given the extensive work demonstrating PAI-1 as a marker of disease progression, the current study might suggest PAI-1 represents a protective mechanism rather than driving disease - This should be noted

8) The limitations for in silico analysis might be more clearly stated - For example - In silico provides an excellent theoretical basis for protein protein interactions but is a theory based analysis of protein - protein interaction - How might this interfere with analyses? - This is in part explained and addressed in the results section lines 212-222 by the use of the HADDOCK program - as noted by the authors "However, many in-vitro models fail to replicate the diverse proteolytic landscape observed in vivo, limiting the effectiveness of these techniques for target discovery. In-silico approaches using motif searches are restricted by known protease motifs and do not integrate structural factors influencing SERPIN- protease interactions. To overcome this limitation, we used the software HADDOCK (High Ambiguity Driven protein-protein Docking (de Vries et al, 2010; van Zundert et al, 2016)) to predict interactions between SERPINs and proteases. HADDOCK predicts how two or more molecules, such as a SERPIN and a protease, interact to form a binding complex (in this case the binding complex depicted in Figure EV1a)." This explanation is good but still - in this reviewer's opinion - cannot fully address the complexity of multiple protein interactions in vivo in circulation or in the lung

The authors have introduced good controls with the use of known protease serpin interactions and also known proteases that do not interact lines 247-249 s follows "We created a panel of 10 SERPINs expressed upon viral infection (Figures 1, 2) and used HADDOCK to generate in-silico complexes with known target and non-target proteases for each."

9) Abstract lines 36-38 - "Our study leverages in-silico docking with full-length 3D protein structures to uncover new SERPIN targets, offering potential therapeutic interventions for viral infections and improvements in the production of adenovirus-based gene therapy vectors." Adenovirus expression was not assessed here. Certainly Adenovirus and AAV gene expression can be considered but what about simple protein therapy? Perhaps restate as "Our study leverages in-silico docking with full-length 3D protein structures to uncover new SERPIN targets, offering potential therapeutic interventions for viral infections"

10) Some references are superscript and some are normal text) - Seen in the Introduction - Sometimes the sentence period is provided before the refs in parentheses and sometimes after.

11) How are the HADDOCK scores calculated?

12) Arrows indicating specific changes in the histology sections Figure 2 a would be helpful

13) Figure 1 has a title and number but Figures 2 and onward figures do not

14) The in silico identified serpin protease interactions are confirmed by in vitro protease fluorescence assays presented in fig 4
- this is excellent

15) CTSL and CTSB abbreviations should be defined

16) The work with PAI-2 and Adenovirus is intriguing - Here the benefit of Adenovirus as a gene expression vector might be blocked by PAI-2 activity - Indicating another SERPIN mediated effect on viral replication, a virus used in therapeutic applications

17) Would the PAI-2 study perhaps provide a separate paper illustrating these findings these findings are somewhat less visible after evaluating all the prior data? This is however indeed an interesting finding and relevant to this paper. This is just a suggestion to perhaps focus the findings in the first figures.

Rev_Com_number: RC-2024-02448

New_manu_number: EMBOJ-2024-118196R-Q

Corr_author: Dittmann

Title: In-silico docking with full-length 3D SERPINS identifies protease targets of human and viral origin

Review commons EMBO J 2024 reviews (round 4)

Referee #1:

Summary:

Rodriguez Galvan et al. use a combined computational and functional approach to identify a new host target for the protease inhibitor, PAI-1 (SERPINE1), and show that exogenous PAI-1 can inhibit SARS-CoV-2 replication. They also identify that PAI-2 (SERPINB2) can inhibit adenovirus protease and adenovirus replication. They first use a COVID-19 dataset to identify SERPINS that are differentially expression in individuals with mild and severe COVID-19. They further use experimental infections of a model of human airway epithelium to identify SERPINS that are upregulated in response to several viruses, as well as treatment with interferon. Using this panel of SERPINS and a panel of host proteases, they use computational docking to predict SERPINS that may inhibit human proteases that may be relevant for viral infection. Using these predictions, they show that PAI-1 inhibits TMPRSS2 (previously shown) and CTSL (newly shown in this study), two proteases with relevance for SARS-CoV-2 infection. They show that extracellular addition of PAI-1 inhibits multicycle replication of SARS-CoV-2. In a second aspect of their work, they computationally predict and validate that PAI-2 inhibits the protease activity of adenovirus protease. They further show that cellular overexpression can inhibit adenovirus replication whereas knockdown of endogenous PAI-2 can increase the number of adenovirus-positive cells in culture.

This work is a substantial advance in the field. The data on SERPIN regulation during viral infection and host protease inhibition by SERPINS was already going to be an important addition to the field. The new data on adenovirus protease inhibition makes this manuscript even more interesting and exciting. The functional conclusions are for the most part well supported by data. Below are a few comments, both major and minor, that could be addressed with additional data. The majority of the comments below, however, are in regards to strengthening the manuscript by filling in critical missing information, softening some conclusions (mostly regarding the generalizability of the computational method), and cleaning up errors in the manuscript.

Major Comments

1) The generalizability of the computational modeling is still not full supported. This is not a problem that needs to be addressed with data necessarily, but rather a less black-and-white interpretation of their data. In response to our previous comments, the authors have now included definitions of likely binder and likely non-binder based on the new normalization methods utilized, and have provided additional test cases. However, it is still unclear, as a reader, how one would use HADDOCK to reliably predict SERPIN-protease interactions. For instance, beginning on line 600, the text states that HADDOCK modeling correctly identified CTSL as a binder and CTSS as a non-binder. However, the HADDOCK scores for CTSS are little different than uPA, which is a known binder but still above their dotted threshold line in Figure 3c. As a reader, how would I interpret the uPA data? Or what would I predict for CTSS, CTSG, etc? It would appear that the HADDOCK data can make nice predictions for new interactions such as PAI-2 and adenovirus protease, but there needs to be more discussion of the limitations of the predictions in the 'grey area' that is occupied by both binders and non-binders (presumably falling approximately between the two dotted lines in Figure 3c).

Thank you for pointing this out. *We have softened the language of the claims and addressed in-depth the 'gray' area in lines 633-648*

2) Likewise, the authors present a black-and-white interpretation of the interaction predictions with regards to 3D fit and specific sequence recognition. On line 329, they state "3D fit, rather than the recognition sequence, is crucial for SERPIN target protease recognition". This is demonstrably false based on their HADDOCK models and functional data that show that changing the recognition sequence impacts SERPIN target protease recognition. Their data do show that 3D recognition is important, and that should be rightly highlighted, but a more nuanced interpretation about the importance of both of these types of molecular interactions is warranted.

Point taken - maybe using the word "motif" would have been a better choice. Sequence and structure are obviously linked; what we wanted to stress is that previous methods looking at motif alone are limited, because they do not take structural fit into consideration. We now provided further explanation as to other biophysical factors relevant for the interactions and adjusted our interpretation accordingly. Please see lines 599-603 and lines 607-610.

3) The last figure panel (EV5f) and interpretations are entirely too preliminary and speculative. Using singular microscopy fields, the authors suggest that PAI-2 expressing cells may cause AdV to become trapped in the endosome. While it is true that there is colocalization of the AdV capsid proteins with endolysosomal markers, there are multiple instances of colocalization in the empty vector cells. Further, to truly say that the particles are trapped would require live imaging that tracks these particles through time since the particles are in the endosome at some point during their lifecycle. Overall, these data are too preliminary and require several more controls to suggest this mechanism, even with their caveat that these are preliminary. It would be better to remove this from the manuscript as it weakens the overall rigor of the otherwise well-supported conclusions.

We removed this data.

4) The blots in Figure 4c-e are not properly controlled and are therefore difficult to interpret. We previously raised a concern about Figure 4e, to which the authors responded that they know that the band that runs near their presumed SERPIN-protease complex is not PAI-1 or CTSL. It would be useful to see the data that support their statement. But the fact remains that in Figure 4d and 4e, the appearance of an upper band that the authors suggest is SERPIN-protease complex is only in lanes that contain more PAI-1 than the control lanes (compare lanes 1 and 4 in Figure 4d and lanes 1 and 2 in Figure 4e). Even in Figure 4c, the lanes (1 versus 4) are marked as having the same PAI-1, but the abundance by gel is visually quite different. In answer to their response to our previous concern, I would prefer they re-run these experiments with the proper controls if they wish to include these data to complement the already convincing functional data about SERPIN-mediated protease inhibition.

We have addressed this concern by re-running all of these experiments with the suggested controls. Please see the new Figure 4 panels c-g.

5) Was PAI-1, or any other SERPIN, tested against adenovirus protease? The interpretation

that PAI-2 is a specific inhibitor of adenovirus protease would be strengthened by testing at least one other SERPIN.

We addressed this concern experimentally as well, please refer to Figure EV5 panel c. The silver gel shows specificity with PAI-2 compared to two other SERPINs in forming (a) covalent complex(es) with active AVP.

6) There are quite a few claims of "firsts" throughout this manuscript. I do not know how EMBO considers these, but most journals discourage this in the way that discourage "novel" and "new". The manuscript is strong without making primacy claims about things like "the first comprehensive exploration of SERPIN targets ... (Lines 643-645)", which is subject to interpretation of the word "comprehensive" anyway.

Thank you for this suggestion. We have softened the language when appropriate but we have kept the primacy claims we believe are relevant to highlight novelty of this study.

7) Overall, there are many more mistakes in this manuscript than one would normally expect for a revised manuscript. Many of those are listed below. Individually, each is a minor comment, but together they become a major comment.

Point taken.

Minor Comments regarding data:

1) In response to author's response to major comment 10, while we do agree that the S2 is indirectly showing S1 cleavage, the intensity of the S2 is unreliable for quantifying how much of the original cleavage is occurring since it is subsequently cleaved. For the purposes of clarity of why the upper FL band is decreased in the PAI-1 condition, can the authors provide a loading control if the FL band difference is explained by loading difference?

Please refer to the new Figure 5 panels f and g, we have added GAPDH as a loading control and hope this clears this reviewer's concern.

2) Please include a similar graph as depicted in Figure 3C for PAI-2 since data is shown for it in Figure 4. This can easily be a supplement but would be helpful for readers to draw conclusions on the utility of HADDOCK predictions for these interactions, which is relevant for major comment 1 above.

We respectfully don't think such graph adds anything substantial to the manuscript, Figure 3 panel c was added to highlight PAI-1's specificity towards Cathepsin L compared to other proteases of the same family, and controls. We provide a similar comparison for PAI-2 and controls in Figure 6c. We believe the reader can visualize PAI-2's HADDOCK scores against any other protease directly from the heatmap provided in Figure 3. Moreover, we believe the reader will not benefit from replotting the PAI-2 data against cathepsins, as our major findings propose PAI-2 as a novel adenovirus protease inhibitor, and it will divert attention from the main findings.

3) Line 392, the authors state spike is not detectable from supernatant without showing any data to support that. It would be better to either include those data or omit this sentence.

We have deleted the statement. See line 386.

4) For figure EV5b, is there a reason why the protease band is not shown in this gel images? It would help the reader to interpret the data in much the same way that all other gel images in the paper show both SERPIN and protease.

The reason the protease band is not present is because this is a western blot probing for PAI-2 and not a silver gel. It shows that PAI-2 is present in the high-molecular-weight complex we see in Figure 6 panel g. We were not able to find a reliable antibody against AVP. A silver gel is now shown in new Figure EV5c.

5) Similarly, in figure 6G, the cropping of the gel is too close to the bands of interest, please provide a blot that is not cropped as much.

This concern has been addressed by re-cropping the gel in Figure 6g. The uncropped image is now part of the source data information.

6) Do the authors have titrating or genome copy measurements that correspond to with data in figure 6J? This would strengthen claims about utility for adenovirus production, since it is not clear how much of an impact siSERPINB2 has on adenovirus production practically.

We now added this experiment; please see the new Figure 6k.

Minor Comments regarding text and figures:

1) In line 151, the authors say with seeming certainty that, because SERPINS are upregulated in epithelial cells, they are involved in cell-intrinsic antiviral response. The authors should tone down the claim and write it as a speculative statement since not all SERPINS that are upregulated are examined in this manuscript. Similarly, line 173 makes a similar claim about the importance of the SERPINS produced in the airway during SARS-CoV-2 infection, tone down the claims as well. As a suggestion, the language in line 184 seems more appropriate.

Thank you, we have addressed this in lines 148-151 and in lines 169-173.

2) In line 488, the authors say cracked but earlier in the text they say sonicated AdV particles. If they are the same, they should use consistent terminology throughout.

We now use 'sonicated' consistently.

3) In Figure 1 and 2, it is strange to use ** for $p < 0.05$ as this is not the standard, nor is it what they use elsewhere in the paper.

Thank you, this has been fixed.

4) The data in Figure 6i are confusing in many ways. First, how is it possible to have a specific infectivity, which is described as infectious particles/genome copy, that is greater than 1 (e.g. 2^0 in Figure 6i)? I understand that the math from data shown in EV5c&d gives that answer, but it seems difficult to have more infectious particles than genomes. Second, why are there three data points in figure 6i and EV5c, but five in EV5d? Were only three of the five points from EV5d used to calculate data in Figure 6i, and if so, how were those chosen?

We appreciate this reviewer for catching this. We previously calculated genome copies per μ l of PCR input, not per μ l of supernatant like the TCID₅₀ value. We now calculate genome copy numbers/mL of supernatant, as seen in the new figure EV5d and used these values to calculate specific infectivity correctly in figure 6i. The data points used were selected randomly as they were sent to our collaborator who ran the AdV5 qPCRs.

Errors or omissions:

1) In Figure 3c, the legend does not describe what the different shapes for the symbols

represent (e.g. diamond, square, triangle). Also, the legend describes one dotted line. Which one is it (presumably green)? What is the other dotted line?

Thank you, we have now added this to the figure legend in lines 1374-1379.

2) In Figure 6d, there are extra lines and a "ns".

Thank you, we have corrected this mistake.

3) Legend for 5f. It would appear that the ratios are $S2'$ to $S2+S2'$, not what is written in the legend currently. Please clarify.

Thank you, we have corrected the explanation in the figure legend lines 1400-1404.

4) In figure legend 6i, the samples are described as 72 hours-post-infection, In line 495 of the text, it says 96 hpi. Please clarify.

The timepoint of collection of AdV supernatants is 96hpi, the timepoint of stopping the titration (TCID50) is 72hpi. We changed the legend to make this clear..

5) Figure legends for Figure 5A+C lack info about the anti-PAI-1 treatment.

Thank you for pointing this out, we have now added the relevant information to the figure legend in lines 1393-1396.

6) It would appear that Figure 6h does not have a legend and that the current legend marked (h) actually refers to panel i.

This has been corrected.

7) For figure 6F, the authors utilize E-64 but do not define what it is in the text or figure legend. This information should be added. Further, do the authors have a speculation on why E-64 is more effective than PAI-2 in the experiment?

We have added explanation and references of E-64 as a non-specific molecule that reacts with cysteines. See lines 467-469.

8) For figure 6l and EV5C, please provide statistical tests and provide p-values.

We provided it, but no statistical significance was achieved with specific infectivity.

9) The authors have figure EV5E that is not reference at all in the text. Can the authors incorporate this data into their text and explain how it differs from figure 6j?

This has been fixed, the explanation is added in lines 524-527.

10) Legend for 6j (currently labeled as 6i) indicates that it is GFP positive cells per well, but the y-axis is labeled % GFP positive/well. Please clarify.

This has been fixed.

11) Line 1433 has a typo.

This has been fixed.

12) Line 452 has a typo.

This has been fixed.

13) Line 504 should say "infectivity" rather than "activity".

This has been fixed.

14) Lines 519-520 refers to Figure EV5f.

This has been fixed.

15) Line 299, Figure 3e is the appropriate figure callout.

This has been fixed.

Referee #2:

Re-Review

Prior Title - In-silico docking platform with serine protease inhibitor (SERPIN) structures identifies host cysteine protease targets with significance for SARS-CoV-2 - Now modified to the following title

"Current title

Ref EMBOJ-2024-118196R-Q

Title In silico docking with full length 3D SERPINs identifies protease targets of human and viral origin

Authors - Joaquín J Rodriguez Galvan, Maren de Vries, Shiraz Belblidia, Ashley Fisher, Rachel A Prescott, Keaton M Crosse, Walter F. Mangel, Ralf Duerr, Meike Dittmann

Summary

The finding that PAI-1 has cross class serpin functions is of definite interest given the roles of PAI-1 in regulation of physiological processes, as a marker for pathology and disease progression as well as potentially in causing or driving pathology. Examining airway epithelial proteases and serpins is of definite interest in respiratory viral infections. Broadening the targets for serpins is also of very definite interest. This study ranges from an overview examining prior published work and further analysis of bronchoalveolar lavage samples from COVID infected patients and associated rtPCR analysis of serpin gene expression, a tissue culture analysis of lung epithelial cells and expression of proteases and serpins is assessed. In addition selective changes in serpin expression and protease targets are assessed by in silico analysis followed by proof of concept - Western blot and fluorometric analysis. This is an extensive study and of definite interest. Prior work has been extended to include further analysis of protease serpin interactions with greater focus on PAI-1

There are some limitations as with any study, albeit the study overall is excellent.

The bronchiolar lavage analysis has been extended to differentiate epithelial and immune cell responses.

Overall this is a very simple, extensive and excellent study analyzing a wide range of data from patients with bronchoalveolar lavage and epithelial cell samples, human epithelial cell cultures after infection with a range of respiratory viruses as well as the development of a 3D in silico analysis of potential protease and serpin interactions. These correlations between changes in serpin and protease expression with viral infections and potential new interactions for serpins with previously non identified proteases is of clear interest. This shows an excellent correlation but as with big data sets this does not provide a true cause

and effect. - rather provides new potential directions for analysis of these interactions in viral infections in lung epithelium and this is valuable as a basis for ongoing studies.

Summary

This is an excellent and innovative analysis of serpins and specifically PAI-1 role in respiratory disease and virus infections. There is extensive experimental data well supported by the studies reported here.

The responses to a prior review are generally well prepared for each critique and provide very good responses with additional work both in silico and in vitro. I would support this paper for publication if the following critiques are carefully and clearly addressed.

Critiques

1) It would be very helpful if the modifications made in the text were marked by highlighting or text colour - The lines referred to in the review response do not always correlate with the lines in the merged pdf file

Between original submission and submission 2, the manuscript underwent considerable changes, so that we felt highlighting differences was not useful. We have now added the appropriate highlights and line numbers for changes of submission 2 to submission 3.

2) Abstract - The references to the SERPIN nomenclature are consistent - However, the commonly used serpin names such as PAI-1 PAI-2 etc are only noted in the abstract - Most readers likely are more familiar with the biologic names such as PAI-1 - I would use both throughout to make reading the paper more accessible to a reader from outside the SERPIN field

Thank you for this suggestion. We decided to use "SERPIN" referring to the gene names (i.e. in transcriptional studies) and the protein names for functional studies. We state both names the first time a new SERPIN is referred to in the manuscript.

3) Abstract lines 34, 35 "Additionally, we identify PAI-2 as the first direct-acting antiviral SERPIN, inhibiting adenovirus protease and reducing adenovirus infectivity." I do believe there was work examining benefits of the serpins C1Inh (C 1 esterase inhibitor) and A1AT (alpha 1 antitrypsin) as beneficial for COVID - Perhaps not direct acting? As noted by the authors in the Introduction lines 94-103 "To date, three SERPINS have been studied in the context of innate antiviral defense: PAI-1 (encoded by SERPINE1) against influenza viruses encoding hemagglutinin H1, Sendai virus, and SARS-CoV-2, through the inhibition of trypsin-like proteases by impeding proteolytic maturation of H1, F, or Spike, respectively(Dittmann et al, 2015a; Rosendal et al, 2022); alpha-1-antitrypsin (encoded by SERPINA1) and antithrombin (encoded by SERPINC1) against SARS-CoV-2, likely through the inhibition of TMPRSS2, by reducing maturation of Spike, although direct inhibition of TMPRSS2 protease activity or formation of "mousetrap" complexes by either SERPIN was not shown(Rosendal et al., 2022). ..."

Thank you for pointing this out. We refer to 'direct' acting as a SERPIN that can directly inhibit a virus by targeting its encoded protease, rather than achieving inhibition indirectly through inhibition of host proteases, as has been previously shown in the literature.

Perhaps the authors wish to refer only to Adenovirus but this appears to ignore the other anti-viral effects - This does explain the focus in the abstract on potentially improving Adenovirus mediated therapy

Thank you.

4) Figure 2- the letters in panel a, Figure 1 - specifically - A1,B1, B9, F1,G1 etc are present labelling the graph but are not defined in the Figure legend nor in the Methods sections - ? Do these represent the individual lung lavage samples? This needs to be defined

Thank you, this has been addressed in line 1335.

5) The figure legends would be much improved if there was a statement at the beginning describing what is illustrated, and specifically what are the findings in each figure

We appreciate this suggestion, however, stylistically, we prefer to state the experiments done and not the conclusions drawn in the figure legends. We believe that the conclusions should be drawn by the readers themselves from the content of the Figures – we would not want to lead them on. For ease of reading, we do state findings in the headers of the results sections.

6) How is the score for each in silico interaction derived?

Please refer to lines 907-929 for a more detailed explanation.

7) The authors response re. cause and effect is somewhat simple - To actually prove a role for PAI-1 as protective in disease would require an animal model - however the further work with epithelial cell cultures provides a good basis. for example lines 390-392 - "PAI-1 given to virus infected cells in culture to assess response "Addition of active PAI-1 to the culture medium of Calu-3 cells significantly reduced the multi-cycle growth of both SARS-CoV-2 WA-1 and Omicron BA.1 compared to the buffer control and heat-inactivated (HI) PAI-1, achieving near-complete inhibition (Figure 5a-d)."

The current study provides the basis for ongoing work and the observation that PAI-1 has the capacity to function as a cross class serpin is intriguing. Given the extensive work demonstrating PAI-1 as a marker of disease progression, the current study might suggest PAI-1 represents a protective mechanism rather than driving disease - This should be noted

We have added a statement in lines 407-408.

8) The limitations for in silico analysis might be more clearly stated - For example - In silico provides an excellent theoretical basis for protein protein interactions but is a theory based analysis of protein - protein interaction - How might this interfere with analyses? - This is in part explained and addressed in the results section lines 212-222 by the use of the HADDOCK program - as noted by the authors "However, many in-vitro models fail to replicate the diverse proteolytic landscape observed in vivo, limiting the effectiveness of these techniques for target discovery. In-silico approaches using motif searches are restricted by known protease motifs and do not integrate structural factors influencing SERPIN- protease interactions. To overcome this limitation, we used the software HADDOCK (High Ambiguity Driven protein-protein Docking(de Vries et al, 2010; van Zundert et al, 2016)) to predict interactions between SERPINS and proteases. HADDOCK predicts how two or more molecules, such as a SERPIN and a protease, interact to form a binding complex (in this case the binding complex depicted in Figure EV1a)." This explanation is good but still - in this reviewer's opinion - cannot fully address the complexity of multiple protein interactions in vivo in circulation or in the lung.

Agreed; we make no such statement.

The authors have introduced good controls with the use of known protease serpin interactions and also known proteases that do not interact lines 247-249 s follows "We created a panel of 10 SERPINs expressed upon viral infection (Figures 1, 2) and used HADDOCK to generate in-silico complexes with known target and non-target proteases for each."

We have added a statement in line 224.

9) Abstract lines 36-38 - "Our study leverages in-silico docking with full-length 3D protein structures to uncover new SERPIN targets, offering potential therapeutic interventions for viral infections and improvements in the production of adenovirus-based gene therapy vectors." Adenovirus expression was not assessed here. Certainly Adenovirus and AAV gene expression can be considered but what about simple protein therapy? Perhaps restate as "Our study leverages in-silico docking with full-length 3D protein structures to uncover new SERPIN targets, offering potential therapeutic interventions for viral infections"

Thank you please refer to the new abstract.

10) Some references are superscript and some are normal text) - Seen in the Introduction - Sometimes the sentence period is provided before the refs in parentheses and sometimes after.

Thank you we have fixed this issue in the references.

11) How are the HADDOCK scores calculated?

Please refer to lines 889-895 for a more detailed explanation.

12) Arrows indicating specific changes in the histology sections Figure 2 a would be helpful
Figure 2a does not depict any specific changes other than to show the architecture of our HAECs. Therefore, we find this not to be necessary.

13) Figure 1 has a title and number but Figures 2 and onward figures do not

The figure title has been removed.

14) The in silico identified serpin protease interactions are confirmed by in vitro protease fluorescence assays presented in fig 4 - this is excellent

Thank you, we appreciate this comment, please also refer to the new figure 4 to find additional controls and silver gels not provided before.

15) CTSL and CTSB abbreviations should be defined

These abbreviations have been explained throughout the text. Thank you for pointing this out.

16) The work with PAI-2 and Adenovirus is intriguing - Here the benefit of Adenovirus as a gene expression vector might be blocked by PAI-2 activity - Indicating another SERPIN mediated effect on viral replication, a virus used in therapeutic applications

This is an interesting discussion point and we have considered it.

17) Would the PAI-2 study perhaps provide a separate paper illustrating these findings these findings are somewhat less visible after evaluating all the prior data? This is however

indeed an interesting finding and relevant to this paper. This is just a suggestion to perhaps focus the findings in the first figures.

Inclusion of PAI-2:AVP was an editorial decision.

Dear Meike,

Thank you for submitting a revised version of your manuscript. We have now received input from both original reviewers, who find that their main concerns have been addressed satisfactorily and now recommend acceptance of the manuscript.

Now there remain only a few editorial points that need to be addressed before I can extend official acceptance of the manuscript:

1. Please submit a complete author checklist, which you can download from our author guidelines (<https://www.embopress.org/pb-assets/embo-site/EMBO%20Press%20Author%20Checklist-1642513524327.xlsx>). Please insert information in the checklist that is also reflected in the manuscript. The completed author checklist will also be part of the Review Process File.

2. Please check that the funding information is correct and identical both in the manuscript and our online system. Currently, the Vilcek Institute of Graduate Biomedical Sciences, NYU Grossman School of Medicine Startup, the Office of Biological and Environmental Research of the U.S. Department Energy under Prime Contract no DE-AC0298CH10866 with Brookhaven National Laboratory are missing from our online system.

3. All Materials and Methods need to be described in the main text using our 'Structured Methods' format.

According to this format, the Methods section includes a Reagents and Tools Table (listing key reagents, experimental models, software and relevant equipment and including their sources and relevant identifiers) followed by a Methods and Protocols section describing the methods, ideally using a step-by-step protocol format. The aim is to facilitate adoption of the methodologies across labs.

Please download and fill our Reagents and Tools Table template (.docx), which you can find in our author guidelines:

<https://www.embopress.org/page/journal/14602075/authorguide#structuredmethods>

When submitting your revised manuscript, please upload it as a separate file choosing the file type "Reagent Table". The information currently provided in Key Resources Table could be adapted to this format.

4. Please rename Tables 1, 3 and 4 into Dataset EV1 - EV3. Please remove the legends from the manuscript text and add to the corresponding datasets in a separate tab/worksheet.

5. Tables 2 and 5 should be renamed Table EV1 and EV2. Please remove the legends from the manuscript text file and add to the corresponding table at the top of the page.

6. Please make sure that the tables and datasets are called out in sequential order.

7. Please remove BioRender disclaimer from figure legends and add to a dedicated section in the Methods section using the following format:

Graphics:

(some of the... OR Figure #... OR synopsis) Graphics were created with BioRender.com.

8. Source data for figures 2a (H&E and immunofluorescence) do not appear to fit to the main figure panels. Please check and correct as needed.

9. Source data for figure 4f are missing.

10. In our standard source data check, we have noted unexplained numerical duplications in the source data for a couple of figures. I have attached the corresponding files with the detected duplications labelled in colour. Please take a look and correct if needed. A brief explanation would be very helpful - I appreciate that these duplications can also occur due to specific measurement or calculation methods used.

11. Our data editors have flagged the following issues in figure legends that need correcting:

- Please provide the exact p values in the legends of figures 4A, B; 5A, C, E; 6J, K; EV2 E, F; EV5 E, F.

- Please indicate the statistical test used for data analysis in the legends of figures 1C, 2C.

- Please note that in figures 6J, EV5 E, F there is a mismatch between the annotated p values in the figure legend and the annotated p values in the figure file that should be corrected.

- Please indicate what */ **/ ***/ **** represents; if this represents p value(s), please indicate the statistical test used and where appropriate, specify the exact p value in the legend(s) of figure(s) 6H.

- Please provide information on the number and nature of replicates in the legends of figures 5A, C, E; 6H; EV2 E, F.

- Please define the error bars in the legends of figures EV2 E, F.

- Please note that scale bar and its definition are missing for figures 5B, D.

12. Papers published in The EMBO Journal are accompanied online by a 'Synopsis' to enhance discoverability of the manuscript. It consists of A) a short (1-2 sentences) summary of the findings and their significance, B) 3-4 bullet points highlighting key results and C) a synopsis image that is 550x300-600 pixels large (width x height, jpeg or png format). You can either show a model or key data in the synopsis image. Please note that the image size is rather small and that text needs to be readable at the final size.

13. The "Significance" section should be removed from the manuscript text and can be repurposed for the synopsis text.

With best wishes,

Ieva

Revision to The EMBO Journal should be submitted online within 90 days, unless an extension has been requested and approved by the editor; please click on the link below to submit the revision online before 18th Aug 2025:

Link Not Available

Referee #1:

The authors have addressed our concerns. This is a nice paper that is a substantial advance in the field.

Referee #2:

The responses to the prior review as well as the additional experiments performed and the modifications to the paper are thoughtful and detailed. We consider the manuscript acceptable for publication

Rev_Com_number: RC-2024-02448

New_manu_number: EMBOJ-2024-118196R1

Corr_author: Dittmann

Title: In-silico docking of human SERPINS identifies protease targets of host and viral origin

The authors addressed the remaining formatting issues.

Dear Meike,

Thank you for addressing the final editorial points. I am now pleased to inform you that your manuscript has been accepted for publication in the EMBO Journal. Congratulations with a nice study!

Before we forward your manuscript to our publishers, we would like to propose some minor edits in the manuscript title, abstract and synopsis (please see below and in the attached file). I have also written a short blurb that will accompany the title of your manuscript in our online table of contents. Please let me know if any corrections or adjustments are needed.

New title option:

Host cell and viral protease targets of human SERPINS identified by in silico docking

Blurb:

PAI-2 (SERPINB2) emerges as a direct inhibitor of a viral protease, protecting against infection by human adenovirus 5.

Synopsis:

Serine protease inhibitors (SERPINS) can protect against viral infection, but their molecular roles remain incompletely understood. This study combines in silico identification of antiviral SERPIN targets with experimental validation to identify roles of SERPINS, PAI-1 and PAI-2, in targeting essential proteases for the lifecycles of SARS-CoV-2 and adenovirus 5 (Adv5), respectively.

- Respiratory virus infection induces expression of SERPINS and proteases in patient airways and in cultured human airway epithelium.
- Full-length 3D structure-based docking predicts new binding candidates for antiviral SERPINS.
- PAI-1 (SERPINE1) suppresses SARS-CoV-2 infection by inhibiting Cathepsin-L and TMPRSS2.
- PAI-2 (SERPINB2) directly inhibits the Adenovirus 5 Protease (AVP) and suppresses Adv5 infection.

If you have any questions, please do not hesitate to contact the Editorial Office. Thank you for this interesting contribution to The EMBO Journal!

With best wishes,

Ieva
